# Exposing flaws of generative model evaluation metrics and their unfair treatment of diffusion models

**George Stein**[*] **Jesse C. Cresswell**[†] **Rasa Hosseinzadeh**[†] **Yi Sui**[†]
**Brendan Leigh Ross**[‡] **Valentin Villecroze**[‡] **Zhaoyan Liu**[‡]
**Anthony L. Caterini**[§] **J. Eric T. Taylor**[§] **Gabriel Loaiza-Ganem**[§]
`{george, jesse, rasa, amy, brendan, valentin.v,`
`zhaoyan, anthony, eric, gabriel}@layer6.ai`
Layer 6 AI

## Abstract

We systematically study a wide variety of generative models spanning semantically-diverse image datasets to understand and improve the feature extractors and metrics used to evaluate them. Using best practices in psychophysics, we measure human perception of image realism for generated samples by conducting the largest experiment evaluating generative models to date, and find that no existing metric strongly correlates with human evaluations. Comparing to 17 modern metrics for evaluating the overall performance, fidelity, diversity, rarity, and memorization of generative models, we find that the state-of-the-art perceptual realism of diffusion models as judged by humans is not reflected in commonly reported metrics such as FID. This discrepancy is not explained by diversity in generated samples, though one cause is over-reliance on Inception-V3. We address these flaws through a study of alternative self-supervised feature extractors, find that the semantic information encoded by individual networks strongly depends on their training procedure, and show that DINOv2-ViT-L/14 allows for much richer evaluation of generative models. Next, we investigate data memorization, and find that generative models do memorize training examples on simple, smaller datasets like CIFAR10, but not necessarily on more complex datasets like ImageNet. However, our experiments show that current metrics do not properly detect memorization: none in the literature is able to separate memorization from other phenomena such as underfitting or mode shrinkage. To facilitate further development of generative models and their evaluation we release all generated image datasets, human evaluation data, and a modular library to compute 17 common metrics for 9 different encoders at `https://github.com/layer6ai-labs/dgm-eval`.

## 1 Introduction

The capability of modern generative models to synthesize fake images that are seemingly indistinguishable from real samples has resulted in much public interest [90, 89, 94]. While the evaluation of such models has a longstanding history [10, 11], the unprecedented fidelity of modern synthetic images (e.g. [25, 93]) raises the question of whether the current tools in use by researchers are sufficient to measure the extent to which these models have truly learned the ground truth distribution, and whether striving to achieve state-of-the-art performance on current metric leaderboards provides optimal targets to drive further algorithmic progress.

Evaluating a single generated image is straightforward, since humans can act as the "ground truth" for determining realism. Evaluating the quality of a model as a whole is much more difficult. Beyond

---

[*][†][‡][§]Authors within each of the four groups contributed equally, see Appendix F for a list of contributions.

37th Conference on Neural Information Processing Systems (NeurIPS 2023).

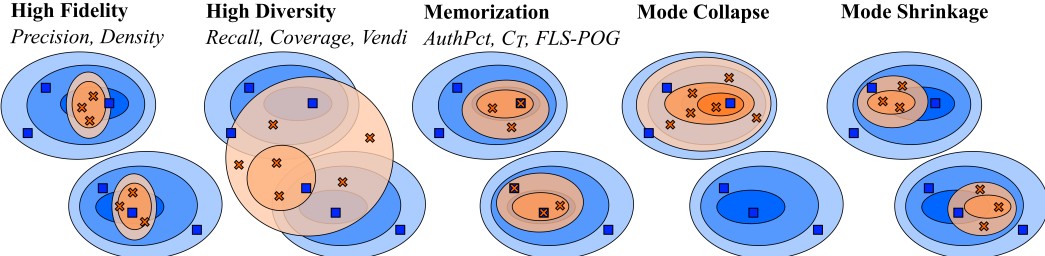

| High Fidelity | High Diversity | Memorization | Mode Collapse | Mode Shrinkage |
|---|---|---|---|---|
| *Precision, Density* | *Recall, Coverage, Vendi* | *AuthPct, $C_T$, FLS-POG* | | |

Figure 1: An illustration of learned distributions and samples (orange, crosses) having different properties with respect to the true distribution and training set (blue, squares). Italicized text indicates metrics that purport to detect these properties.

quantifying to what extent images resemble those from the training set (*fidelity*), we must determine how well the generated samples span the full training distribution (*diversity*), and whether they are truly novel or are simply reproductions of training samples (*memorization*), as illustrated in Figure 1. An ideal generative model will synthesize high fidelity and diverse samples without memorizing the training set (the latter becoming a prominent concern for models trained on unlicensed data [13, 101]). Researchers are well-practiced in ranking generative models by metrics such as the Fréchet Inception distance (FID) [48], Inception score (IS) [96], and many others [8, 76, 53, 91] which group fidelity and diversity into a single value without a clear tradeoff. Other popular diagnostic metrics separate sample quality from diversity such as precision/recall [95, 64] and density/coverage [75]. However, relating such metrics to human evaluation of image quality is not straightforward [120].

These metrics generally follow a two-step design: extract a lower-dimensional representation of each image, then calculate a notion of distance between true and generated samples in this space. The goal of the representation extractor, or encoder, is to embed images into a representation space that has a generalized perceptual relevance across the span of natural images. The implicit assumption in the de-facto use of the (pool3, 2048 dimensional) Inception-V3 network [103] trained for ImageNet1k [23] classification is that it provides such a space. Yet major concerns have been raised: it has been shown to be agnostic to features unrelated to the 1k classes of ImageNet [63], ImageNet classifiers in general are biased towards texture over shape [36, 47], and many other criticisms [120, 82, 72, 7].

Obvious choices for more universally applicable representation spaces are modern self-supervised learning (SSL) models [4] trained on large and diverse datasets, as they have proven to extract representations that excel at a number of generalized downstream tasks [19, 14, 20, 40, 15, 87, 44, 80]. While initial studies of a few self-supervised encoders reported that representations from these networks can produce more adequate rankings for generative adversarial networks (GANs) [38] on non-ImageNet domains [72, 7], it is an open question as to which SSL methods and families provide the best perceptual representation space for evaluating natural images more generally. For example, SSL methods based on contrastive learning strategies utilizing strong augmentations tend to learn features that are more invariant to those augmentations [30]. Thus, while the criteria for choosing an SSL encoder for classification tasks is straightforward – choose one that achieves strong linear classification accuracy – it is not clear that such a model will extract a general representation for generative evaluation rather than one that over-relies on object-based semantic information.

Understanding the interdependence of evaluation metrics, representation extractors, and their relation to human evaluation of generated images requires a large scale study of each component across a diverse set of datasets. Here we select 41 state-of-the-art generative models spanning diffusion models [100], GANs, variational autoencoders (VAEs) [60], normalizing flows [92, 26], transformer-based models [9], and consistency models [102], and generate 4.1M images to provide such a study:

**Human evaluation** We designed and funded extensive human subject experiments to establish a robust baseline for generated image fidelity, and find that $(i)$ no current metric strongly correlates with human evaluators, and that $(ii)$ diffusion models significantly outperform GANs and all other generative techniques at producing images that are indistinguishable from training data.

**Self-supervised representations and evaluation metrics** We show that $(iii)$ the Fréchet distance, kernel distance (KD) [8], precision, and density calculated with the Inception-V3 network do not correlate well with human evaluation. We then investigate the semantic information distilled from self-supervised methods spanning a wide variety of families, showing that $(iv)$ the perceptual qualities

of their representation spaces can strongly depend on training procedure and architecture, that $(v)$ supervised networks do not provide a perceptual space that generalizes well for image evaluation, and that $(vi)$ replacing Inception-V3 with DINOv2 ViT-L/14 [80] *solves the discrepancy with human evaluators* while the previously proposed SwAV and CLIP-B/32 replacements [72, 7] are sub-optimal.

**Diversity, rarity, and memorization** By leveraging the recently proposed Vendi [33] and rarity [42] scores, we show that $(vii)$ the discrepancy between human evaluators and FID is not due to models trading off fidelity for diversity, nor to human evaluators assessing rare images as fake. We see these results as evidence that human error rate is a sensible "ground truth" to align FD metrics with. Finally, we answer the question: are the best performing models according to our DINOv2-based metrics memorizing their training data? In doing so, we $(viii)$ find clear evidence of memorized samples across models, particularly on CIFAR10, and show that current memorization metrics and tests fail to capture this [70, 1].

**Summary** Our multifaceted investigation of generative evaluation shows that diffusion models are *unfairly punished by the Inception network*: they synthesize more realistic images as judged by humans and their diversity more closely resembles the training data, yet are consistently ranked worse than GANs on metrics computed with Inception-V3. While FID is already known to have shortcomings, we advocate for a complete replacement of Inception-V3 in all evaluation metrics of images, and show that DINOv2-ViT-L/14 allows for much richer evaluation of generative models.

## 2 Datasets, metrics, and encoders

**Generated datasets** We investigate a wide range of generative models trained on a diverse set of image datasets (CIFAR10 [61], ImageNet1k [23], FFHQ [58], LSUN-Bedroom [115]) using a variety of generative techniques (diffusion, GAN, VAE, normalizing flow, Transformer-based, consistency). For each dataset we include current state-of-the-art models as ranked by FID, as well as models spanning different generative procedures. We include 13, 11, 9, and 8 models for the respective datasets listed above, for a total of 41 generated datasets. To decouple the effects of model architecture/training procedure from the training data used, we focus only on generative models that did not include any data external to the respective dataset during training. Models for ImageNet1k, FFHQ, and LSUN-bedroom were trained at a resolution of $256{\times}256$. We chose to generate 100k images from each model, with an equal number of images per-class for class conditional models, and used the checkpoints and hyperparameters that achieved the lowest FID for each model; Appendix A details the full generation procedure. In total we assess each generative model across 17 metrics. We group the metrics by category here and provide full definitions in Appendix B.

**Metrics for ranking generative models** We include the well-known FID [48], which computes the Fréchet distance (FD) between sets of 50k real and generated samples in the representation space of the Inception-V3 network. We study FD in several alternative representation spaces, and refer to these by the model used to extract the representation from each image, e.g. $\text{FD}_{\text{CLIP}}$. We include alternatives to the FID such as the spatial FID (sFID) [76], $\text{FID}_\infty$ [22], IS [96], kernel Inception distance (KID) [8], and the feature likelihood score (FLS) [53].

**Metrics for diagnosing fidelity, diversity, rarity, or memorization** As proxies for sample fidelity we consider precision [95, 64] and density [75], while our human evaluation baseline, human error rate, provides a direct measurement of fidelity. To quantify sample diversity we consider recall [95, 64] and coverage [75], and for sample rarity we consider the rarity score [42]. To study inter- vs. intra-class diversity for class-conditional image generation we utilize the Vendi score [33]. While recall and coverage can also be determined per class, the small number of generated samples available for each class (e.g. 100 for each ImageNet class) results in difficulty constructing robust nearest-neighbour-based estimates. We also investigate a form of overfitting that we term *memorization*, in which models memorize individual images from their training data and emit them at generation time. We perform a direct check for pixel-wise memorization for each generative model for each of our datasets and report this value as the memorization ratio in Section 5.2. We include automated metrics which claim to isolate memorization from the effects of overfitting or mode collapse: the percentage of authentic samples [1], $C_T$ score [70], and the percentage of overfit Gaussians from FLS [53].

**Representation spaces for generative evaluation** With the aim of finding a more general perceptual representation space across the span of natural images, we employ a number of alternative encoders beyond the standard Inception-V3 [103] trained for supervised classification on ImageNet. First, as a

more modern supervised benchmark we use a ConvNeXt-large architecture trained on ImageNet22k [67], a larger dataset with more classes. We also include a number of self-supervised feature extractors as alternatives to supervised learners. While such networks have proven to be useful for concurrent vision tasks including classification, object-detection, and segmentation [14, 119, 5], it remains an open question as to how the objective and augmentations used for training affect the representation space for generative evaluation. Thus we include seven self-supervised methods from a variety of families [4] – contrastive (SimCLRv2 [19]), self-distillation (DINOv2 [80]), canonical correlation analysis (SwAV [14]), masked image modelling (MAE [44] and data2vec [3]), and language-image (CLIP [87], using the OpenCLIP implementation [51] trained on DataComp-1B [35]). We also consider DreamSim [34], an ensemble of three self-supervised models (DINO [15], CLIP, and OpenCLIP) that have each been fine-tuned to better align with human perception of image similarity using a dataset of human similarity judgments over image pairs. We design experiments to qualitatively and quantitatively understand their respective feature spaces.

For CNN-based models we use the ResNet50 [45] architecture while for vision transformers (ViT) [28] we use ViT-L/14 as we found it provides a good tradeoff between representation quality and computation cost. For both we select weights that were trained with the dataset and hyperparameters that achieved the highest linear classification on ImageNet. Full details are in Appendix B.4.

# 3   Human evaluation of generated data

The goal of our human subject experiments is to establish a large-scale, scientifically grounded baseline for image generation fidelity. Our experiments specifically target image realism, not the diversity of a model's learned distribution, nor whether samples are memorized, as realism is unequivocally a property for which humans can provide "ground truth". We find that the vast majority of models tested per dataset can be separated in terms of their realism with statistical significance, thus providing a clear ranking which can be compared to calculated metrics like FID. To our knowledge, this is the largest human subject experiment on the evaluation of generative models performed to date, with over 1000 paid participants, and 207k individual responses collected.

**Experimental design**   We evaluated human perception of generated image realism for each of the 41 models across 4 datasets described in Section 2. Our experimental design was informed by experts and best practices in psychophysics, the study of the human perceptual system. It follows the design of experiments for the $\text{HYPE}_\infty$ metric [120], with several modifications to increase the quality of collected data and expand the scope of models and datasets tested. Each trial is a two alternative forced choice task where a participant is shown either a generated image from one model, or an image from the training dataset, and must choose if it is real or fake. Models were evaluated based on *human error rate* [120], the fraction of images which were incorrectly classified. This is a simple metric with the intuition that models with better fidelity produce images that are more difficult to distinguish from real images. We take human perceptions of image realism as ground truth, noting the expansive efforts of the community to generate images that appear photo-realistic to humans, and will be used by humans. Full design details are provided in Appendix C.

Concurrent work has also targeted human perceptual benchmarks for image synthesis on a smaller subset of GAN and diffusion models [114] with 10 times fewer participants. While we are excited to see human benchmarks gaining popularity, we note a number of concerns with their methods and thus downstream conclusions. Specifically, observers had no training period nor knowledge of the training set, and were tasked to judge whether images were "photo-realistic". We believe this task contains much more ambiguity than our two alternative forced choice assessment, and introduces various response biases into participants' judgments. In their second user study, observers were asked to evaluate the relative realism of sets of images. This task is difficult to evaluate because participants can use a single image in the set to guide their decision for the entire set.

**Results and analysis**   The main results of our experiments are shown in Figure 2. We plot the mean human error rate along with standard error for each model, and sort models on the $x$-axis by their FID (lower is better). If FID correlated well with human perception of fidelity, each plot would have a monotonically increasing trend, but this does not appear to hold. By inspection, the models with highest fidelity are almost always diffusion models, although GAN models often have lower FID. We find similar results for alternatives to the FID score in Appendix D.1.

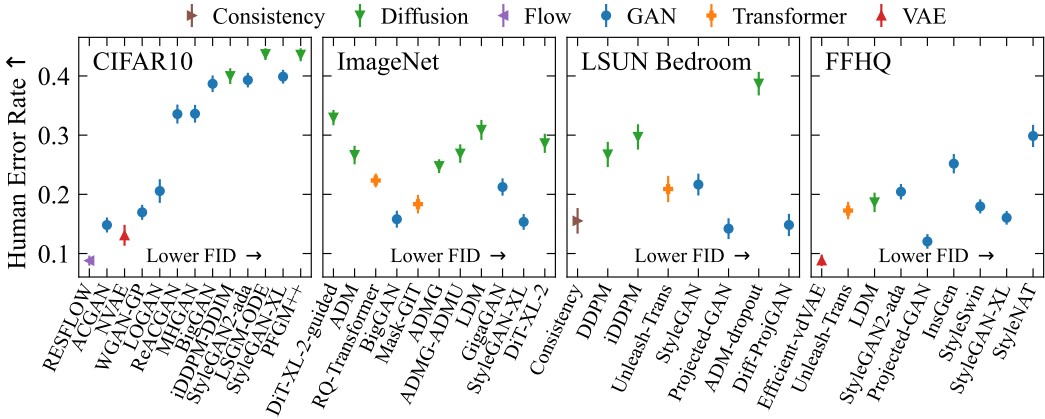

Figure 2: Human error rate on models ranked by FID. Data is displayed as the mean across participants with error bars showing the unbiased standard error.

To formally assess the relative performance of different model types, we separate participants' mean error rate into four one-way analyses of variance (ANOVA) – one per dataset – with model type (e.g. GAN, diffusion) as a between-subjects variable. We also performed planned comparisons between model types to probe omnibus effects. Analysis revealed effects of model type in all four ANOVAs, indicating significant differences between model types' effect on human error rate (all $F$'s > 12.09, all $p$'s < 0.001 [77]). Post hoc comparisons of mean error rate between model types using Bonferroni correction are shown in Table 1, where > indicates a significant difference, = indicates no significant difference, and model type is listed in order of descending mean error rate. Diffusion models were a clear standout for all datasets except FFHQ, where they performed on par with other model types (noting that only GANs appeared in more than one experiment on FFHQ). Coupling the results in Table 1 with the FID rankings in Figure 2, we conclude that current diffusion models produce the most realistic images according to human perception, but are downranked by FID.

## 4 Improved representation spaces for generative evaluation

### 4.1 Qualitative examination of perceptual spaces

To qualitatively visualize what parts of an image the Fréchet distance "perceives", we follow the gradient based visualization technique of [63], which focused on FID. Here we adapt and apply it to each of our CNN and ViT encoders. For CNN encoders our method is identical, while for ViTs we use the Grad-CAM variation introduced by [37]. Experimental details can be found in

Table 1: Model types ranked by human error rate.

| Dataset | Error rate ranking | | | |
|---|---|---|---|---|
| CIFAR10 | Diff. > GAN | > VAE | > Flow | |
| ImageNet | Diff. > Transf. | = GAN | | |
| LSUN | Diff. > Transf. | = GAN | = Consistency | |
| FFHQ | GAN = Diff. | = Transf. | > VAE | |

Appendix D.2.1. Figure 3 shows two visualized samples from each high-resolution dataset. We find qualitative differences between CNN and ViT architectures – ViTs have a more global receptive field [88] – but also find starkly different characteristics between supervised and self-supervised models. In agreement with [63], regions deemed important by Inception are far from optimal for datasets outside of the ImageNet domain. For FFHQ the important features according to Inception are typically not part of the person's face, while for LSUN-Bedroom the focus is on a single object in the scene; features simply correspond with the Inception model's top-1 class prediction. We find that Inception does not perceive a holistic view of images even on its ImageNet training set, and see similar characteristics for ConvNeXt, indicating that such glaring issues are not mitigated by supervised training on more modern architectures, larger datasets, nor a larger numbers of classes.

Meanwhile SwAV and SimCLR often ignore important features, with SwAV exhibiting the strongest overlap with the classification networks. CLIP puts a large focus on the few main objects of an image – typically the main facial features (lips, eyes, etc.) or objects (beds) – which we presume is an outcome of the language-image pretext arising from image captions that do not describe texture or finer details of the image. DINOv2 usually focuses on the image structure as a whole while still identifying objects of importance. We argue that this is closer to the behaviour we would hope to

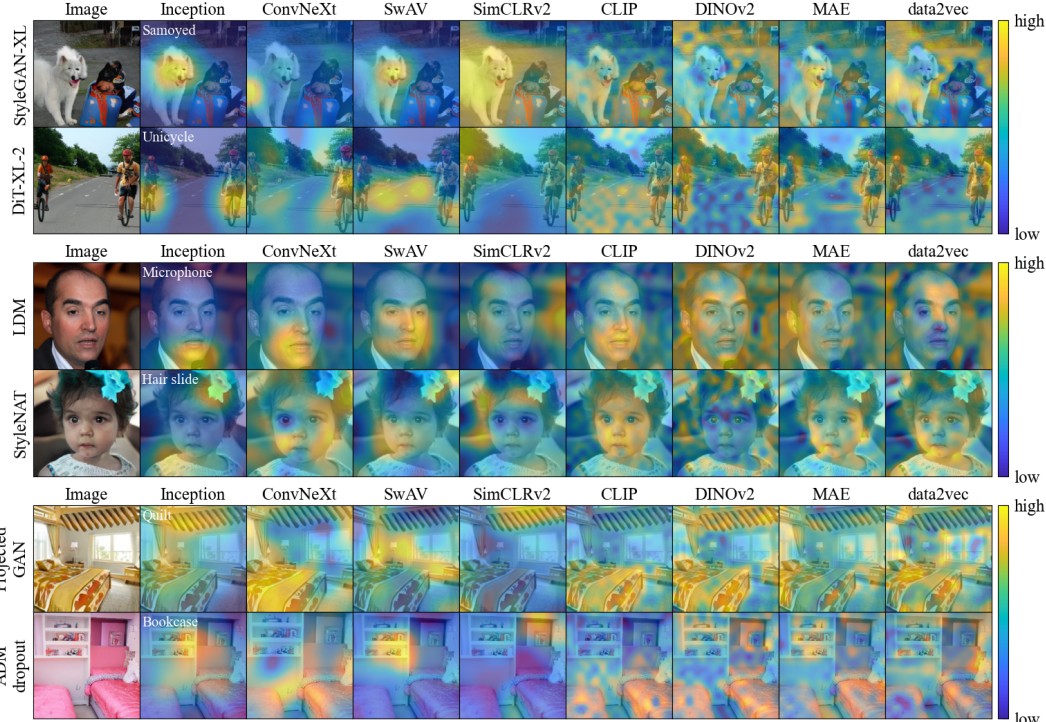

Figure 3: Heatmaps visualizing what the Fréchet distance "perceives" for each encoder. The sign of the heatmap is given by the activations of the saliency layer that is visualized and does not reflect the sign of the gradient w.r.t. the FD – both bright yellow and deep blue can thus show an encoder's focus. Additional examples are shown in Appendix D.2.1.

have from an encoder meant to evaluate images, as it emphasizes the important objects while still being able to pick up on elements elsewhere. MAE and data2vec, both trained using masked image modelling, have a widespread focus on textures and shapes, with an often smaller importance for the semantic information related to the main object. We exclude DreamSim from this analysis as it is not straightforward to use Grad-CAM on its output, which is the concatenation of the output of multiple encoders. Appendix D.2.2 includes quantitative analyses, finding that masked models put more weight towards low-level image features rather than clustering classes by object semantics, while others (CLIP in particular) distill a more object-focused representation space – both in alignment with the qualitative analysis shown here.

In summary, we conclude that the representation spaces of self-supervised methods are more appropriate for generative evaluation than supervised approaches, and that self-distillation (DINOv2) provides the best balance between focusing on important objects and holistic image structure.

## 4.2 The (mis)alignment of evaluation metrics and human assessment

In conjunction with our human evaluation baseline, we evaluated the 17 ranking and diagnostic metrics outlined in Section 2 for each encoder and generated dataset. Figure 4 (top) shows the relation of human error rate with FD and precision. We investigate diversity further in the following section. We include the correlation of 7 common metrics and human error rate (bottom, best viewed while zoomed in). Note that across encoders, FD, $FD_\infty$, and KD are very highly correlated, resulting in essentially the same model rankings. This suggests that all these metrics provide sensible ways of quantifying distances between probability distributions, provided a good encoder is chosen. Overall, we find that despite some sample complexity issues, the FD metric – when paired with an appropriate encoder – provides a strong way of evaluating generative models, see Appendix D.3 for details.

We find no strong correlation between human evaluation and any common metrics computed in the Inception representation space outside of the simplistic CIFAR10 dataset, showing that Inception fails to encode perceptually relevant features for the larger, more diverse, and complex datasets that are the testbeds driving advancement of generative model development. In agreement with the previous section we find relatively poor performance of the SwAV model, which was proposed as an alternative

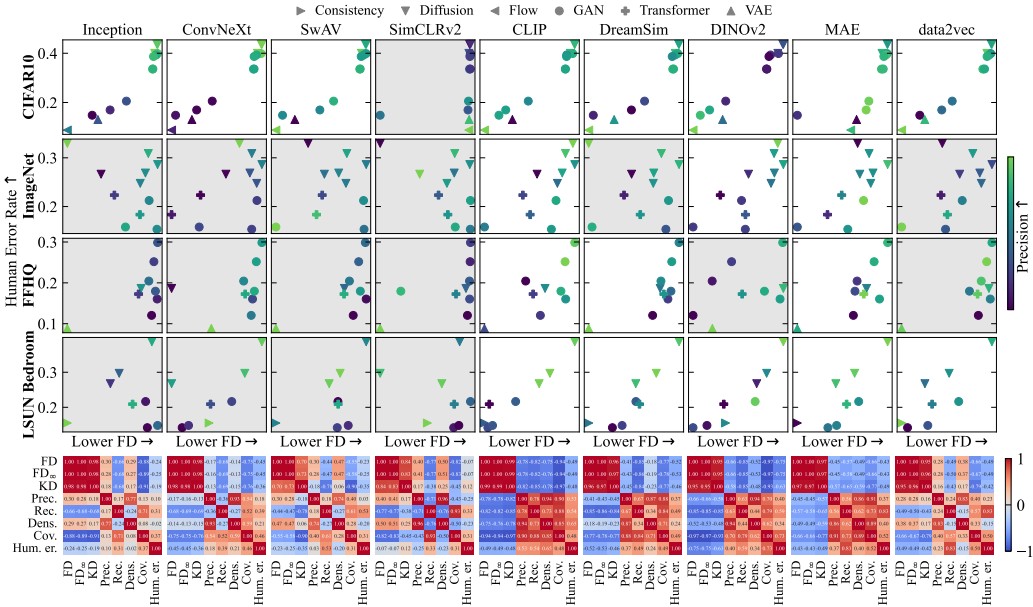

Figure 4: **Top**: Fréchet distance, precision, and human error rate for each generative model as measured by different encoders (columns) on different datasets (rows). Marker styles denote different generative techniques. Panels with a shaded background do not have strong ($|r| \geq 0.5$) and significant ($p \leq 0.05$) correlations between FD and human error rate. **Bottom**: Pearson correlation of metrics over the three high-resolution datasets.

to the Inception model in [72] (in Appendix B.4.1 we also show poor alignment with a smaller CLIP-B/32 model investigated in a number of toy examples in [7]). CLIP VIT-L/14, DINOv2, and MAE display far greater alignment with the human experiment baseline, as does DreamSim, except on ImageNet – which we believe to be a particularly important dataset, as it is the most complex one being considered here, and it is thus used to train the most realistic models. Note that large precision values (which aims to quantify fidelity) do not consistently correspond to high human error rate, even for encoders whose FD strongly correlates with human evaluation: precision is thus likely measuring more than just fidelity, and FD should be preferred over it. We include analogous results for recall in Appendix D.1, showing that it does not only capture diversity.

We find that diffusion models are often driving the discrepancy in alignment with human evaluation for the Inception network: FID prefers GANs over diffusion while FD determined by self-supervised models trained on very large and diverse datasets does not. This discrepancy does not only occur on non-ImageNet benchmarks, as previous works have shown for GANs, but is also true on ImageNet.

## 5   Alternative explanations: diversity, rarity, and memorization

We have found that FID does not correlate with human error rate, and have shown that replacing the Inception-V3 network with an SSL encoder such as DINOv2-ViT-L/14 both recovers the correlation of FD with human error rate, and results in a metric which qualitatively focuses on the more relevant parts of images. While these are very promising characteristics for an evaluation metric, alternative explanations need to be ruled out before such a metric can be confidently adopted as a community standard. In this section we first verify that the lack of correlation between FID and human error rate is not due to models with large human error rates lacking diversity, nor to humans wrongly classifying rare real images as fake. This justifies our use of human error rate as "ground truth" for fidelity, and confirms that a lack of alignment with human error rate is a flaw of the encoder/metric pair. Then, we investigate whether the best performing generative models are memorizing their training data.

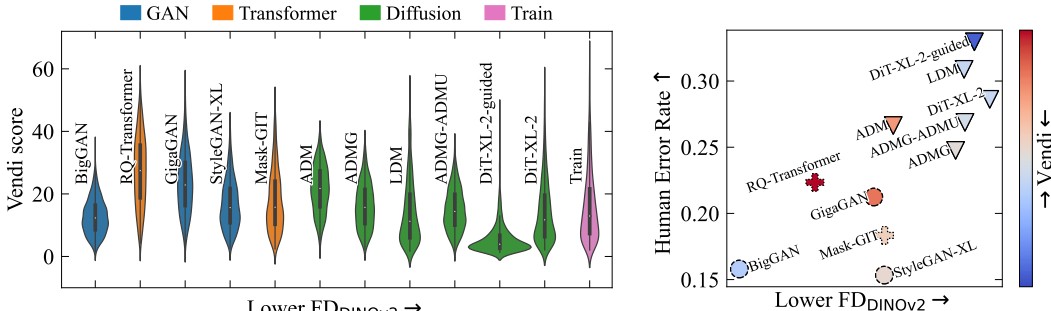

Figure 5: **Left**: Per-class Vendi scores of ImageNet models, in decreasing order of $FD_{DINOv2}$ score. **Right**: $FD_{DINOv2}$ on ImageNet, coloured by average per-class Vendi score (white corresponds to the train dataset).

## 5.1 Diversity and rarity

**Diversity** FD-based metrics combine both fidelity and diversity into a single score, whereas human evaluation focuses only on the former. We must thus independently measure model diversity in order to confirm whether the lack of strong correlation between FID and human evaluation is due to the FID score being flawed as an evaluation metric which focuses on fidelity, or if the discrepancy can simply be explained by high fidelity models having worse diversity. To decide between these two alternatives we explore the extent to which $FD_{DINOv2}$ and FID align with diversity measures.

Our diversity analysis focuses on the Vendi score [33]. We justify this choice in Appendix D.4 and verify that the Vendi score meaningfully quantifies diversity locally, but the same is not true globally. For example, Figure 6 displays samples from a DiT model on ImageNet with and without strong classifier-free guidance (cfg=4 and 1.5, respectively; we refer to the former model as DiT-guided), where it is evident that DiT-guided exhibits much lower per-class semantic diversity. Yet, Appendix D.4 shows that the overall Vendi score is higher for DiT-

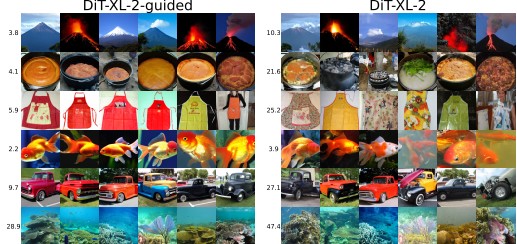

Figure 6: DiT-guided and DiT samples, labelled with per-class Vendi scores using DINOv2.

guided for almost all choices of encoders, while per-class Vendi scores are consistently lower for DiT-guided across encoders. These results justify the use of the per-class Vendi score as a sensible diversity metric; the overall Vendi score is mostly measuring inter-class diversity – which is not particularly meaningful for class-conditional models such as the ones commonly used on ImageNet – whereas the per-class scores focus on intra-class diversity consistently across encoders, and thus provide a more meaningful quantification of semantic diversity.

Equipped with the per-class Vendi scores, we evaluate the diversity of ImageNet models using the DINOv2 encoder in Figure 5 (left), where we can see that differences in diversity do not explain discrepancies between FID and human evaluations. For example, GigaGAN has diversity scores which are much farther away from those of the training data than most diffusion models, yet achieves a better FID (Figure 2). We see this as strong evidence of a limitation of the use of the Inception network to measure fidelity with FID. We perform the same analysis using the FD metric with the DINOv2 encoder in Figure 5 (right): this evaluation metric is not only much more correlated to human evaluators, but diversity also better explains the few discrepancies between the two, e.g. the lack of diversity in DiT-guided results in a worse $FD_{DINOv2}$ score than DiT despite having a better human error rate. Nonetheless, we highlight that the $FD_{DINOv2}$ score emphasizes fidelity more than it does diversity (e.g. the DiT-guided model still obtains a very strong score).

**Rarity** To ensure that participants are not confusing "unrealism" with "unlikeliness" and assessing rare images as fake – which would result in more diverse generative models ranking worse on human error rate – we investigated whether the human error rate on each real image (individual images were evaluated by an average of 13 humans) was correlated with the image's "rarity score" [42]. Experiments are detailed in Appendix D.4, which show that human evaluators are *not* confusing

"unrealism" with "unlikeliness", and thus human error rate is a sensible ground truth for image fidelity. Combined with the diversity analysis above, this rules out diversity as an alternative explanation for the lack of alignment between human assessment and FID, and proves that alignment of FD metrics with human error rate is a desirable property for the studied generative models.

## 5.2 Memorization

Recent works have shown that diffusion models are particularly prone to *memorization* issues, in which models memorize individual images from their training data and emit them at generation time. Memorized samples may be near-pixel-wise identical, or semantically equivalent to their source object while differing in terms of pixel-wise identity, the latter termed *reconstructive memory* [101]. Both predominantly occur either when the training set is small [13] or when there are a number of duplicate training samples [101]. While it has been shown that diffusion and GAN models can memorize CIFAR10 samples [101] – a dataset that contains many duplicates [6] – to our knowledge it is currently unknown whether this occurs on larger, more diverse, and higher resolution datasets such as ImageNet, FFHQ, or LSUN-Bedroom, and whether this affects any of the metrics we report. We set out to investigate this here.

Our experiments (refer to Appendix D.6) suggest that the larger datasets (e.g. ImageNet) are not memorized by even the largest models we considered. This indicates that the models that are measured as superior in DINOv2 space are not capitalizing on memorization of the training data.

**Collecting memorized samples** We perform a direct check for pixel-wise memorization for each of the 100k images from each of our 41 generative models using the calibrated $l_2$ distance proposed in [13]. We find strong evidence on CIFAR10 that most generative models exhibit exact memory and showcase a set of memorized samples in Figure 7. We also report the memorization ratio of all models – the proportion of generated samples that match source samples in the training set – to illustrate degrees of exact memorization across different models. We found no conclusive evidence of pixel-wise memorization by any model on ImageNet, FFHQ, or LSUN-Bedroom, but found evidence of models exhibiting reconstructive memory on ImageNet and LSUN-Bedroom [101]. We find that less than 0.5% of DiT-XL-2 images exhibit close reconstructive memory. Although reconstructive memory on complex datasets is not necessarily a major concern, we recommend monitoring the memorization ratio, especially as models become more powerful and pixel-wise memorization becomes more likely. We include examples in Figure 7 (left) and additional visualizations and details in Appendix D.6. We note that our study was not explicitly designed to detect a more "copy-paste" approach wherein models memorize aspects of different training images and combine them when sampling, and thus we cannot rule out all forms of memorization. We leave such investigations for future work.

**Evaluating memorization metrics** We evaluate the main memorization metrics in the literature in light of the memorized CIFAR10 samples we uncovered: the percentage of authentic samples (AuthPct) [1], the $C_T$ score [70], and the percentage of overfit Gaussians in FLS (FLS-POG) [53]. We first measure each metric's sensitivity to memorization in a controlled experiment, where we sample from the approximate posterior at different depths of a VDVAE's [21] hierarchical structure to generate a collection of synthetic datasets that serve as increasingly less faithful reconstructions of the training set. Appendix D.6.1 contains the full description and results, and establishes that AuthPct, the $C_T$ score, and FLS-POG are sensitive to memorization in an ideal scenario where all samples become increasingly memorized. In Figure 7 (right) we investigate whether these automated metrics can differentiate models in practice, based on their measured CIFAR10 memorization ratios. Here we used the DINOv2 representation space, while results for other encoders are shown in Appendix D.6.2. For most encoders, the $C_T$ score trends in the correct direction, whereas AuthPct and FLS-POG trend inconsistently and fail to differentiate between models with different numbers of memorized samples.

By swapping the training set with a *test set* that the model cannot possibly memorize, we check whether memorization metrics are sensitive to confounding properties other than memorization, such as fidelity or mode collapse. This experiment is not possible for FLS-POG as it does not use the training set, and for the $C_T$ score we split the test set into two as it is already used. The results for $C_T$ and AuthPct on test data are depicted with low opacity in Figure 7 (right). Surprisingly, in both cases the results against the test set closely follow those against the training set, meaning that the $C_T$ score and AuthPct are dominated by some property other than memorization. Based on our analysis in Appendix D.6.3, we postulate that these metrics focus more on mode shrinkage and image fidelity

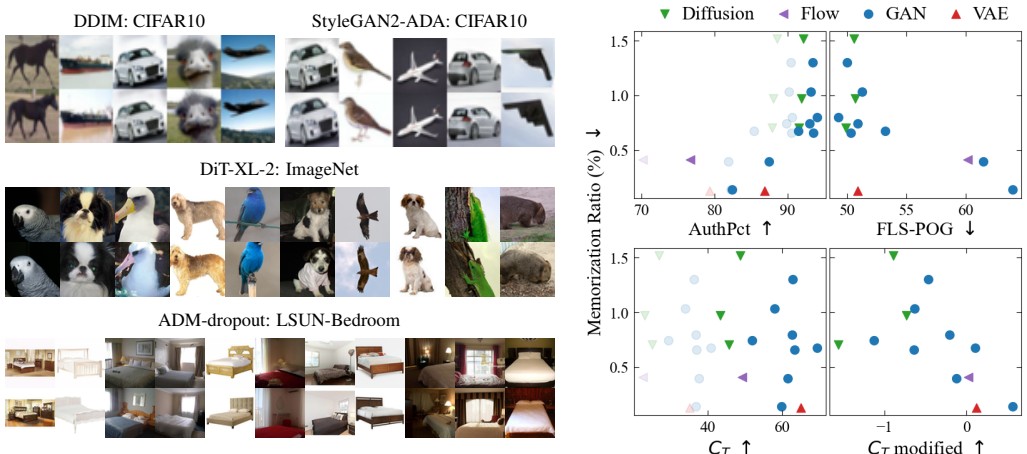

Figure 7: **Left**: Generated samples on top with matched training samples on the bottom showcasing exact memorization on CIFAR10 and reconstructive memorization on ImageNet and LSUN-Bedroom. **Right**: CIFAR10 models plotted by memorization ratio vs. metric. Memorization metrics against the test set instead of the training set are shown with low opacity.

(see Figure 1), respectively. From this analysis, we note that modifying the $C_T$ score by swapping the roles of the training and generated datasets makes it insensitive to mode collapse. We include this modification of the $C_T$ score in Figure 7 (right).

In conclusion, we find that none of AuthPct, the $C_T$ score, or FLS-POG is a reliable metric for memorization. FLS-POG correlates poorly with our estimates of the percentage of memorized samples, while the $C_T$ score and AuthPct detect mode shrinking and image fidelity more than memorization. The reason behind these deficiencies is left to future work. Concurrent work [59] shows that in high dimensions moving the support of a distribution can drastically change precision and recall (measured using $k$-nearest neighbors), and the observed phenomenon here might have a similar cause. Our recommended modification to the $C_T$ score improves on the $C_T$ score and AuthPct, but still does not correlate well with the memorization ratio. Instead of using these metrics, we recommend researchers directly search for and collect memorized images using the calibrated $l_2$-distance as described above, even though it is labour-intensive and requires tuning.

## 6   Conclusions

We carried out the largest and most comprehensive assessment of generative model evaluation metrics to date. We found that currently prevalent metrics such as FID are not strongly predictive of human error rate, and that diffusion models achieve a higher human error rate than their GAN counterparts, yet often are ranked worse according to FID. Our multiple investigations show that this discrepancy is not caused by model diversity, nor by humans assessing rare but real images as fake. Together, these findings imply that the differences between FID and human assessment are unfairly punitive towards diffusion models (in terms of assessing fidelity). We also showed that these deficiencies can be mostly addressed by replacing the Inception encoder by DINOv2 ViT-L/14. We include a table with the $FD_{DINOv2}$ score and many other metrics for all the models we considered in Appendix E, which we hope will be useful as an updated leaderboard of model performance. Finally, FD-based metrics are not designed to detect memorization, and we show that except on CIFAR10, the best performing models in terms of $FD_{DINOv2}$ are not memorizing their training data. In doing so, we also found that while the calibrated $l_2$ distance proposed in [13] is a reliable way to identify memorized samples, other metrics to detect memorization are not ideal. We thus advocate for work proposing new generative models to report the ratio of memorized samples using the calibrated $l_2$ metric alongside metrics computed with DINOv2-ViT-L/14. Because it requires tuning, finding more automated metrics that reliably detect memorization will be a productive avenue for future research.

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

**Broader impact**  While we do not develop or release any generative models in this work, our work can be used to improve the quality of generative models in the future. Such generative models, despite having a number of beneficial uses, have the potential to be used for negative applications such as the generation of deepfakes. We note that all generative models, encoders, and datasets explored in this work are already publicly available assets.

**Limitations**  We take human perceptions of image realism as a ground truth for measuring image fidelity, noting the expansive efforts of the community to generate images that appear photo-realistic to humans, and will be used by humans. But, for certain generative applications, such as the use of generative models as augmentations to improve performance of classifiers [2], humans are not always the end users. In this case the end user is the downstream classifier, and models that achieve the best $FD_{DINOv2}$ score may not directly translate to optimal improvements in downstream classification. Additionally, while the image datasets (and hence generative models) in our study were chosen to span a wide range of natural images, they were all 3-channel RGB format of human-identifiable objects. It is an open question whether our proposed methods of evaluating generative models translate to e.g. medical images or astronomical images, which are also RGB, or more broadly to scientific data of N-dimensions.

**Compute**  Experiments were conducted without a requirement for significant computational resources. Generating datasets was performed either on NVIDIA Titan V GPUs with 12GB of RAM or a small local cluster, and required a total of ∼1100 hours of GPU time, a significant fraction of which was spent generating images from the DiT and LDM models. The analysis was conducted on NVIDIA Titan V GPUs with 12GB of RAM, and computing all metrics for each encoder and generative model required ∼48 hours on a single GPU.

**Assets and code**  As stated in the main text, to facilitate further development of generative models and their evaluation we have prepared a public release of all generated image datasets, human evaluation data, and a modular library to compute 17 common metrics for 9 different encoders at `https://github.com/layer6ai-labs/dgm-eval`. We provide detailed descriptions and links to the public generative models, metrics, and encoders used in Appendices A and B.

# A  Generated datasets

For each generative model across the four datasets studied in this work we generated 100,000 images, unless otherwise stated. For class-conditional models we generated an equal number of samples from each class – 10,000 per class for CIFAR10 and 100 per class for ImageNet. We chose the checkpoints and set parameters (if applicable) to the ones that achieved the lowest FID as stated in the paper, or on the project's github if not otherwise stated. For GANs we set truncation=1.0 throughout, and for class-conditional diffusion models we set the guidance parameter to the reported values. To validate that our set of images for each model was generated correctly we compare the FID calculated with our code (using 50k samples) to the reported value in the paper or in the code release. Any discrepancies were investigated by contacting the authors or raising issues on the project github.

When downsampling images from their native sizes to the ones used in this work we note that significant differences in FID can result from different choices of interpolation methods. As such, we report the interpolation techniques we used to downsample ImageNet and FFHQ to 256 x 256 pixels, and note that these are required for reproducing our results when using the original images (the resized datasets are provided along with our code). Differences in our FIDs and the reported values in various papers could partially be attributed to this effect - for example using Box filtering instead of Lanczos increases the FID of StyleGAN2-ada on FFHQ by 1.0.

Below we list ($i$) the generative models used for each dataset, ($ii$) links to the project pages and checkpoints used, ($iii$) the reported FID in comparison to the value determined from our generated images, and ($iv$) any special notes.

## A.1  CIFAR10

- StudioGAN        models        [55]        `https://github.com/POSTECH-CVLab/`
  `PyTorch-StudioGAN`,  checkpoints  are  in  folder  `https://huggingface.co/`

`Mingguksky/PyTorch-StudioGAN/tree/main/studiogan_official_ckpt/`
`CIFAR10_tailored/`. We used current weights from each folder.

- ACGAN [79] `CIFAR10-ACGAN-Mod-train-2022_03_06_02_24_19`. Reported FID=33.39, ours=35.47.
- BigGAN-Deep [12] `CIFAR10-BigGAN-Deep-train-2022_02_02_21_56_10` Reported FID N/A, ours=3.91.
- LOGAN [111] `CIFAR10-LOGAN-train-2022_03_12_04_15_31`. Reported FID=20.65, ours=17.87.
- ReACGAN [54] `CIFAR10-ReACGAN-train-2022_01_24_23_46_59`. Reported FID=3.87, ours=4.40.
- MHGAN [104] `CIFAR10-MHGAN-train-2022_02_14_18_23_18`. Reported FID=3.95, ours=4.22.
- WGAN-GP [41] `CIFAR10-WGAN-GP-train-2022_01_25_16_34_00`. Reported FID=53.98, ours=26.25.

- LSGM-ODE [107] `https://github.com/NVlabs/LSGM`. We used the FID checkpoint, sampling with ODE framework. Reported FID=2.10, ours=2.12.

- iDDPM-DDIM [78] `https://github.com/openai/improved-diffusion`. Reported FID=2.94, ours=3.27.

- PFGM++ [112] `https://github.com/newbeeer/pfgmpp`. We used the following checkpoint `https://drive.google.com/drive/folders/1IADJcuoUb2wc-Dzg42-F8RjgKVSZE-Jd?usp=share_link`. Reported FID 1.74, ours=1.79.

- RESFLOW [18] `https://github.com/rtqichen/residual-flows`. Reported FID=46.37, ours=48.29.

- NVAE [106]. Reported FID N/A, ours=32.53.

- StyleGAN2-ada [57] `https://github.com/NVlabs/stylegan2-ada-pytorch`. We used the following checkpoint `https://nvlabs-fi-cdn.nvidia.com/stylegan2-ada-pytorch/pretrained/cifar10.pkl`. Reported FID=3.49, ours=2.55.

- StyleGAN-XL [98] `https://github.com/autonomousvision/stylegan-xl`. We used the following checkpoint `https://s3.eu-central-1.amazonaws.com/avg-projects/stylegan_xl/models/cifar10.pkl`. Reported FID=1.85, ours=1.87.

## A.2 ImageNet

- Four models used sets of 50k publicly available images provided at `https://github.com/openai/guided-diffusion/tree/main/evaluations` [25].
    - ADM [25]. Reported FID=10.94, ours=11.84.
    - ADMG [25]. Reported FID=4.59, ours=5.58.
    - ADMG-ADMU [25]. Reported FID=3.94, ours=4.75.
    - BigGAN [12]. Reported FID=6.95, ours=7.94.

- DiT-XL-2 [84] `https://github.com/facebookresearch/DiT`. Reported FID=2.27, ours=2.80.

- DiT-XL-2-guided [84]. This model was equivalent to the above, but used a stronger classifier free guidance term cfg $= 4.0$ instead of cfg $= 1.5$. As described in Section 5.1, this model was included to study the effects of a model that produces very realistic images, but sampled with a lower intra-class diversity. Reported FID=N/A, ours=17.24.

- GigaGAN [56] with 100k images provided privately by authors. Reported FID=3.45, ours=4.16.

- LDM [93] `https://github.com/CompVis/latent-diffusion`. Reported FID=3.60, ours=4.29.

- Mask-GIT [17] `https://github.com/google-research/maskgit`. We used the following checkpoint `https://storage.googleapis.com/maskgit-public/checkpoints/maskgit_imagenet256_checkpoint`. Reported FID=6.06, ours=5.63.

- RQ-Transformer [65] `https://github.com/kakaobrain/rq-vae-transformer`. We used the 1.4B model from the following checkpoint `https://arena.kakaocdn.net/brainrepo/models/RQVAE/6714b47bb9382076923590eff08b1ee5/imagenet_1.4B_rqvae_50e.tar.gz`. Reported FID=8.71, ours=9.71.

- StyleGAN-XL [98] `https://github.com/autonomousvision/stylegan-xl`. We used the following checkpoint `https://s3.eu-central-1.amazonaws.com/avg-projects/stylegan_xl/models/imagenet256.pkl`. Reported FID=2.26, ours=2.91.

### A.2.1 A note on the curation of ImageNet

The treatment of the raw ImageNet dataset [23] is often not explicitly described in the literature – particularly for building generative models on ImageNet – and we have found some inconsistencies on how FID is reported. Therefore we describe our exact approach to calculating FID on ImageNet, and provide our parsing scripts with the goal of standardizing the process across papers. The full dataset was obtained from `https://www.kaggle.com/competitions/imagenet-object-localization-challenge/data`, as the data is no longer available from the ImageNet website (`https://www.image-net.org/index.php`).

Raw ImageNet is a dataset of 1,000 classes, each with roughly 1,300 images. The images themselves are generally non-square, with image height and width both always at least 256. To get from the raw rectangular ImageNet images to the $256 \times 256$ resolution of ImageNet256 we perform the following two operations:

1. Center crop along the long edge. This results in a square image of side length $N = \min\{H, W\}$, given individual image height $H$ and width $W$.
2. Downsample from $N \times N$ to $256 \times 256$ using bicubic interpolation, as suggested by [82]. We also experimented with Lanczos interpolation with only minimal change to the results.

This procedure results in roughly $1.3 \times 10^6$ images of size $256 \times 256$. We construct a reference batch of 100,000 training images matching our generated sets by sampling 100 images from each class without replacement. We then calculate FD using 50,000 images drawn without replacement each from the training set and the generated set. These choices result in a slight increase in FID values for most of the ImageNet models reported above, as most determine the FID of 50k generated samples and all 1.3MM training samples, but do not affect any model rankings.

### A.3 FFHQ

Following the examples set in the StyleGAN repositories, we downsampled the original 1024 x 1024 FFHQ dataset to $256 \times 256$ using Lanczos interpolation.

- Efficient-vdVAE [43] `https://github.com/Rayhane-mamah/Efficient-VDVAE`. We used the 8-bit version from the following checkpoint `https://storage.googleapis.com/dessa-public-files/efficient_vdvae/Pytorch/ffhq256_8bits_baseline_checkpoints.zip`. Reported FID N/A, ours=34.88.

- Insgen [113] `https://github.com/genforce/insgen`. We used the following checkpoint `https://drive.google.com/file/d/1OtSwESM_8S60EtiSddR16-gzo6QW7YBM/view?usp=sharing`. Reported FID=3.31, ours=3.46.

- LDM [93] `https://github.com/CompVis/latent-diffusion`. Reported FID=4.98, ours=8.11. To our knowledge the settings we used to generate the images were consistent with the codebase and with those used in the paper. We contacted the authors about this via email (as the GitHub page is full of unresolved issues) but received no response.

- Projected-GAN [97] `https://github.com/autonomousvision/projected-gan`. Reported FID=3.39, ours=3.46.

- StyleGAN2-ada [57] `https://github.com/NVlabs/stylegan2-ada-pytorch`. We used the following checkpoint `https://nvlabs-fi-cdn.nvidia.com/stylegan2-ada/pretrained/paper-fig7c-training-set-sweeps/ffhq70k-paper256-ada.pkl`. Reported FID=4.30, ours=5.30. *The discrepancy*

*is purely due to different choices for downsampling*: We used Lanczos filtering to downsample the original 1024x1024 pixel FFHQ images to 256x256, while their work used box filtering. Box filtering exactly reproduces the results reported in the StyleGAN2-ada paper, and we thank the authors for their work to help us solve this discrepancy. Please see the following github issue for more details `https://github.com/NVlabs/stylegan2-ada-pytorch/issues/283`.

- StyleGAN-XL [98] `https://github.com/autonomousvision/stylegan-xl`. We used the following checkpoint `https://s3.eu-central-1.amazonaws.com/avg-projects/stylegan_xl/models/ffhq256.pkl`. Reported FID=2.19, ours=2.26.

- StyleNAT [109] `https://github.com/SHI-Labs/StyleNAT`. We used the following checkpoint `https://shi-labs.com/projects/stylenat/checkpoints/FFHQ256_940k_flip.pt`, Reported FID=2.046, ours=2.11.

- StyleSwin [116] `https://github.com/microsoft/StyleSwin`. We used the following checkpoint `https://drive.google.com/file/d/1OjYZ1zEWGNdivORFKv7KhXRmYko72LjO/view?usp=sharing`. Reported FID=2.81, ours=2.89.

- Unleashing-Transformers [9]: `https://github.com/samb-t/unleashing-transformers`. We used the following checkpoint `https://github.com/NVlabs/ffhq-dataset`. Reported FID=7.12, ours=9.02. We note that other discrepancies with the reported FID on FFHQ have previously been raised via the issues function of GitHub.

## A.4 LSUN-Bedroom

- Four models used sets of 50k publicly available images provided at `https://github.com/openai/guided-diffusion/tree/main/evaluations` [25].

  - ADMNet-dropout [25]. Reported FID=1.90, ours=2.20.
  - DDPM [49]. Reported FID=4.89, ours=5.18.
  - iDDPM [78]. Reported FID=4.24, ours=4.54.
  - StyleGAN [58]. Reported FID=2.35, ours=2.65.

- Consistency [70]. We used the consistency training model provided at `https://github.com/openai/consistency_models`. Reported FID=7.85, ours=8.27.

- Diffusion-Projected GAN [110]. We used the pretrained projected GAN model provided at `https://github.com/Zhendong-Wang/Diffusion-GAN`. Reported FID=1.43, ours=1.79.

- Projected GAN [97]. We used the pretrained projected GAN model provided at `https://github.com/autonomousvision/projected-gan`. Reported FID=1.52, ours=2.23. The settings we used to generate the images were consistent with the codebase defaults, which were, as far as we could tell, consistent with those used in the paper. We contacted the authors both via email and the issues function of GitHub but received no response.

- Unleashing Transformers [9]. We used the pretrained model provided by the authors at `https://github.com/samb-t/unleashing-transformers`. Reported FID=3.64, ours=3.58.

# B  Metrics and encoders

To assess all aspects of generative models we include a number of diagnostic metrics that are designed to directly quantify the fidelity, diversity, rarity, or memorization of our 100k image samples. Such metrics are valuable for diagnostic purposes but difficult to use to directly rank generative models. Ranking requires metrics which group these concepts into a single value without a clear tradeoff. In total we assess each generative model across 17 metrics, both ranking and diagnostic. These metrics are computed in a number of supervised and self-supervised representation spaces. In total we include 9 encoders. Throughout this section, we denote generated samples as $\{x_i^g\}_{i=1}^n$, and real samples as $\{x_j^r\}_{j=1}^m$.

The following appendix ($i$) details each of the metrics, ($ii$) presents controlled experiments and additional checks for diversity and memorization metrics, and ($iii$) describes the encoders and motivates the choice of ViT model size used.

## B.1 Metrics for ranking generative models

**FD**  The Fréchet distance (FD) uses ($\mu_r, \Sigma_r$) and ($\mu_g, \Sigma_g$), the sample mean and covariance of the real and generated representations, respectively, and is determined through:

$$\text{FD}(\mu_r, \Sigma_r, \mu_g, \Sigma_g) = \|\mu_r - \mu_g\|_2^2 + \text{Tr}\left(\Sigma_r + \Sigma_g - 2(\Sigma_r\Sigma_g)^{\frac{1}{2}}\right). \tag{1}$$

The FD corresponds to the Wasserstein-2 distance between Gaussians with the corresponding means and covariances, and is thus a valid metric between the first two moments of real and generated distributions. If these two distributions happen to be characterized by their first two moments (e.g. they are both Gaussian), then the FD becomes a metric between distributions, not just their first two moments. We use 50,000 images for both the real and generated images. To move away from the standard Inception-V3 network which [103] used as a feature extractor we drop the "I" from this metric and the following, where appropriate.

**FD∞**  $\text{FD}_\infty$ [22] aims to remove the inherent bias of the FD due to the finite number of samples. It is determined by evaluating the FD at 15 regular intervals over the number of samples $N$, from $N = 5,000$ to $N = 50,000$, fitting a linear trend to the 15 data points, and using the trend to infer a FD value at $N = \infty$, $\text{FD}_\infty$.

**sFID**  The spatial FID (sFID) [76] is the FID computed using a representation from the intermediate mixed 6/conv layer of the Inception-V3 network [103] trained for ImageNet1k [23], rather than the standard (pool3, 2048 dimensional) layer. The sFID relies on the Inception-V3 network, and thus we do not report it for other architectures.

**KD**  The Kernel distance (KD) [8] aims to replace the FD with something that is a proper distance between distributions regardless of whether the distributions are characterized by their first two moments. In order to this, an unbiased estimate of the maximum mean discrepancy [39] is used:

$$\text{KD}\left(\{x_i^g\}_{i=1}^n, \{x_j^r\}_{j=1}^m\right) = \frac{1}{n(n-1)}\sum_{i \neq i'}^n k(x_i^g, x_{i'}^g) + \frac{1}{m(m-1)}\sum_{j \neq j'}^m k(x_j^r, x_{j'}^r) - \frac{2}{nm}\sum_{i=1}^n\sum_{j=1}^m k(x_i^g, x_j^r), \tag{2}$$

where $k$ is a positive definite kernel that is chosen as a hyperparameter. We use the standard 3rd degree polynomial kernel.

**IS**  The Inception score (IS) [96] is maximized when the entropy of the distribution of labels predicted by the Inception-V3 model is minimized for every input (i.e. generated image) and when the predictions are evenly distributed across all 1000 possible labels. It is given by:

$$\text{IS}\left(\{x_i^g\}_{i=1}^n\right) = e^{\frac{1}{n}\sum_{i=1}^n \mathbb{KL}(p(y|x_i^g)\|p(y))}, \tag{3}$$

where $p(y|x)$ denotes the output of the Inception-V3 network when $x$ is the input, and $p(y)$ the observed frequencies of labels in the training data. We use a refactored version from `https://github.com/sbarratt/inception-score-pytorch/blob/master/inception_score.py` to compute this metric. The IS relies on the Inception-V3 network, and thus we do not report it for other architectures.

**FLS**  The feature likelihood score (FLS) [53] is a recently proposed density estimation method that requires training, generated, and test samples. The goal of FLS is to be as close to likelihood evaluation even when there is no likelihood available. In order to do this, the FLS fits a kernel density estimate (KDE) to the generated samples (after having been transformed by the encoder). The bandwidths of the KDE are chosen so as to maximize the log-likelihood of training data, and the FLS is then given by (an affine transformation of) the test log-likelihood obtained by the KDE. We use the default implementation from `https://github.com/marcojira/fls`.

## B.2 Metrics for fidelity, diversity, and rarity

**Precision, recall, density, coverage** As proxies for sample fidelity we use precision [95, 64] and density [75]. Both precision and density are based upon nearest neighbours computed in a representation space. Precision counts the binary decision of whether a generated sample is contained in any neighbourhood sphere of training points, while density counts how many real-sample neighbourhood spheres contain the sample. To quantify sample diversity we use recall [95, 64] and coverage [75], which similarly to their counterparts precision and density, are based upon nearest neighbours computed in a representation space. Formally, precision is given by:

$$\text{precision}\left(\{x_i^g\}_{i=1}^n, \{x_j^r\}_{j=1}^m\right) = \frac{1}{n}\sum_{i=1}^n \mathbb{1}\left(x_i^g \in S(\{x_j^r\}_{j=1}^m)\right), \tag{4}$$

where $\mathbb{1}(\cdot)$ denotes the indicator function, $S(\{x_j^r\}_{j=1}^m) = \cup_{j=1}^m B(x_j^r, \text{NND}_k(x_j^r))$, where $B(x, r)$ denotes a Euclidean ball centered at $x$ with radius $r$, and $\text{NND}_k(x_j^r)$ is the distance between $x_j^r$ and its $k^{th}$ nearest neighbour in $\{x_j^r\}_{j=1}^m$, excluding itself. Similarly, recall is given by:

$$\text{recall}\left(\{x_i^g\}_{i=1}^n, \{x_j^r\}_{j=1}^m\right) = \frac{1}{m}\sum_{j=1}^m \mathbb{1}\left(x_j^r \in S(\{x_i^g\}_{i=1}^n)\right), \tag{5}$$

density is given by:

$$\text{density}\left(\{x_i^g\}_{i=1}^n, \{x_j^r\}_{j=1}^m\right) = \frac{1}{kn}\sum_{i=1}^n\sum_{j=1}^m \mathbb{1}\left(x_i^g \in B(x_j^r, \text{NND}_k(x_j^r))\right), \tag{6}$$

and coverage is given by:

$$\text{coverage}\left(\{x_i^g\}_{i=1}^n, \{x_j^r\}_{j=1}^m\right) = \frac{1}{m}\sum_{j=1}^m \max_{i=1,\ldots,n} \mathbb{1}\left(x_i^g \in B(x_j^r, \text{NND}_k(x_j^r))\right). \tag{7}$$

Precision, recall, density, and coverage are determined with the code provided at `https://github.com/clovaai/generative-evaluation-prdc` associated with the work [75]. We use 5 nearest neighbors ($k = 5$) and 10,000 samples throughout, as standard.

**Rarity score** To determine the "rarity" of an individual image $x$ we use the rarity score [42]. Similar to precision, recall, density, and coverage, the rarity score uses $k$-nearest neighbours:

$$\text{rarity}\left(x, \{x_j^r\}_{j=1}^m\right) = \min_{j \in J(x, \{x_j^r\}_{j=1}^m)} \text{NND}_k\left(x_j^r\right), \tag{8}$$

where $J(x, \{x_j^r\}_{j=1}^m) = \{j = 1, \ldots, m \mid x \in B(x_j^r, \text{NND}_k(x_j^r))\}$.

**Vendi score** To study inter- vs intra-class diversity for class-conditional image generation we utilise the Vendi score [33]. This score does not require a reference dataset. Formally, the Vendi score is given by:

$$\text{VS}(\{x_i^g\}_{i=1}^n) = e^{-\sum_{i=1}^n \lambda_i \log \lambda_i}, \tag{9}$$

where $\lambda_i$ is the $i^{th}$ eigenvalue of the $n \times n$ matrix $K/n$, where $K_{ii'} = k(x_i^g, x_{i'}^g)$ for a positive semi-definite kernel $k$ (chosen as a hyperparameter) with $k(x, x) = 1$. The convention that $0\log 0 = 0$ is used to compute the Vendi score. The Vendi score over the whole dataset reports the overall sample diversity, and when conditioned on class it quantifies the intra-class diversity of samples. We separate the two in order to study mode concentration of class conditional models - we find some models have high inter-class diversity, which is encouraged through the class conditioning, while lacking intra-class diversity. While recall and coverage can also be determined per class, the small number of generated samples available for each class (e.g. 100 for each ImageNet class) results in difficulty constructing robust nearest-neighbour-based estimates. As per the original paper, we use a linear kernel to compute the score. We also tried using the same polynomial kernel that we used for computing KD, but found no clear benefit from doing so.

## B.3 Metrics for memorization

We also investigate a form of overfitting that we term *memorization*, in which models memorize individual images from their training data and emit them at generation time.

**AuthPct** AuthPct [1] deems each generated sample either as authentic or inauthentic and returns the fraction of authentic samples. Inauthentic samples are those for which the distance to the nearest point in the training set is less than the distance between that training sample and its nearest neighbour in the training set.

**$C_T$** The $C_T$ score [70] is a global three sample non-parametric test explicitly designed to detect memorization. It uses the training set, the test set, and the generated samples, and summarizes how often training samples are closer to generated samples than they are to samples in the test set through a Mann-Whitney hypothesis test [69]. The data is split into a number of cells using $k$-means clustering, the score is determined in each cell, then the per-cell score is averaged into the final one. We use the default implementation from `https://github.com/casey-meehan/data-copying`, namely $k = 3$ and preprocessing the representation space using a 64 component PCA.

**FLS-POG** We denote the percentage of overfit Gaussians as determined in FLS as FLS-POG [53]. For each generated sample we compute the difference between the train and test log-likelihoods, and report percentage of generated samples that had a higher log-likelohood under the training set than the test set. We use the default implementation from `https://github.com/marcojira/fls/`.

**Memorization ratio with calibrated $l_2$ distance** The calibrated $l_2$ distance [101] of a generated sample $x_i^g$ is given by:

$$l(x_i^g, \{x_j^r\}_{j=1}^m) = \frac{\|x_i^g - \text{NN}_1(x_i^g)\|_2}{\frac{1}{k}\sum_{k'=1}^k \|\text{NN}_1(x_i^g) - \text{NN}_{k'}(\text{NN}_1(x_i^g))\|_2}, \tag{10}$$

where $\text{NN}_k(x)$ denotes the $k^{th}$ nearest neighbour (in Euclidean distance) to $x$ in the training data without $x$, i.e. $\{x_j^r\}_{j=1}^m \setminus \{x\}$. Note that this metric is computed directly on pixel space, with no encoder being used. Low values of $l(x_i^g, \{x_j^r\}_{j=1}^m)$ indicate that $x_i^g$ memorized a training sample. A threshold $\tau$ is set and the generated sample $x_i^g$ is then considered as "memorized" if its calibrated $l_2$ distance is smaller than $\tau$. The memorization ratio is given by the percentage of memorized samples:

$$\text{memorization\_ratio}\left(\{x_i^g\}_{i=1}^n, \{x_j^r\}_{j=1}^m\right) = \frac{1}{n}\sum_{i=1}^n \mathbb{1}\left(l(x_i^g, \{x_j^r\}_{j=1}^m) < \tau\right). \tag{11}$$

While the memorization ratio can provide a good assessment of how much a model memorized its training samples (Section 5.2), no default configuration of the $k$ and $\tau$ hyperparameters achieves this across all models and datasets: these values need to be hand-tuned in order to meaningfully flag memorization. Thus, despite our recommendation to use this metric to detect memorization, we do not believe that it should be used as a basis to compare models.

## B.4 Encoders

With the aim of finding a more generalized perceptual representation space across the span of natural images we employed a number of alternative encoders. Here we outline the specific implementations used, and provide evidence for the ViT-L/14 architecture choice. For self-supervised methods we include short descriptions on the family of training employed, grouping into categories based on those of [4].

**ConvNeXt** As a more modern supervised benchmark we use a ConvNeXt-large architecture trained on ImageNet22k [67]. We use the `timm` implementation provided here `https://github.com/huggingface/pytorch-image-models/blob/main/timm/models/convnext.py` and select the `convnext_large_in22k` model.

**SwAV** SwAV [14] featured heavily in a previous work advocating for the use of self-supervised feature extractors for generative model evaluation [72], as at the time it was a state-of-the-art self-supervised image representation model. SwAV is trained on ImageNet1k using a canonical correlation analysis method, and learns by clustering data while enforcing consistency between cluster assignments produced for different views of an image. We use the ResNet50 architecture and weights from [72], with code provided

at `https://github.com/stanis-morozov/self-supervised-gan-eval/blob/main/src/self_supervised_gan_eval/resnet50.py`.

**SimCLRv2**    SimCLRv2 [19] is contrastive method which learns visual representations by encouraging the similarity between two often heavily augmented views of an image, while discouraging the similarity between distinct training images. SimCLRv2 is a CNN trained on ImageNet1k. We used the PyTorch implementation available at `https://github.com/Separius/SimCLRv2-Pytorch`.

**CLIP**    CLIP is a multi-modal language-image model [87], which trains an image encoder and text encoder to predict which images were paired with which texts. We use CLIP ViT [87]. CLIP-B/32 was studied for generative evaluation in a controlled experiment in [7] (while they do not state the architecture used in the paper we found it at `https://github.com/eyalbetzalel/fcd/blob/main/fcd.py`). In the next section we show that this choice is sub-optimal, and that a better choice is the OpenCLIP ViT-L/14 implementation [51] trained on DataComp-1B [35]. Thus, we use OpenCLIP ViT-L/14 trained on DataComp-1B.

**DINOv2**    DINOv2 [80] is from the self-distillation/discriminative family, which learns representations by passing two different views of an image to two encoders and mapping one to the other by means of a predictor. DINOv2 is a ViT-L/14 architecture trained on a large custom dataset of 142M images built by combining common datasets for classification, segmentation, depth estimation, and retrieval tasks, plus additional images from the internet. We justify the architecture choice alongside CLIP in the following section.

**MAE**    Masked autoencoder (MAE) [44] is from the masked image modelling family, and learns to directly reconstruct masked image patches. We use the ViT-L PyTorch implementation from `https://github.com/facebookresearch/mae`, which was trained on ImageNet1k.

**data2vec**    data2vec [3] is from the masked image modelling family, and learns by predicting latent representations of the full input data based on a masked view of the input. We use the Hugging Face implementation found here `https://huggingface.co/docs/transformers/model_doc/data2vec`. We use ViT-L, which was trained on ImageNet1k.

**DreamSim**    The DreamSim method [34] uses a dataset of human similarity judgments over image pairs in order to fine-tune encoders to better align with human perception of image similarity. DreamSim was publicly released after the initial submission of this paper. We also highlight that the focus of our work is different than DreamSim: our human trials focus on image quality rather than similarity, and our subsequent analysis uses these human judgements to evaluate current metrics for generative models, while DreamSim creates a dataset of human-perceived image-to-image similarities and designs an encoder that reflects these human judgements. In other words, DreamSim focuses on obtaining an encoder which maps images that humans assess as similar to nearby points in latent space, whereas we focus on finding an encoder where distances between probability distributions on its latent space, such as FD, correlate with human judgment of fidelity. Our experiments use the full DreamSim ensemble, the concatenation of the representations of three different self-supervised models (DINO [15], CLIP, and OpenCLIP) that have each been fine-tuned on the DreamSim dataset.

We note that while separating our encoders into such self-supervised families allows for a broad understanding of the different methodologies employed during training and hints at potential characteristics at their learned representation spaces, models can in practice employ a mix of approaches. In particular, DINOv2 also employs a masked modeling objective, however it was not masking images but rather the latent-space with a teacher network used to provide targets [80].

### B.4.1    Determining ViT model size for generative evaluation

To determine which model size provides a good perceptual space for generative evaluation with ViT-based models we calculated FD and precision of the 11 ImageNet generative models under varying model sizes of CLIP and DINOv2 encoders. For CLIP we also studied models trained using different sets of training data (OpenAI LAION-400M, LIAON-2B, and DataComp-1B), while DINOv2 encoders all used an identical training set. CLIP models not denoted with OpenAI were from OpenCLIP. While larger models achieve better accuracy under a linear evaluation protocol

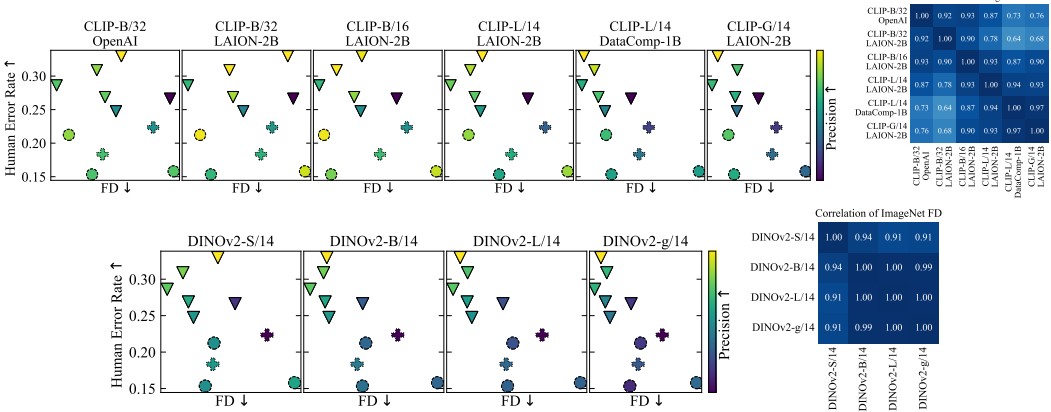

Figure 8: ImageNet results as a function of ViT model size for CLIP (top) and DINOv2 (bottom).

on ImageNet1k, which may be evidence towards using them for generative evaluation, we argue that the computational cost of such a large model is restrictive to the development of generative models. Often, metrics need to be tracked during model development for numerous training setups, hyperparameter settings, and epochs, which motivates networks with a lower computational cost. For reference, on a workstation with a single NVIDIA Titan V GPU, determining the representations of 50k samples takes ∼4 minutes with ViT-B/14, ∼15 minutes with ViT-L/14, and ∼85 minutes with ViT-G/14. Thus, smaller encoders that exhibit representation spaces matching the utility of larger encoders should be preferred.

Figure 8 shows the results of the model size sweep. Models are listed by ascending size. For CLIP we find that CLIP-B/32-OpenAI, CLIP-B/32-LAION2B, and CLIP-B/16-LAION2B show a much smaller correlation between FD and human error rate than the ViT-L/14 and ViT-G/14 models. We see that the model best aligned with the expensive ViT-G/14 - the highest performing CLIP model under linear evaluation protocols - is CLIP-L/14-DataComp-1B. We quantify this agreement with the FD correlation matrix shown on the right (the precision matrix, while not shown displays the same trends), and find a very high correlation between CLIP-L/14-DataComp-1B and CLIP-G/14-LAION2B.

For DINOv2 we see lower alignment of FD and human error rate when using the ViT-S/14 model in comparison to the larger three, and also find good agreement between the ViT-B/14, ViT-L/14, and ViT-G/14 models. We again quantify this agreement with the FD correlation matrix shown on the right (the precision matrix, while not shown displays the same trends), and find that the ViT-L/14 shows the highest correlation with ViT-G/14, although ViT-B/14 is highly correlated as well.

As evidenced by both the CLIP and DINOv2 results, ViT-L/14 provides a perceptual space nearing the ImageNet qualities of the largest encoders at a much lower computational cost. Thus we believe that ViT-L/14 provides a good tradeoff between representation quality and computation cost, and use it as the basis for all ViT-based encoders in this work. While we propose DINOv2-L/14 as the encoder that best improves upon Inception-V3, we suggest that the strong correlation of DINOv2-B/14, yet a factor of ∼4 lower computational cost, will provide useful during generative model development - metrics can be tracked using this model size during training, and final leaderboard values can be reported after training has concluded using DINOv2-L/14. Our codebase allows to easily do this.

## C  Human subject experiments

### C.1  Test description

As noted in Section 3, the goal of our human subject experiments was to evaluate the fidelity of generated images in a scientifically grounded way across many models and datasets. The design of our tests was therefore specifically tailored to address the realism of generated images, and not other aspects of the modeled distributions, such as diversity. We further emphasize that our focus is on the realism of images that are generated for human use, not other uses (for example generating synthetic data to improve classifier performance [46], to debias datasets [108], or for dataset distillation [16]).

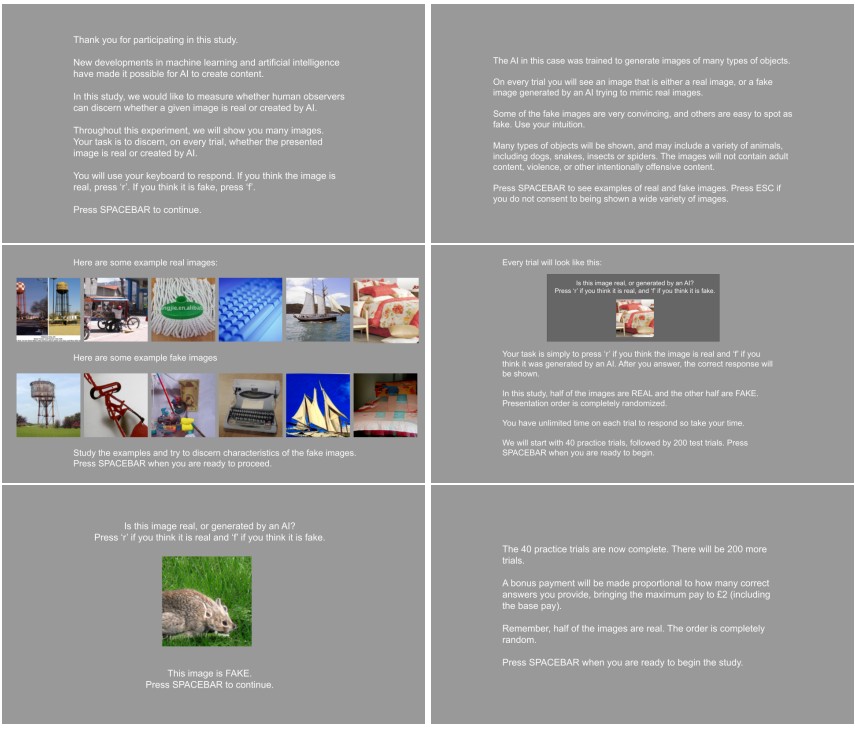

Figure 9: Screens displayed to participants during our StyleGAN-XL evaluation for ImageNet. Other experiments followed the same template. The bottom left shows the format of the 40 practice and 200 test trials, with the correct answer text shown only after a participant entered their response.

We found that the experiments from [120] for the HYPE$_\infty$ metric supported these goals, and thus we used them as a template. We first describe our method in detail, compare the differences from [120], and then present our full results.

In each test, generated images from a single model are directly compared to images from the training dataset - an absolute comparison [81]. Previous human experiments have used comparative ratings rather than absolute, for example by directly comparing pairs of models [99, 105, 117, 90, 94], or grouping together several models with a baseline and comparing all at once [24, 99, 96, 50, 27, 97]. The absolute approach has several advantages:

- Models are tested against a single, consistent baseline.
- The number of experiments scales linearly with the number of models to be tested. Pairwise testing requires a quadratic number of experiments.
- Experiments can be extended with new models in a consistent manner - additional experiments do not depend on previously used models.
- Comparisons of new results against previously-gathered results are analytically simple using between-subjects comparisons. The comparative paradigm would require mixing between-subjects and within-subjects designs.
- The performance of any given model does not depend on which other models were tested in the same experiment.
- Scores can indicate whether models generate images indistinguishable from the baseline, or if there is room for improvement (see Figure 2 where the optimal error rate of 0.5 is approached only for CIFAR10).
- Experiment results will not be outdated when researchers make progress with new models.

In total, 41 distinct experiments were conducted, each consisting of an instruction and training phase, followed by a testing phase. Research on how to collect high quality data from crowd sourcing platforms [71] shows that providing training on the task to be completed is the best way to improve

data quality, while providing financial incentives can also work, but pre-screening individuals by their on-platform reputation is ineffective. Hence, in the first phase we provide participants with background information, a request for consent, and clear instructions on their task, including the fact that exactly half of the presented images would be real. We then conducted 40 practice trials which were not used in the analysis of our results. Each trial was a two-alternative forced choice task, in which the participant was shown a single image which was either a "real" image from the training dataset, or a "fake" image generated by the model being tested. The participant was simply asked to judge if the image was real or fake with no alternatives, and they were provided feedback on the correct answer directly after they responded. In the testing phase, subjects are shown an additional 200 trials of the exact same format. Participants were unconstrained in their time to view and judge each image. There was a 200 ms inter-trial interval before presentation of the subsequent image, and a delay of 100 ms after image presentation before participants could enter a response to prevent accidentally advancing through trials. For reproducibility, in Figure 9 we show screenshots from an example test, the StyleGAN-XL model on ImageNet, as they were displayed to the participants. The experiments were created using PsychoPy [85] and hosted on Pavlovia [83]. The experiments were only administered on desktop or laptop computers, not on mobile devices or tablets, and image size was scaled to the user's display.

The choice to reveal correct answers to participants after they guess is not completely standard across the literature. Some authors have provided feedback [96, 120, 50], while others have avoided it [117, 66], and often no justification is provided. Ultimately, we want participants to make the most accurate decisions they are capable of, which is why we use a long training phase, and provide feedback throughout. The main argument against providing feedback during testing is that participants may "learn" and adjust their responses, which may in effect change the distribution of experimental data throughout the test. We do not find this to be concerning for the following reasons. First, our long training period is meant for participants to learn and improve their performance on the real task; most learning will occur early during the training phase, less so during the testing phase. Second, even without feedback, participants can still learn and change their approach to the task as they see more examples. Third, with or without feedback all participants have the same opportunity to learn, and so our experimental datapoints, the error rates of individual participants on 200 trials, are drawn from the same hypothetical distribution. In preliminary experiments we observed no significant difference between the error rates on the first and second halves of the test trials, showing that learning during the test phase does not create a large effect.

**Ethical considerations**   Our institution does not have an internal review board (IRB) approval requirement for running crowd-sourced experiments, and thus has no formal approval process for experiments such as ours. In place of an IRB approval we followed an informal process internally prior to running the experiments, as recommended in the NeurIPS 2023 guidelines. The process involved: confirming what internal review processes existed and were required of us at our institution, ensuring that none of the images that would be shown to participants had any explicit or offensive content, informing participants that they may be shown images that could trigger common phobias, taking the test ourselves, and paying the participants above minimum wage.

**Participant recruitment and compensation**   Human subject experiments for generative model evaluation are quite common, but often are ad-hoc and cannot be considered as randomized, controlled experiments. Sometimes authors themselves are the participants [48, 68], or volunteers known to the authors are used [24, 117, 66, 97], raising questions of bias. Luckily, there are several crowd-sourcing platforms to recruit willing and unbiased participants from the general population. For our experiment, participants were recruited from Prolific [86] which has been shown to provide the highest quality data in scientific studies about behavioural research platforms [31, 29]. While we did not filter eligible participants by their on-platform reputation, we did require participants to be fluent in English, the language used throughout the experiment, and to have completed college, or a university degree at least at the Bachelors level (we used this filter to maximize the probability of fluency in English, and thus of instructional obedience during the trials). No other filters were used, and statistics on the demographics about consenting participants are shown in Table 2. Comparisons between models and datasets are therefore between-subjects. For each experiment, which took a median of 15 minutes to complete, participants were paid at a median rate of £7.60/hr, including a base rate for completing the study which complied with the payment standards of the Prolific platform, and an incentive of £0.001 for each correct answer.

Table 2: Demographics and performance for the 897 of 1036 participants who voluntarily consented to release their demographics through Prolific. The normalized error rate accounts for the varying difficulty of tasks over different combinations of (dataset, model). A negative value means better performance on the task, with a value of 1 meaning 1 standard deviation above the mean within their task cohort of ~25 participants. As intuitively expected, there is no significant performance difference based on demographic group.

| | Group | # Participants | Normalized error rate |
|---|---|---|---|
| **Age group** | 19-29 | 574 | -0.01±0.93 |
| | 30-39 | 200 | -0.18±0.95 |
| | 40-49 | 77 | -0.01±1.07 |
| | 50+ | 46 | 0.52±1.21 |
| **Sex** | Female | 389 | 0.01±0.95 |
| | Male | 507 | -0.05±0.99 |
| | Prefer not to say | 1 | - |
| **Highest education level** | Doctorate degree | 19 | -0.32±0.70 |
| | Graduate degree | 294 | 0.10±0.98 |
| | Technical/community college | 121 | 0.02±1.02 |
| | Undergraduate degree | 463 | -0.09±0.96 |
| **Continent of birth** | Africa | 127 | 0.21±0.94 |
| | Asia | 40 | 0.19±1.07 |
| | Europe | 557 | -0.02±0.96 |
| | North America | 129 | -0.19±0.99 |
| | Oceania | 13 | -0.46±0.99 |
| | South America | 29 | -0.36±0.87 |
| | Unknown | 2 | 0.42±1.61 |
| **Ethnicity** | Asian | 44 | 0.34±1.12 |
| | Black | 103 | 0.35±0.89 |
| | Mixed | 86 | -0.10±0.96 |
| | Other | 44 | -0.26±0.85 |
| | White | 620 | -0.08±0.96 |

The experiments for each of the 41 models were completed by at least 25 unique recruits (see Tables 4 through 7), totalling 1,036 different people, and providing over 207k individual test responses (along with more than 41k practice responses not included in our statistics), making this the largest human study evaluating generated image quality that we know of.[2] A small number of experiments were completed by more than 25 participants due to unstable connections to the Prolific and Pavlovia servers causing logging issues.

**Image selection** Within each experiment, the images shown to a given participant were selected and randomized as follows. First, 2,000 images from the training dataset were randomly selected without replacement with classes in proportion to the dataset if applicable. The same set of training images were used for each model corresponding to that dataset. In a similar fashion, 2,000 model samples were selected (see Section 2 and Appendix A), in proportion to the dataset's class distribution if the model was class-conditional. For each participant, 100 real and 100 generated images were selected at random from these pools, again respecting class distributions where applicable. The number of times each generated image was viewed by a human therefore follows a binomial distribution with $n \geq 25$ and $p = 0.05$. The 200 selected images were presented to the participant in a randomized order. The 40 practice images were chosen in a similar fashion, but had no overlap with the 4,000 test images, and were reused for all participants. Appendix D shows image samples for each model.

---

[2]The next largest similar experiment that we know of collected about 138k individual responses [120]. Human evaluations have also been conducted on text-to-image generation quality, where [90] collected about 40k individual responses, [94] collected about 25k, and more recently [81] collected about 33k responses.

## C.2  Comparison to HYPE$_\infty$

Our experimental design was inspired by the HYPE$_\infty$ score [120], but we made several changes to improve the quality of collected data. We summarize the differences in Table 3. We doubled the number of images shown to each participant for more accurate estimation of each participant's error rate, while slightly lowering the number of participants per model. We used Prolific rather than MTurk because research shows that Prolific can provide better data quality due to their participant screening practices [31, 29]. Prolific also requires fair base pay rates based on time, so compensating participants solely on their number of correct responses is disallowed. The pre-screening test from [120] used a mix of data from all models they tested, which does not align with our preference for absolute comparisons and extensible experiments given in Appendix C.1, so we did not use it.

## C.3  Results

Each of the 41 models were evaluated based on *human error rate* [120], the fraction of the 200 test images which were incorrectly classified. This is a simple metric with the intuition that models with better fidelity produce images that are more difficult for humans to distinguish from real images. Detailed results by dataset that were summarized in Figure 2 and Table 1 from the main text are given in Tables 4 through 7. Along with the overall error rate, was also provide the error rates on the real and fake images separately. As in [120], we observe that the error rate on real images increases for models with higher fidelity, even though real images are drawn from the same pool of 2000 examples across models. We also provide the mean rate that participants guessed "real", noting that they were instructed that exactly half of presented images would be real. Finally, we give the mean time spent on the 200 test examples.

Table 3: Comparison between design of HYPE$_\infty$ experiments [120] and ours.

| | **Ours** | **HYPE$_\infty$** |
|---|---|---|
| Correct answer revealed to participant after their response | Yes | Not specified |
| Number of real/fake images in selection pools | 2000 | 5000 |
| Number of images shown to each participant | 200 | 100 |
| Number of participants for each experiment | $\geq 25$ | 30 |
| Number of experiments conducted | 41 | 13 |
| Number of responses collected | 207k | 39k |
| Recruitment platform | Prolific | MTurk |
| Payment | Base pay £1.80 + £0.001 per correct answer | $1 for passing qualifying test + $0.02 per correct answer on main test |
| Qualification | Fluent in English, minimum education requirement | Achieve $\geq 65\%$ accuracy on test using a mix of data from all models tested |

Table 4: Detailed human experiment results on CIFAR10

| Model | # Tests | Error Rate ↑ | R Error Rate ↑ | F Error Rate ↑ | R Answer Rate | Time (s) |
|---|---|---|---|---|---|---|
| ACGAN | 25 | 0.148 ± 0.013 | 0.181 ± 0.021 | 0.116 ± 0.016 | 0.467 | 217 |
| BigGAN | 25 | 0.387 ± 0.014 | 0.381 ± 0.021 | 0.393 ± 0.022 | 0.506 | 284 |
| iDDPM-DDIM | 26 | 0.400 ± 0.013 | 0.390 ± 0.019 | 0.410 ± 0.022 | 0.510 | 268 |
| LOGAN | 25 | 0.206 ± 0.020 | 0.204 ± 0.021 | 0.208 ± 0.026 | 0.502 | 220 |
| LSGM-ODE | 27 | 0.437 ± 0.010 | 0.417 ± 0.021 | 0.456 ± 0.018 | 0.519 | 232 |
| MHGAN | 25 | 0.336 ± 0.015 | 0.331 ± 0.016 | 0.341 ± 0.021 | 0.505 | 267 |
| NVAE | 25 | 0.131 ± 0.018 | 0.128 ± 0.015 | 0.134 ± 0.025 | 0.503 | 240 |
| PFGM++ | 25 | 0.436 ± 0.011 | 0.422 ± 0.024 | 0.450 ± 0.025 | 0.514 | 243 |
| ReACGAN | 25 | 0.336 ± 0.016 | 0.326 ± 0.017 | 0.345 ± 0.026 | 0.510 | 247 |
| RESFLOW | 25 | 0.088 ± 0.008 | 0.090 ± 0.010 | 0.086 ± 0.011 | 0.498 | 199 |
| StyleGAN2-ada | 25 | 0.393 ± 0.012 | 0.387 ± 0.020 | 0.399 ± 0.022 | 0.506 | 328 |
| StyleGAN-XL | 25 | 0.399 ± 0.012 | 0.378 ± 0.018 | 0.420 ± 0.022 | 0.521 | 251 |
| WGAN-GP | 25 | 0.170 ± 0.013 | 0.158 ± 0.013 | 0.181 ± 0.022 | 0.511 | 221 |

Table 5: Detailed human experiment results on ImageNet

| Model | # Tests | Error Rate ↑ | R Error Rate ↑ | F Error Rate ↑ | R Answer Rate | Time (s) |
|---|---|---|---|---|---|---|
| ADM | 25 | 0.266 ± 0.016 | 0.254 ± 0.021 | 0.279 ± 0.018 | 0.513 | 369 |
| ADMG | 25 | 0.248 ± 0.012 | 0.236 ± 0.021 | 0.259 ± 0.016 | 0.512 | 396 |
| ADMG-ADMU | 25 | 0.269 ± 0.016 | 0.240 ± 0.017 | 0.298 ± 0.024 | 0.529 | 344 |
| BigGAN | 25 | 0.158 ± 0.014 | 0.155 ± 0.016 | 0.161 ± 0.019 | 0.503 | 326 |
| DiT-XL-2 | 25 | 0.286 ± 0.016 | 0.266 ± 0.026 | 0.307 ± 0.024 | 0.521 | 341 |
| DiT-XL-2-guided | 27 | 0.330 ± 0.013 | 0.329 ± 0.019 | 0.330 ± 0.017 | 0.500 | 352 |
| GigaGAN | 25 | 0.212 ± 0.015 | 0.212 ± 0.019 | 0.212 ± 0.015 | 0.500 | 398 |
| LDM | 26 | 0.309 ± 0.017 | 0.294 ± 0.023 | 0.323 ± 0.022 | 0.515 | 369 |
| Mask-GIT | 25 | 0.183 ± 0.016 | 0.164 ± 0.018 | 0.203 ± 0.025 | 0.520 | 278 |
| RQ-Transformer | 26 | 0.223 ± 0.012 | 0.205 ± 0.014 | 0.242 ± 0.017 | 0.518 | 308 |
| StyleGAN-XL | 25 | 0.153 ± 0.013 | 0.145 ± 0.014 | 0.162 ± 0.018 | 0.509 | 315 |

Table 6: Detailed human experiment results on LSUN-Bedroom

| Model | # Tests | Error Rate ↑ | R Error Rate ↑ | F Error Rate ↑ | R Answer Rate | Time (s) |
|---|---|---|---|---|---|---|
| ADM-dropout | 25 | 0.387 ± 0.020 | 0.334 ± 0.027 | 0.440 ± 0.022 | 0.553 | 306 |
| Consistency | 25 | 0.155 ± 0.022 | 0.131 ± 0.026 | 0.179 ± 0.027 | 0.524 | 325 |
| Diff-ProjGAN | 26 | 0.148 ± 0.019 | 0.130 ± 0.018 | 0.167 ± 0.023 | 0.519 | 335 |
| DDPM | 25 | 0.267 ± 0.021 | 0.259 ± 0.026 | 0.276 ± 0.021 | 0.509 | 354 |
| iDDPM | 25 | 0.297 ± 0.021 | 0.263 ± 0.025 | 0.331 ± 0.028 | 0.534 | 336 |
| Projected-GAN | 25 | 0.142 ± 0.018 | 0.144 ± 0.022 | 0.140 ± 0.017 | 0.498 | 266 |
| StyleGAN | 26 | 0.217 ± 0.018 | 0.179 ± 0.017 | 0.254 ± 0.025 | 0.537 | 427 |
| Unleash-Trans | 25 | 0.209 ± 0.022 | 0.178 ± 0.024 | 0.240 ± 0.027 | 0.531 | 305 |

Table 7: Detailed human experiment results on FFHQ

| Model | # Tests | Error Rate ↑ | R Error Rate ↑ | F Error Rate ↑ | R Answer Rate | Time (s) |
|---|---|---|---|---|---|---|
| Efficient-vdVAE | 25 | 0.088 ± 0.010 | 0.081 ± 0.013 | 0.096 ± 0.010 | 0.507 | 282 |
| InsGen | 25 | 0.255 ± 0.018 | 0.275 ± 0.025 | 0.236 ± 0.016 | 0.480 | 326 |
| LDM | 25 | 0.186 ± 0.016 | 0.195 ± 0.016 | 0.178 ± 0.022 | 0.492 | 429 |
| Projected-GAN | 25 | 0.120 ± 0.012 | 0.107 ± 0.011 | 0.134 ± 0.018 | 0.514 | 318 |
| StyleGAN2-ada | 25 | 0.215 ± 0.017 | 0.215 ± 0.019 | 0.216 ± 0.021 | 0.500 | 359 |
| StyleGAN-XL | 25 | 0.160 ± 0.012 | 0.163 ± 0.008 | 0.158 ± 0.021 | 0.498 | 340 |
| StyleNAT | 25 | 0.299 ± 0.019 | 0.314 ± 0.023 | 0.283 ± 0.022 | 0.485 | 387 |
| StyleSwin | 25 | 0.180 ± 0.012 | 0.198 ± 0.023 | 0.161 ± 0.010 | 0.481 | 288 |
| Unleash-Trans | 27 | 0.173 ± 0.015 | 0.142 ± 0.015 | 0.203 ± 0.022 | 0.530 | 330 |

Table 8: Layer used for GradCAM for each encoder.

| Model | Layer Name |
|---|---|
| Inception | blocks.3.2 |
| ConvNeXt | stages.3.blocks.2 |
| SwAV | layer4.2 |
| SimCLRv2 | net.4.blocks.2.net.3 |
| CLIP | visual.transformer.resblocks.11.ln_1 |
| DINOv2 | blocks.23.norm1 |
| MAE | blocks.23.norm1 |
| data2vec | model.encoder.layer.23.layernorm_before |

# D  Additional experimental results

## D.1  Additional scatter plots and correlations

Figure 10 displays additional views of Figure 4, showing trends as a function of other popular metrics (recall, precision, and density), while Figures 11 and 12 display the correlation of additional metrics.

## D.2  Representation spaces for generative evaluation

### D.2.1  Experimental details on GradCAM and additional heatmaps

We follow the gradient based visualization technique of [63]. As the FD is defined only between large sets of images (50k), the method pre-computes the representations of 50k images from the real dataset, and for a set of 49,999 generated images, then adds the additional image of interest using Grad-CAM to visualize the parts of the image that have the largest influence on FD. For ViT-based encoders, we use the Grad-CAM variation introduced by [37]. The layer GradCAM was applied to for each encoder can be found in Table 8.

To unify the color scheme across encoders, we changed the sign on the heatmap for CLIP, DINOv2, ConvNeXt and SimCLRv2. Indeed, the sign of the heatmap is given by the activations of the saliency layer that is visualized and does not reflect the sign of the gradient w.r.t the FD. Both bright yellow and deep blue can thus show an encoder's focus. The sign was changed for the listed encoders as it seemed to make better sense with the semantics of the images observed.

Additional heatmaps for all encoders on Imagenet, FFHQ and LSUN-Bedroom can be seen in Figures 13 and 14. On those heatmaps, the "generated" set is taken to be the real dataset, and not from any specific generative model.

### D.2.2  Quantitative representation spaces

In conjunction with the preceding qualitative analysis we performed two quantitative analyses of the representation spaces, and display the results in Figure 15. First, we measured the perceptual score in representation space using the BAPPS 2AFC dataset [118], data from a two alternative forced choice (2AFC) test that asked human evaluators which of two distorted views is more similar to a reference. The perceptual score measures the alignment of the human preference with the relative distances between the reference and patches in representation space. In the "traditional" experiment patches were distorted using traditional photometric, noise, blur, spatial, and compression distortions, while another experiment distorted patches using "CNN-based" distortions such as autoencoding, denoising, colorization, and super-resolution. Interestingly, better classifiers achieve worse perceptual scores [62]. We find that SimCLR ranks worst on the perceptual score while the masked methods of MAE and data2vec score the highest. These results are perhaps unsurprising given their respective self-supervised training procedures - the contrastive SimCLR relies on a set of heavy augmentations while MAE and data2vec work closer to pixel space and do not use augmentations - but such results point to clear differences between perceptual spaces as a direct result of the self-supervised training objective. We note that for generative evaluation, distortions such as jitter do not necessarily change the semantic information of the image and thus should not change the representation. Such distortions are a minor fraction of the total dataset.

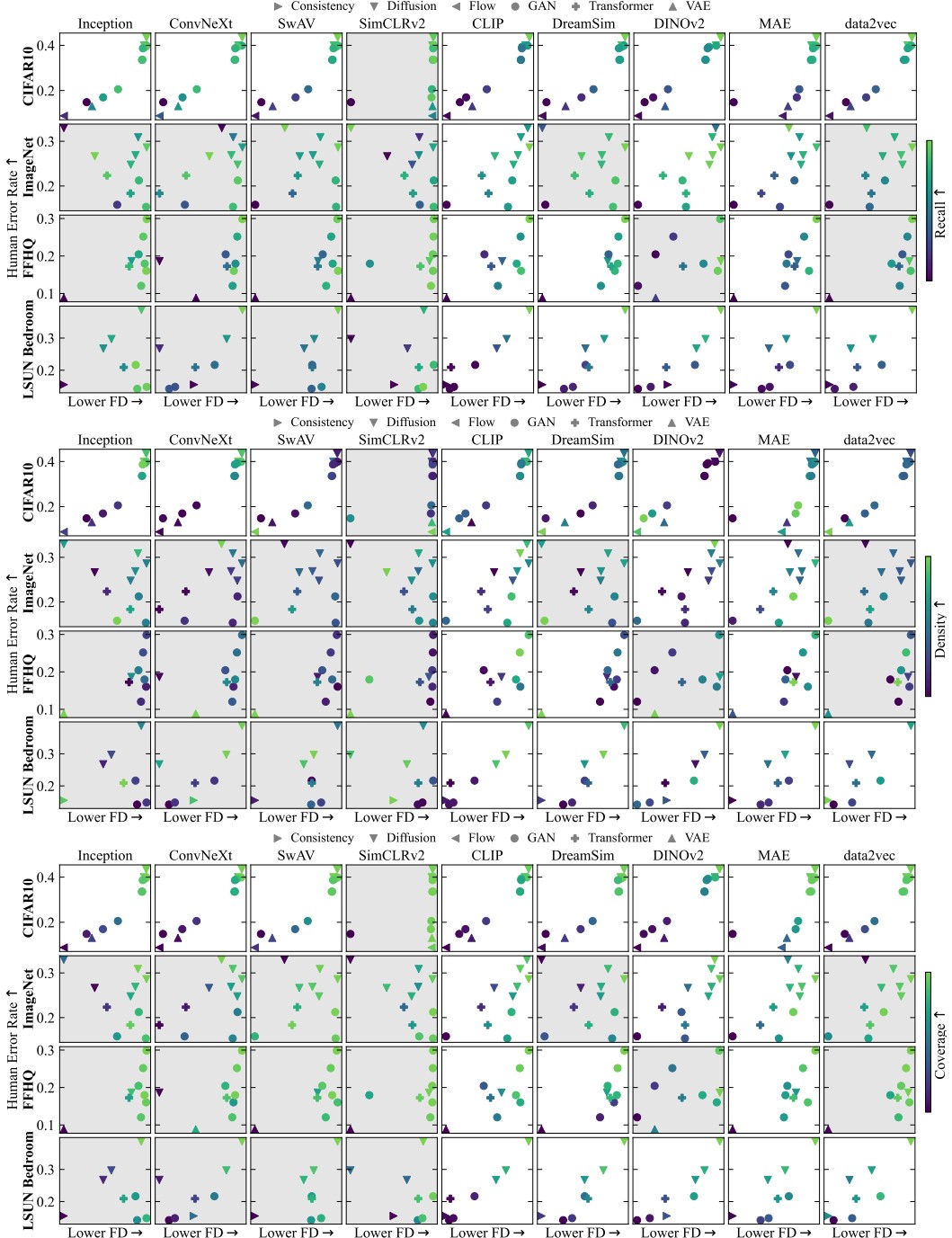

Figure 10: We show analogous results to Figure 4, with points coloured by additional metrics. Fréchet distance and human error rate for each generative model as measured by different encoders (columns) on different datasets (rows). Panels with a shaded background do not have strong ($|r| \geq 0.5$) and significant ($p \leq 0.05$) correlations between FD and human error rate.

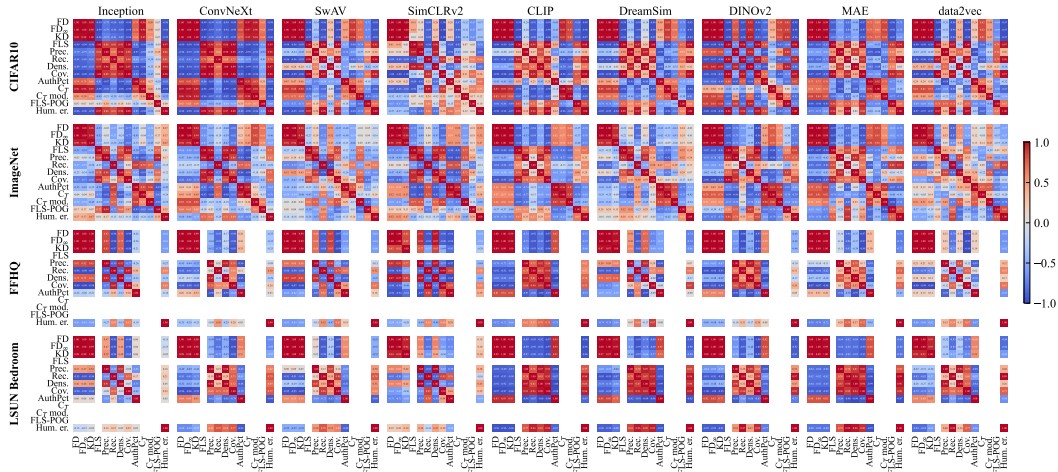

Figure 11: Pearson correlation for all metrics using different encoders (rows) on different datasets (columns). Missing metrics require a validation or test set, which some datasets do not have.

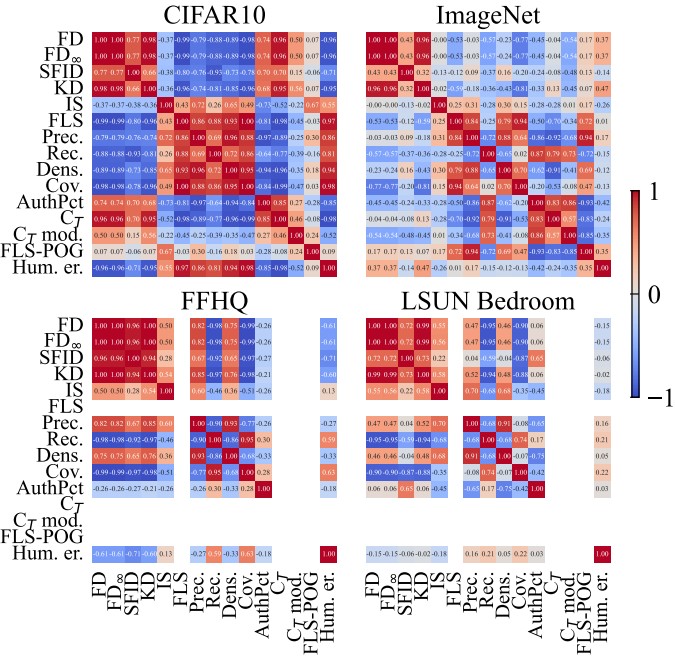

Figure 12: Pearson correlation for additional metrics (including IS and sFID) for the Inception-V3 network. Supporting the conclusions of the main text, we find that these additional metrics are not strongly correlated with human evaluation on datasets more complex than the simplistic CIFAR10. Missing metrics require a validation or test set, which some datasets do not have.

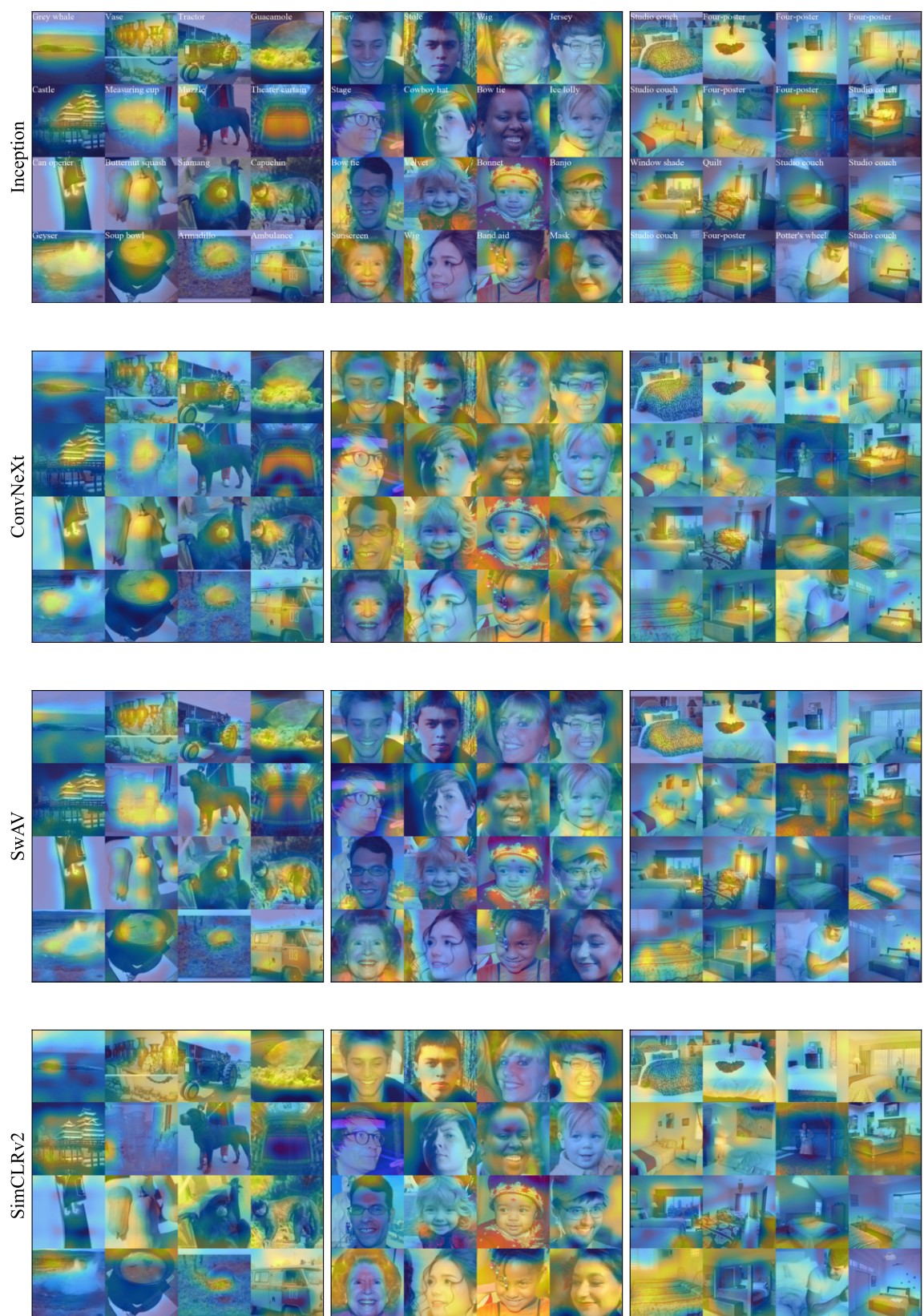

Figure 13: Additional GradCAM visualisations (1/2).

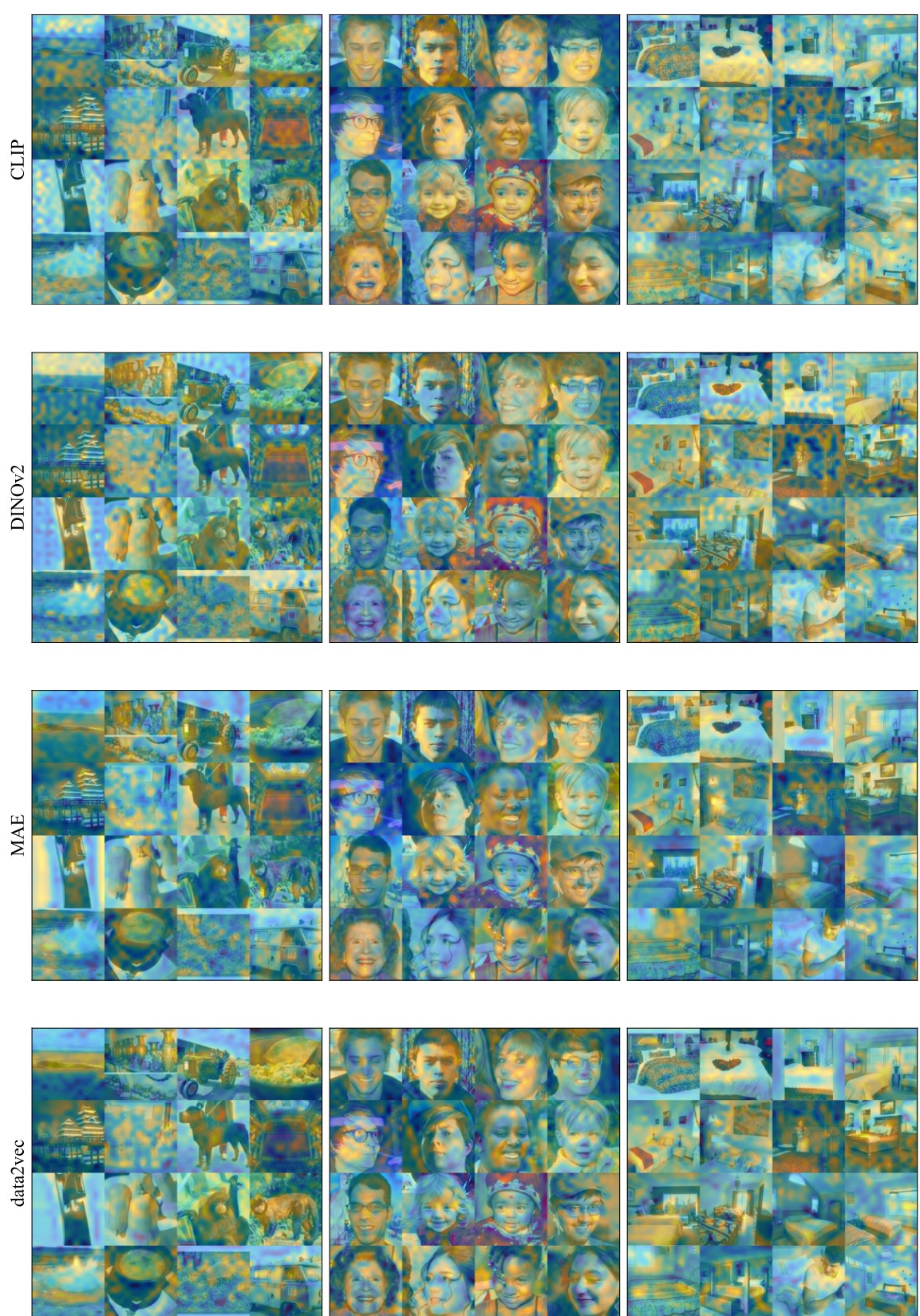

Figure 14: Additional GradCAM visualisations (2/2).

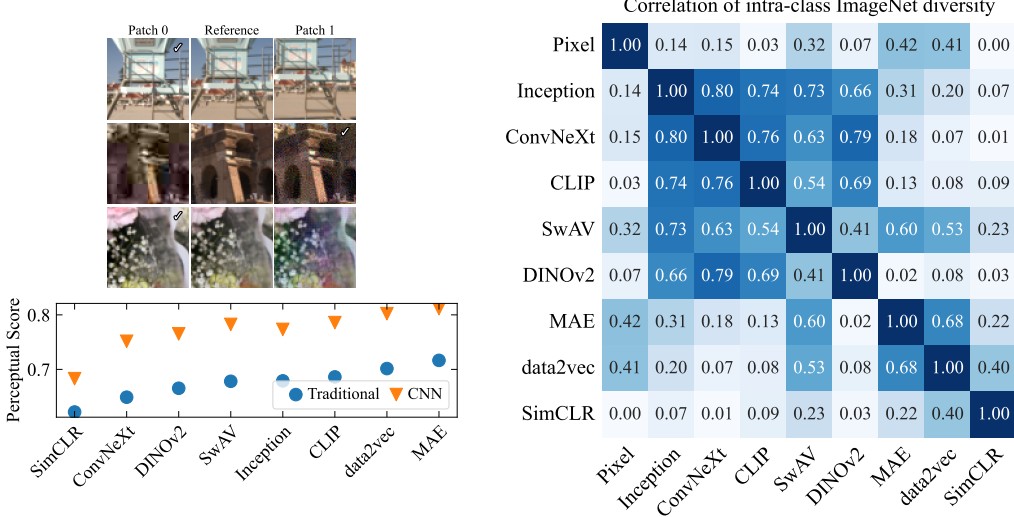

Figure 15: **Left:** Perceptual score [62] for each encoder measured using the BAPPS [118] dataset. Samples from the dataset are shown above, where ✓ denotes the patch that humans judged to be closer to the reference. **Right:** correlation of intra-class diversity on ImageNet.

To probe the level of semantic and pixel-wise diversity captured by each encoder we calculate the intra-class Vendi score over each class of the ImageNet training set (including in pixel space). This score is proportional to the effective number of independent samples within the class after transforming to the representation space. Section D.4 provides more details. A low score reflects a low diversity within the class, which is an indicator of a feature space that overemphasizes a small set of class specific features rather than the wide diversity of semantic in pixel-space. The correlation of the score over all classes across our set of encoders in shown in Figure 15. Sorting the encoders by their correlation with Inception we find two distinct groupings. The masked models, MAE and data2vec, have a intra-class diversity that is highly correlated with each other, but also with the diversity as measured in image-space. This indicates that such representation spaces put more weight towards low level image features rather than clustering classes by object semantics, which is a quantitative agreement with the characteristics of the heatmap visualizations. Interestingly MAE achieves high linear-classification accuracy but performs very poorly on $k$-nearest neighbours evaluation protocols [44], perhaps also indicating a lack of clustering based on class-specific semantic information. ConvNeXt, CLIP, SwAV, and DINOv2 show a high correlation with Inception yet a low correlation with image-space, indicating that their feature spaces distill a more object-based semantic information. Of the ViTs, CLIP has the smallest alignment with pixel space diversity and the largest with Inception space.

### D.3 FD bias

It is known that using finite datasets for computing FID (and similarly for FD in general) results in a biased estimate of the quantity that one would obtain if an infinitely large amount of data was observed. As mentioned in Appendix B.1, $FD_\infty$ [22] aims to remove this inherent bias. Here we measure the biases of FD and $FD_\infty$ in a realistic setting with a number of encoders used in our work in order to test whether the default 50k generated samples are sufficient for our proposed replacement to the Inception-V3 feature extractor.

We use the 100k samples generated with DiT-XL-2 [84] trained on ImageNet-256, and calculate the relevant statistics as a function of the number of training samples used by sub-sampling both the DiT generated set and the set of training images. Figure 16 shows the results across 4 encoders - Inception, CLIP, DINOv2, and MAE. We observe that while $FD_\infty$ has less bias compared to FD, it does not completely solve the problem, and continues to decrease with the number of samples used. We also find that the self-supervised encoders exhibit less biased FD than the Inception network. For example, DINOv2 with 20k samples has the same relative error compared to the full 100k set

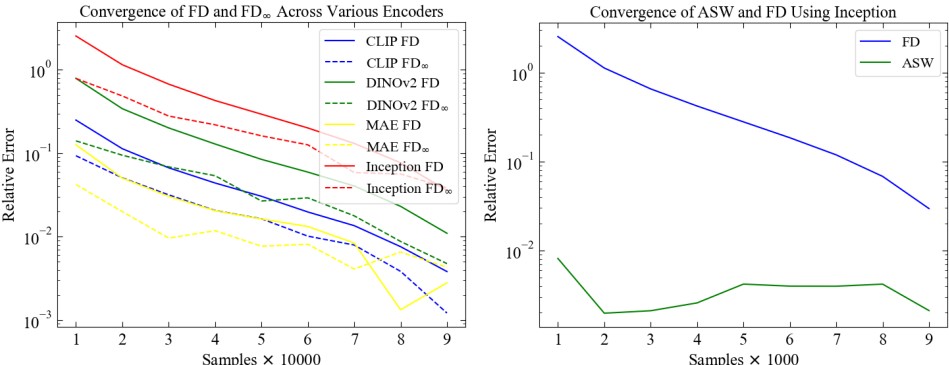

Figure 16: **Left:** Convergence of $FD_\infty$ and FD using different representation spaces. **Right:** Convergence of FD and ASW using Inception as the encoder.

as Inception does using 50k samples. While one could argue that this means only 20k samples are required for DINOv2, we retain consistency with FID and use 50k samples for FD with all encoders.

Naïvely, the FD thus has two issues: the aforementioned bias, and the fact that it only compares the first two moments of its corresponding distributions (see Appendix B.1). Since the FD can be understood as approximating its distributions with Gaussians of matching moments and then computing the Wasserstein-2 distance, $\mathbf{W}_2$, between them, it is sensible to consider other metrics based on Wasserstein distance as potential evaluation metrics. Unfortunately, computing this distance is in general computationally expensive [32].

In an attempt to address these issues, we considered using the sliced Wasserstein distance, $\mathbf{SW}_2$, instead. Let $\mathbb{P}$ and $\mathbb{Q}$ be distributions on $\mathbb{R}^d$, which we should think of as the generated and real distributions, respectively. Let $u_\#(\cdot)$ denote the push-forward of measures under the mapping $x \to \langle u, x \rangle$. Then $\mathbf{SW}_2$ is defined through:

$$\mathbf{SW}_2^2(\mathbb{P}, \mathbb{Q}) = \mathbb{E}_u\left[\mathbf{W}_2^2(u_\#\mathbb{P}, u_\#\mathbb{Q})\right], \tag{12}$$

where the expectation is taken with respect to the uniform measure on the unit sphere in $\mathbb{R}^d$.

It can be shown that $\mathbf{SW}_2$ has a lower sample complexity compared to both $\mathbf{W}_2$ and its relaxation [73]. Calculating optimal transport in one dimension requires significantly less computation than in higher dimensions as it can be solved via sorting. Therefore, the RHS of Equation 12 can be approximated well using Monte-Carlo methods. However, [74] introduces a closed form approximate sliced Wasserstein (ASW) distance with convergence guarantees that does not require Monte-Carlo, making the final result a formula based on the first two moments of $\mathbb{P}$ and $\mathbb{Q}$:

$$\text{ASW}(\{x_i^g\}_{i=1}^n, \{x_j^r\}_{j=1}^m) = \mathbf{W}_2\left(\mathcal{N}\left(0, \frac{M_2(\{x_i^g\}_{i=1}^n)}{d}\right), \mathcal{N}\left(0, \frac{M_2(\{x_j^r\}_{j=1}^m)}{d}\right)\right) + \frac{1}{d}\|\mu_g - \mu_r\|_2^2, \tag{13}$$

where

$$M_2(\{x_i^g\}_{i=1}^n) = \left(\frac{1}{n}\sum_{i=1}^n \|x_i^g - \mu_g\|_2^2\right) + \|\mu_g\|_2^2, \tag{14}$$

(and analogously for $M_2(\{x_j^r\}_{j=1}^m)$), and where $\mu_g$ and $\mu_r$ denote the sample mean of generated $\{x_i^g\}_{i=1}^n$ and real samples $\{x_j^r\}_{j=1}^m$, respectively. Note that the $\mathbf{W}_2$ term required to compute the ASW can be trivially computed through the 1-dimensional FD formula. The ASW resembles FD, but with the advantages of having both convergence and sample complexity guarantees.

To test the sample stability we use the same setting that we did for FD and $FD_\infty$. Figure 16 (right) depicts the relative error of ASW and compares it with FD using Inception. It is clear that the ASW has less dependence on sample size, indicating that even small batches of around 10k samples may be sufficient (note that the plot is $\log$-scale).

Despite its theoretical guarantees, we found that ASW does not correlate well with human perception across various encoders and does not seem to be affected by mode shrinkage. Since it is not widely

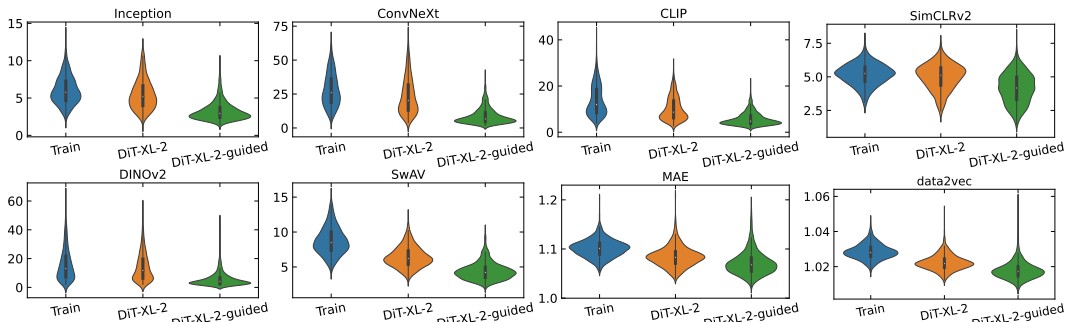

Figure 17: Per-class Vendi score (higher is more diverse) distributions with different encoders. By visual inspection, DiT-XL-2 is more diverse than DiT-XL-2-guided.

used in the literature, we chose to not include it in the main text. These results do however raise the question: why does the FD perform so well in practice compared to the ASW, despite the latter being seemingly more principled? We hypothesize that there are two factors at play: $(i)$ the encodings are likely approximately Gaussian (e.g. [52]), meaning that comparing the first two moments is enough; and $(ii)$ while the computation of the FD is biased, if the bias behaves similarly across generated datasets, it will have no impact in model rankings. While we leave a formal exploration of these hypotheses for future work, we point out that the strong correlation of FD with KD (Figure 4) suggests that the FD only comparing the first two moments and having sample complexity issues are not problems that are borne out in practice, and we thus recommend using FD as-is given that its use is already widespread.

### D.4    Diversity

In this section, we outline the rationale behind our decision to utilize the per-class Vendi score with DINOv2 as the metric for our diversity analysis.

Based on an analysis of the DiT-XL-2 and DiT-XL-2-guided datasets, the strong classifier guidance value of DiT-guided in comparison to DiT resulted in samples that lacked diversity within each class (examples were shown in Figure 6). We pose that a robust diversity metric should flag such a lack of intra-class diversity, and thus these two models are the basis of the following analysis. As mentioned in the main manuscript, much like precision measures more than just fidelity (Figure 4, and similarly for density, see Figure 10), we find that recall and coverage also quantify more than just diversity, which can be clearly seen in Figure 10 (as both of these metrics have relatively strong correlations with human evaluation when using good encoders – a property that diversity metrics should not exhibit). On the other hand, the per-class Vendi score, which we will further justify shortly, has the intended property of being uncorrelated with human error rate (Figure 5, right).

We computed the Vendi score for the DiT-XL-2 and DiT-XL-2-guided models over each individual ImageNet class (100 images per class), and averaged the results, which are presented in Table 9. The distributions of the per-class Vendi scores are plotted in Figure 17. The per-class Vendi scores consistently ranked DiT-XL-2 higher than DiT-XL-2-guided for every encoder, indicating that DiT-XL-2 exhibits higher intra-class diversity. This observation aligns with our inspection, where it is apparent that the guided model produces less diverse intra-class samples. Hence, we adopted the per-class Vendi score for our analysis.

We next report the overall (not conditioned on class) Vendi score in Table 10, using 50k images from each set. It is important to note that Vendi scores represent the effective number of unique samples in a dataset. Therefore, it is counterintuitive to obtain very low values, such as those observed in MAE, data2vec, SimCLRv2, and even with simple resizing, where there are only a few effective samples among the 50k images. Moreover, not all encoders ranked DiT-XL-2 above DiT-XL-2-guided based on the overall Vendi scores, except for DINOv2, SimCLRv2, and ConvNext. However, SimCLRv2 exhibited minimal differences across ImageNet train, DiT-XL-2, and DiT-XL-2-guided. Additionally, neither SimCLRv2 nor ConvNext showed FD rankings that aligned well with human perception. Consequently, we pose that this diversity analysis provides additional support for the use of DINOv2 for generative evaluation.

Table 9: Mean Vendi score for different encoders on ImageNet train and generated samples.

| Encoder | ImageNet train | DiT-XL-2 | DiT-XL-2-guided |
|---------|----------------|----------|-----------------|
| Inception | 6.04 | 5.50 | 3.25 |
| ConvNeXt | 28.08 | 23.52 | 8.73 |
| CLIP | 13.74 | 10.17 | 5.72 |
| SimCLRv2 | 5.17 | 4.99 | 4.15 |
| DINOv2 | 15.93 | 14.17 | 5.61 |
| SwAV | 8.72 | 6.42 | 4.40 |
| MAE | 1.10 | 1.08 | 1.07 |
| data2vec | 1.03 | 1.02 | 1.02 |
| Resize (to 32x32) | 1.09 | 1.09 | 1.09 |

Table 10: Overall diversity for ImageNet training and generated images.

| Encoder | ImageNet train | DiT-XL-2 | DiT-XL-2-guided |
|---------|----------------|----------|-----------------|
| Inception | 88.57 | 89.03 | 129.92 |
| ConvNext | 951.83 | 824.33 | 687.58 |
| CLIP | 36.56 | 27.61 | 31.90 |
| SimCLRv2 | 6.74 | 6.77 | 6.38 |
| DINOv2 | 762.52 | 688.53 | 671.54 |
| SwAV | 55.61 | 43.51 | 59.77 |
| MAE | 1.15 | 1.14 | 1.17 |
| data2vec | 1.04 | 1.04 | 1.04 |
| Resize(to 32x32) | 1.12 | 1.13 | 1.18 |

## D.5 Rarity

Here we perform experiments to investigate whether participants confuse fakeness with unlikeliness. Such an effect would result in images generated from more diverse models being more likely to be assessed as "fake" and hence a diverse model receiving a lower human error rate compared to a model of lower quality but less diversity. The following experiments rule out such a flaw in our experimental setup and ensure that our human error rate dataset is not misleading.

To isolate this effect, we on focus comparing the error rate on "rare" real samples versus "common" real samples. For each of the 2000 real images we used for each dataset (individual images were evaluated by an average of 13 humans), we determined the fraction of humans that labeled it as fake, while as a measure of image rarity/unlikeliness we calculate the "rarity score" (RS) of [42]. The RS is determined on image representations, so we performed the calculation using both Inception-v3 and DINOv2 to quantify the dependency of RS on the embedding space.

Table 11: Correlation of the fraction of humans that labelled a real image as fake and the rarity score [42] of that image. The rarity score can only be determined for images that fall "on manifold".

| Dataset | Encoder | % on manifold | r | p-value | r (94%) | p-value (94%) |
|---------|---------|---------------|------|---------|---------|---------------|
| CIFAR10 | Inception | 76.3 | 0.278 | 0.000 | 0.216 | 0.000 |
| CIFAR10 | DINOv2 | 88.3 | 0.062 | 0.009 | 0.0004 | 0.859 |
| ImageNet | Inception | 82.2 | -0.034 | 0.163 | -0.002 | 0.922 |
| ImageNet | DINOv2 | 92.9 | 0.008 | 0.726 | -0.012 | 0.624 |
| LSUN-Bedroom | Inception | 70.3 | 0.110 | 0.000 | 0.045 | 0.106 |
| LSUN-Bedroom | DINOv2 | 90.2 | 0.102 | 0.000 | 0.021 | 0.386 |
| FFHQ | Inception | 79.3 | -0.027 | 0.286 | -0.024 | 0.357 |
| FFHQ | DINOv2 | 89.7 | -0.019 | 0.431 | -0.040 | 0.103 |

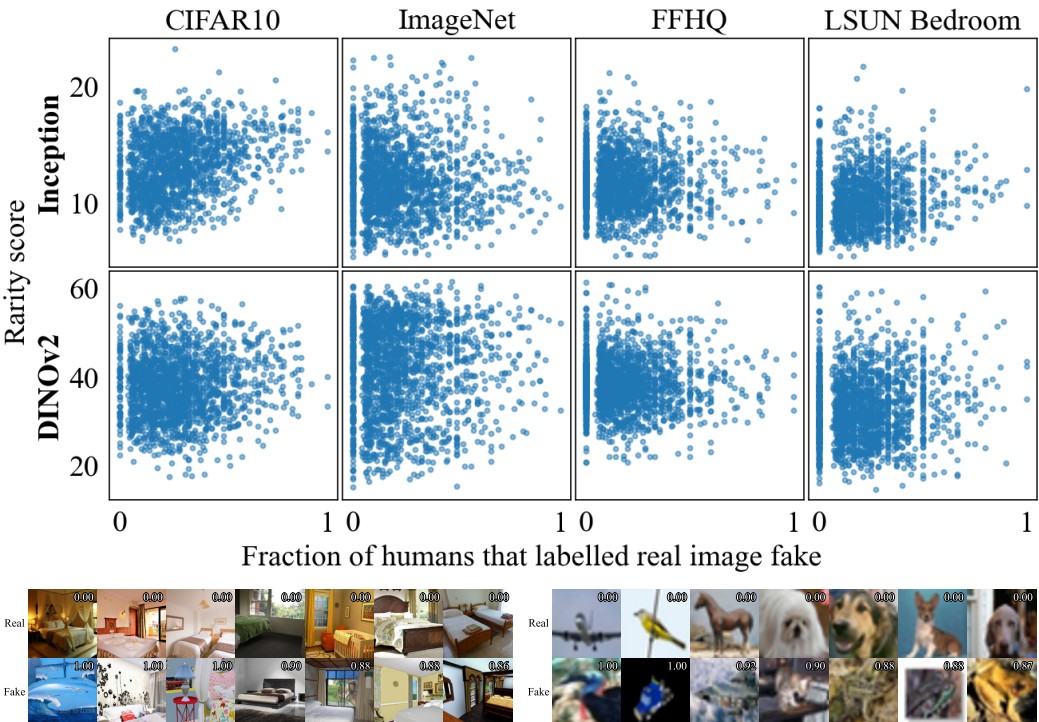

Figure 18: **Top**: Human error rate and rarity score for each of the 2000 training images. Individual images were evaluated by an average of 13 humans. **Bottom**: Examples on LSUN Bedroom (**left**) and CIFAR10 (**right**), with the top rows displaying training images correctly judged by humans as real, and the bottom rows displaying real images "incorrectly" judged as fake. The fraction of participants who judged the image as fake is denoted on the top right of each image.

As seen in Figure 18 and Table 11 we find no correlation between the human error rate and RS on ImageNet and FFHQ. On CIFAR-10 and LSUN-Bedroom we find a small but statistically significant correlation, which we identify as driven by dataset issues: the non-zero correlation is caused by a very small percentage of "real" images which are clearly taken from 3D-generated scenes (instead of bedroom photographs in LSUN-Bedroom), or from 2D-generated scenes or low quality images (in CIFAR-10). These results show that $(i)$ humans are more likely to label generated scenes as fake/generated (LSUN-Bedroom, CIFAR-10), and $(ii)$ humans are more likely to label low-quality images as fake/generated (CIFAR-10). Such images have a higher than average RS, and hence the small correlation between human evaluation and RS on CIFAR10 and LSUN-Bedroom is due to humans properly identifying these dataset issues. We find that removing just 6% of the "fakest" (as measured by humans) real images on LSUN-Bedroom removes the correlation of RS and human error rate - quantitative proof that the small correlation is driven by dataset issues, and not due to humans associating diversity with fakeness. Rare defects in the training set are not enough to affect our results: being so rare ( 6%) means they barely affect the average error rate; and this training-set effect is the same for every generative model evaluation and thus does not change their rankings.

This analysis provides additional validation that our experimental design and human error rate metric are accurately measuring the fidelity of images, and are not effected by extraneous factors such as image unlikeliness/rarity.

## D.6 Memorization

In this subsection we include a detailed description of our procedure to $(i)$ collect memorized samples for generative models trained on CIFAR10, ImageNet, FFHQ, and LSUN-Bedroom, $(ii)$ summarize a number of controlled experiments to probe the effectiveness of memorization metrics in ideal scenarios, and $(iii)$ show additional results for memorization metrics in different representation spaces.

**Memorization check with calibrated $l_2$ distance**  In Figure 19, we show the memorized ratio for each of the 13 generative models we evaluated on CIFAR10 using the metrics described in Section 5.2. We also tailored $k$ – which is the hyperparameter specifying the number of nearest neighbours used to compute the metric – to each dataset. We use $k = 50$ for CIFAR10, FFHQ, and LSUN, while for ImageNet with its limited examples per class in the training set, we set $k = 3$. In addition, we exclusively conduct intra-class nearest neighbour search on ImageNet, as opposed to performing the search across the entire training set. We find that more complex models are more likely to memorize on CIFAR10, whereas less performant models such as ResFLOW and NVAE are less likely to memorize.

Moreover, we provide additional visualization results to further showcase the exact memory exhibited from DDIM on CIFAR10 in Figure 20, and the reconstructive memory exhibited from DiT-XL-2 on ImageNet in Figure 21 and from ADM-dropout on LSUN in Figure 22. Based on Figure 20, the presence of exact memorization is evident, raising concerns that need to be considered on CIFAR10. However, each "memorized" sample found with the above metric in Figure 21 on ImageNet and Figure 22 on LSUN – while not being pixel-wise identical – retains a higher degree of semantic similarity compared to the three alternative matches in the training set. We can also observe that, for some cases, the generated image does not exhibit significantly closer similarity to the matched training sample compared to the training sample's nearest neighbors. While it seems that the model tends to show reconstructive memory on samples that have similar duplicates in the training set, we note that this marks potential failures of the memorization metric, and thus do not report a total number of samples without further work to understand the reasons behind this. This memory can also be interpreted not necessairily as memorization, but as a form of generalization, as the model captures and reproduces common patterns or characteristics present in the training data. Our analysis leads us to conclude that reconstructive memory on complex datasets is not currently a major concern, as out of millions of training images we found a small number of matches. Nevertheless, we re-iterate our stance from the main text that reconstructive memory should continue to be monitored as models themselves become increasingly more complex.

Other than the pixel-wise check, we also revisit the nearest neighbour search in DINOv2 representation space in small scale and find decently-matched results for complex datasets; qualitatively, the results are similar to the pixel-wise check from Figure 21. This could be even further explored in future work. Yet it is still unclear whether or not we have exact memorization - the matched cases mostly occur within classes exhibiting low intra-class diversity, where the samples in training set are extremely similar with each other.

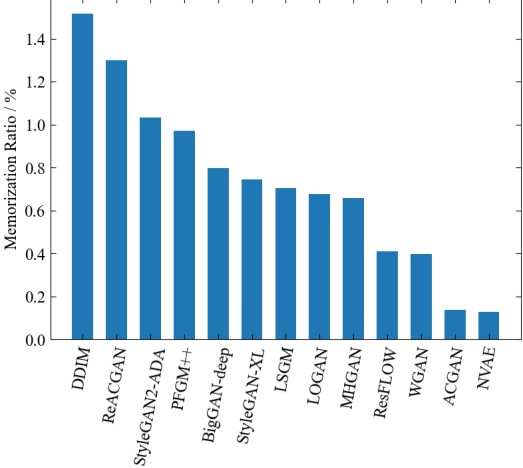

Figure 19: Memorization ratio for 13 generative models on CIFAR10.

### D.6.1  Controlled experiment for memorization metrics

To study the response of metrics to memorization we first constructed a controlled experiment where generated samples become increasingly memorized from a realistic set of training images.

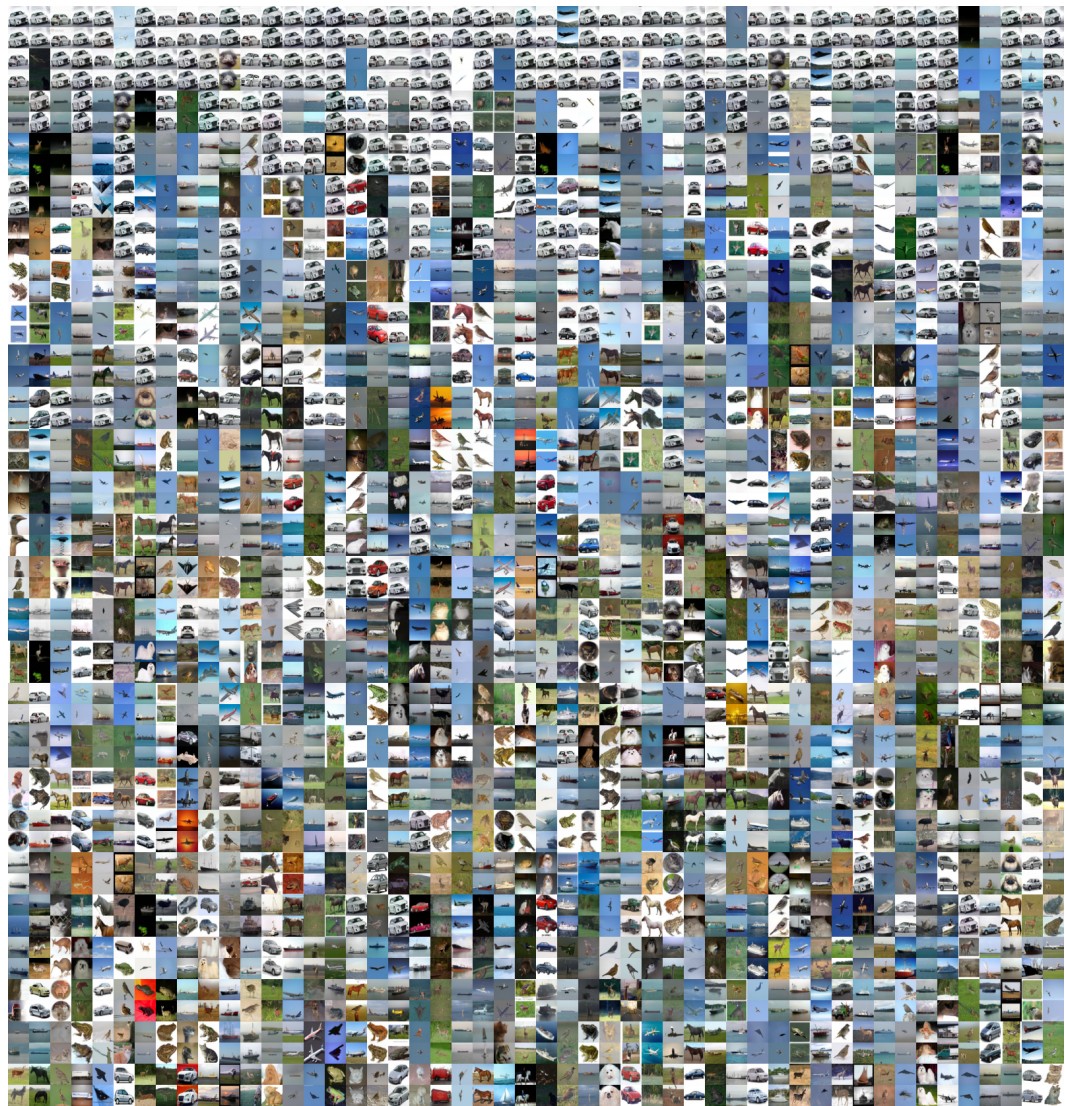

Figure 20: Memorized samples from DDIM on CIFAR10. Images are arranged in vertical pairs, where the upper image is generated, and the lower image is a training set image.

We generate the synthetic datasets using the VDVAE architecture [21], which is a hierarchical VAE. Specifically, we used an FFHQ-256 pretrained model, which has 62 layers, and for each training example we sampled from the approximate posterior until a certain layer, and sampled from the prior in later layers. In this manner, going deeper with the approximate posterior corresponds to more closely memorizing the training example. In particular, using depth 0 means sampling from the model's prior, while using the posterior until depth 62 corresponds to a reconstruction that is a very close reproduction of the original image. For a qualitative example of depth and memorization refer to Figure 23.

Formally, a VAE consists of three functions: $(i)$ the *generator* $p_\theta(x|z)$, $(ii)$ the *approximate posterior* $q_\phi(z|x)$, and $(iii)$ the *prior* $p_\theta(z)$, where $z$ is the latent variable. In the hierarchical setting, it is assumed that $z = (z_0, z_1, \cdots, z_n)$ and the prior and the approximate posterior are factorized as:

$$p_\theta(z) = p_\theta(z_0)p_\theta(z_1|z_0)\cdots p_\theta(z_n|z_{<n}), \tag{15}$$

$$q_\phi(z|x) = q_\phi(z_0|x)q_\phi(z_1|x, z_0)\cdots q_\phi(z_n|x, z_{<n}). \tag{16}$$

The distribution of the image conditioned on this $z$ is $p_\theta(x|z)$ i.e. the generator. To sample from this model, $p_\theta(z)$ is used to sample $z$ and then to sample $x$ conditioned on $z$. This corresponds to our

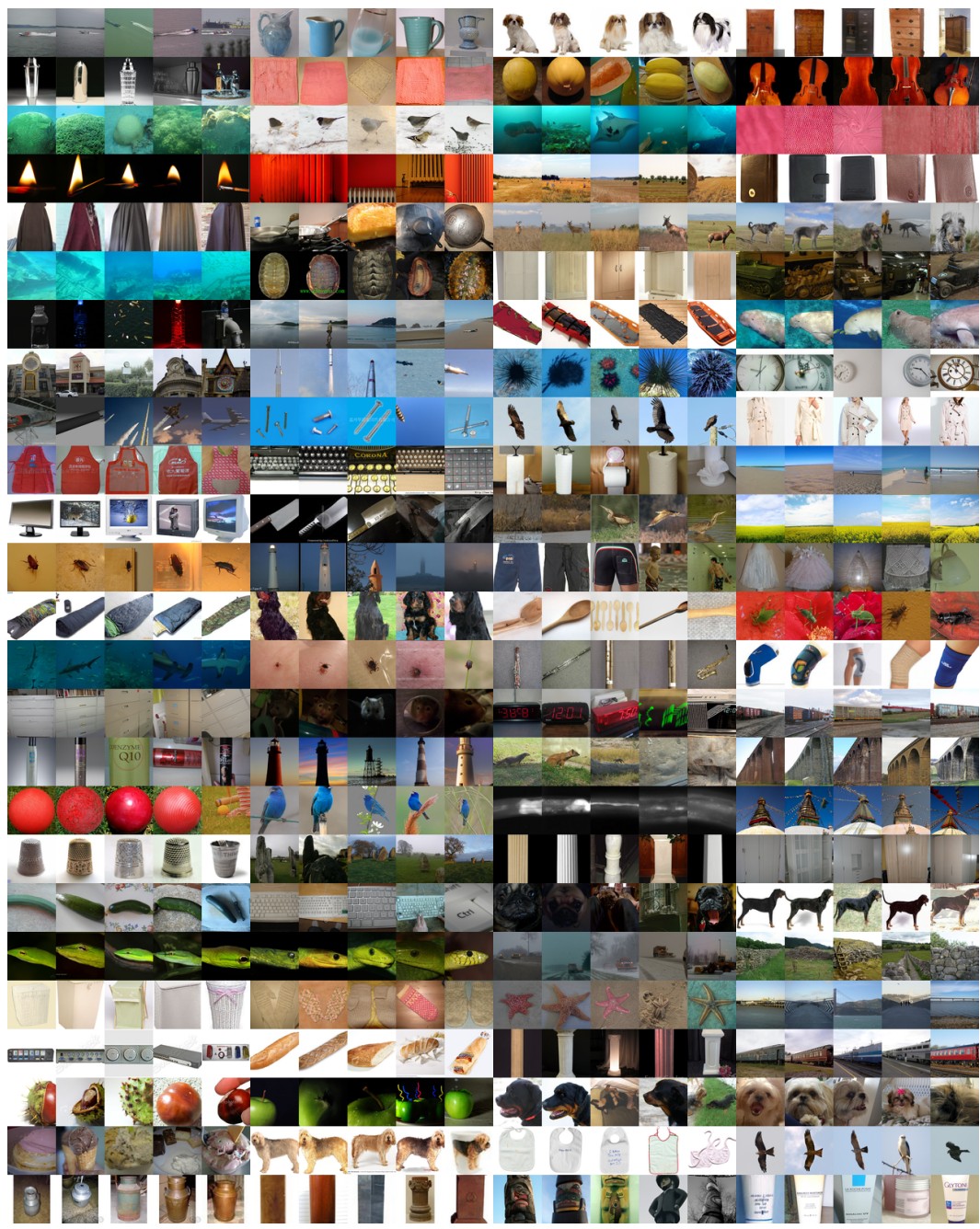

Figure 21: Reconstructive memorized samples from DiT-XL-2 on ImageNet. Each set of 5 images in a row includes a generated sample, a matched training sample, and three nearest neighbors from the training set associated with the matched training sample.

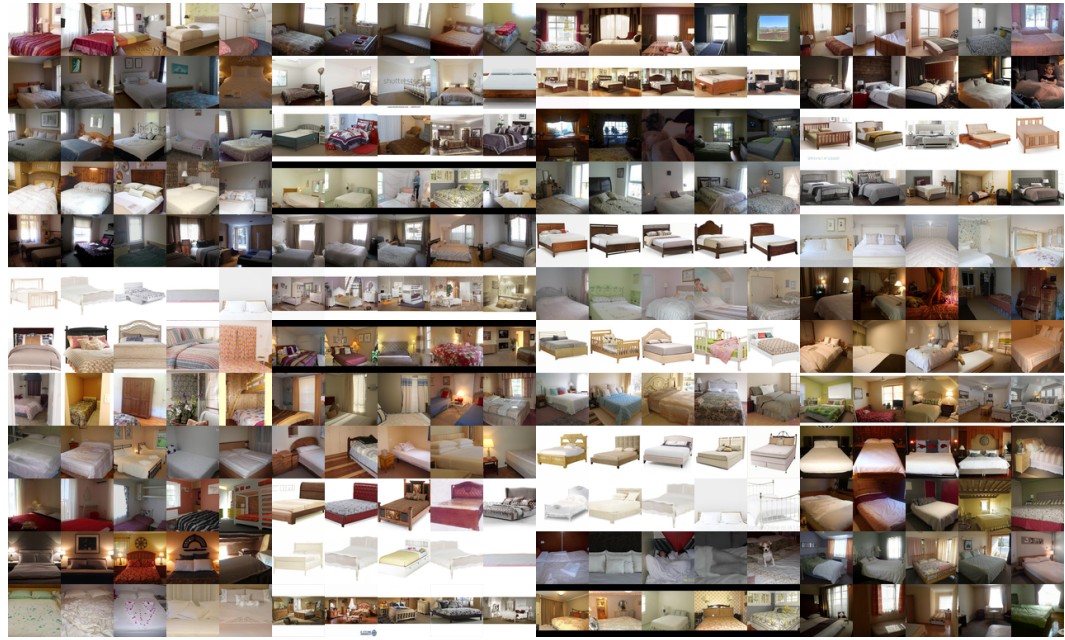

Figure 22: Reconstructive memorized samples from ADM-dropout on LSUN. Each set of 5 images in a row includes a generated sample, a matched training sample, and three nearest neighbors from the training set associated with the matched training sample.

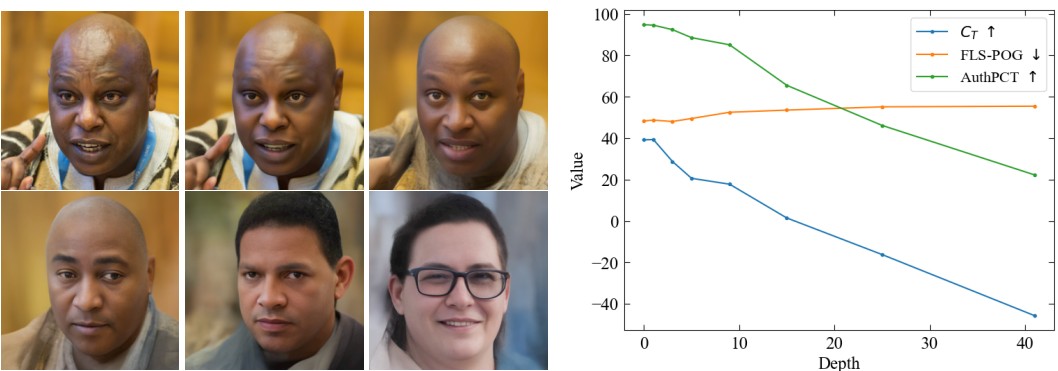

Figure 23: **Left**: Qualitative example of synthetic data for the controlled memorization experiment. Top left image is a training sample and the next 5 images correspond to using the approximate posterior up to depth 41, 15, 5, 3, and 0. **Right**: Values of the $C_T$ score, AuthPct, and FLS-POG at each depth. The DINOv2 encoder was used for these metrics.

depth 0 setting. To make samples resemble the training set, we can use $q_\phi(z|x)$ instead of $p_\theta(z)$. The distribution is modified as follows

$$p(z|x) = q_\phi(z_0|x) \cdots q_\phi(z_m|x, z_{<m}) p_\theta(z_{m+1}|z_{<m+1}) \cdots p_\theta(z_n|z_{<n}). \qquad (17)$$

In terms of sampling, this corresponds to using the training image $x$ and the approximate posterior to generate $z_0, z_1, \cdots, z_m$ and then using the prior to generate the rest of $z$. As $m$ (which we refer to as depth) increases, the mutual information between $x$ and $z$ increases, which in turn results in a more faithful reconstruction.

Using the described procedure, 30,000 samples are generated at each of the following depths: 0 (meaning no conditioning), 1, 3, 5, 9, 15, 25, and 41. We use 63,000 images as the training set and 7,000 as test images; these are the same sets that are used to train and test the VDVAE. We calculate AuthPct, $C_T$, and FLS-POG on each set of images and display the results in Figure 23. We find that both $C_T$ score and AuthPct have characteristics that indicate they can correctly identify memorization

in this experiment. At depths less than 15, $C_T$ is a positive value, indicating the training samples are no closer to the generated samples than they are to ones from the test set (are not memorized), while at depths greater than 15, $C_T$ becomes increasingly negative, indicating a stronger detection of memorization. AuthPct starts from close to 100%, meaning all samples are authentic, and decreases to 20% at a depth of 41, indicating that generated samples are closer to training samples than expected. Since a depth of zero means no conditioning on individual training images (except for the training of the original model), the result confirms the effectiveness of the proposed method for this particular point. As the depth increases, AuthPct correctly decreases the percentage of authentic samples in a rate that is almost equal to the decrease in $C_T$. FLS-POG has the correct trend as it increases as samples become more memorized. However, the rate of the increase is minimal, does not reflect the level of memorization seen at a depth of 41 (as seen in the images training samples which are almost exactly reproduced at this depth), and is not comparable to the change in the other two metrics. From this controlled experiment we conclude that, in the absence of mode collapse/mode shrinkage when training samples are uniformly memorized, $C_T$ and AuthPct can flag memorization in practice while FLS-POG does not flag memorization strongly enough.

### D.6.2 Memorization metric quality

With a better understanding of AuthPct, FLS-POG, and the $C_T$ score, we calculate them on the sets of CIFAR10 generated images and compare to the measured memorization ratios determined through the analysis above. Results for the DINOv2 representation space were shown in Section 5.2, while here we show the scores for every encoder in Figure 24. We also include our modification to the $C_T$ score which swaps the roles of the training and test samples (details and motivation for the modified $C_T$ score follow in Appendix D.6.3). We remind the reader that *higher* AuthPct, $C_T$ score, and modified $C_T$ score are meant to indicate *less* memorization, whereas *lower* FLS-POG is meant to indicate *less* memorization.

Here we detail the characteristics of each encoder. For Inception, we see that a higher memorization ratio seems to correctly correlate with lower AuthPct and $C_T$ score. However there is no discernible pattern for FLS-POG and $C_T$ modified. Meanwhile, the SwAV encoder seems to provide the correct directionality for *all* memorization metrics. CLIP produces no discernible trend for AuthPct and FLS-POG, a slightly accurate trend for $C_T$, and a strongly accurate trend for modified $C_T$. MAE is similar to CLIP except with the roles of FLS-POG and $C_T$ perhaps flipped. data2vec is somewhat similar to SwAV with the correct directionality – particularly for $C_T$ score – but with a slightly weaker trend for the other metrics. Lastly, DINOv2 shows a correct correlation for the modified $C_T$ score, but no strong trends for the others. While the performance of each encoder shows a large variation, it is important to note that we illustrate clear issues with the memorization metrics in the next section (D.6.3). We find that modified $C_T$ score performs the best in this section, with $C_T$ score next, then AuthPct, then FLS-POG last. This agrees with our previous experiment that shows FLS-POG is the weakest detector of memorization.

### D.6.3 Low-dimensional experiments for memorization metrics

Table 12: Memorization metrics on synthetic datasets

| Metric | True Distribution | Shrinkage | Memorized | Underfit 1 | Underfit 2 | Underfit 3 |
|---|---|---|---|---|---|---|
| AuthPct | 41.20 | 20.00 | 0.00 | 46.60 | 67.20 | 77.30 |
| $C_T$ Score | -0.23 | -16.14 | -25.26 | 7.16 | 15.22 | 18.42 |
| $C_T$ Mod. | 0.05 | -0.86 | -16.71 | -0.62 | 0.23 | 1.37 |
| FLS-POG | 60.10 | 60.00 | 89.70 | 59.00 | 56.30 | 55.20 |

While the previous subsections have confirmed that $C_T$ and AuthPct can flag memorization in ideal scenarios (Section D.6.1) and seemingly show appropriate trends in some representation spaces (Section D.6.2), it is not clear that they are explicitly flagging memorization over other forms of overfitting. Curiously, the metrics computed against the test set resemble those against the training set, which the models cannot have memorized outside of data-leakages between the construction of the train and test sets. While we note that such leakages do occur for CIFAR10 [6], in the following toy scenario we demonstrate the inability of the metrics to separate types of overfitting from memorization.

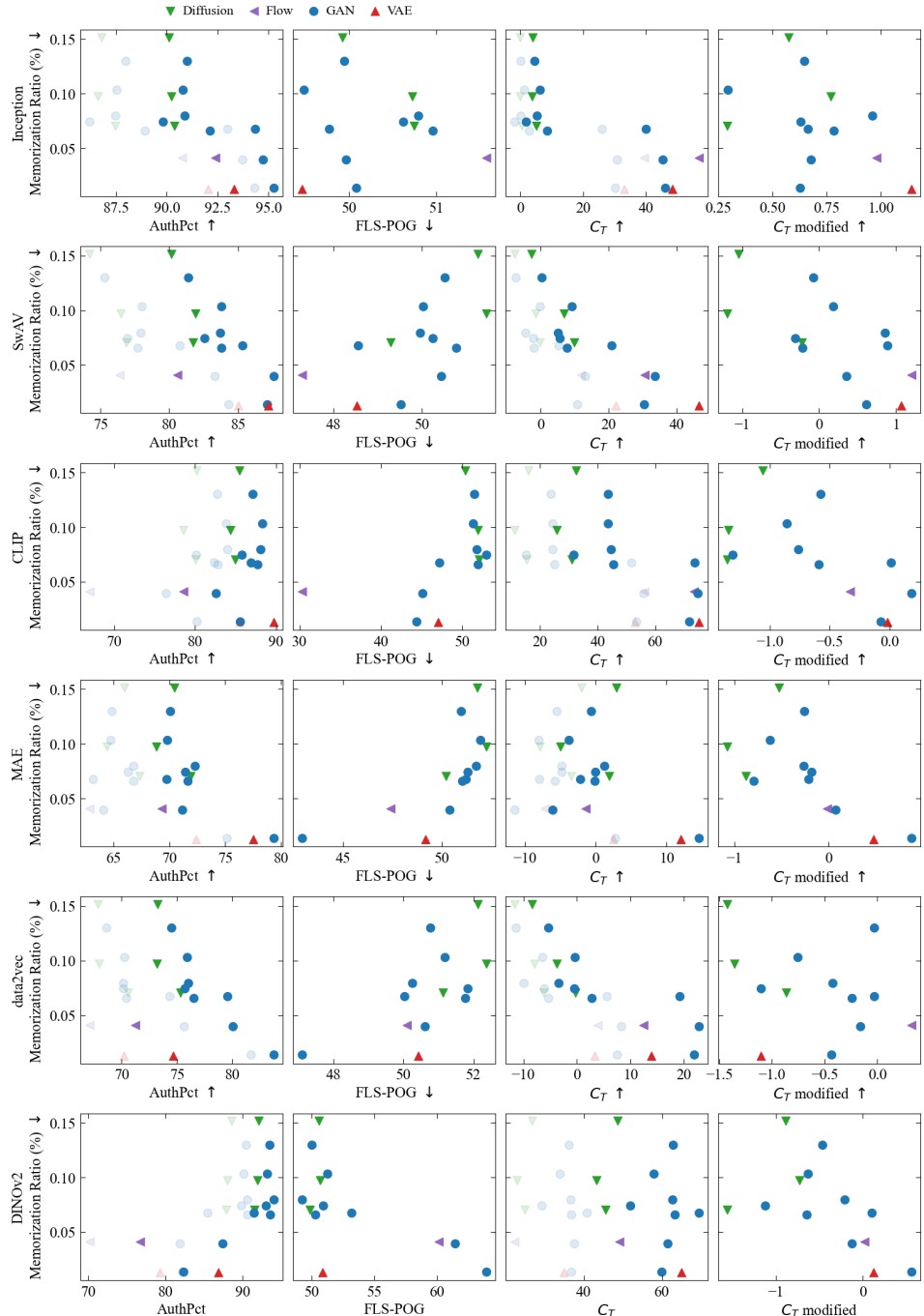

Figure 24: Memorization ratio vs. metric value for each metric out of AuthPct, FLS-POG, $C_T$, and the modified $C_T$, evaluated on the models from Figure 19. The same metric values computed against the test set are shown with lower opacity.

To understand the impact of various modelling phenomena on memorization metrics we design a simple 2-dimensional experiment. As a ground truth distribution, we take a uniform mixture of 5 Gaussian components. Each component's mean is sampled from a unit diagonal Gaussian distribution, while each component's covariance matrix is diagonal with diagonal elements sampled from a uniform distribution over the interval $[0.01, 0.09]$. From this distribution we sample 1000 training and 1000 test points.

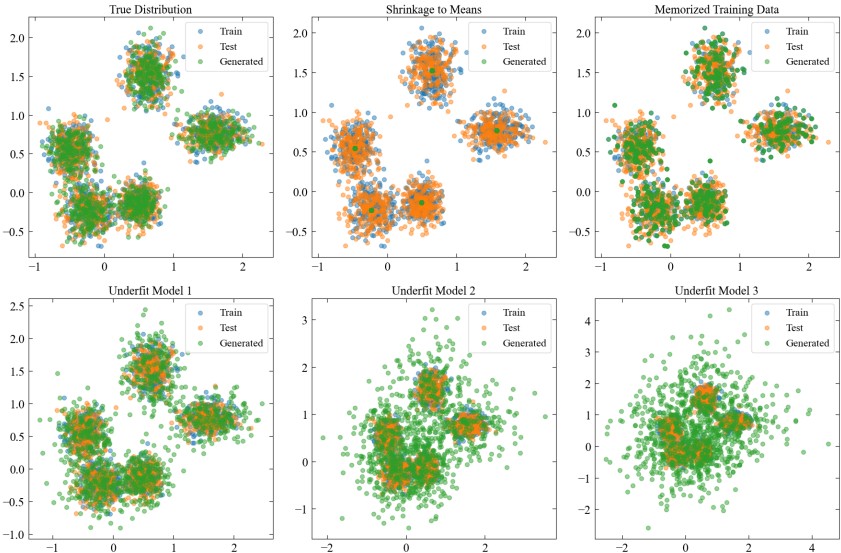

Figure 25: Toy datasets generated for the purpose of investigating memorization metrics.

We then generate 1000 points from various "models" from which to measure memorization:

- A perfect model with zero memorization, represented by independent samples from the same ground truth distribution.

- A model experiencing total mode shrinkage: it uniformly samples a mean from each Gaussian in the mixture.

- A model that has perfectly memorized the training data: it amounts to randomly sampling from the training data with replacement.

- Three "underfit" models, which in fact have learned the true distribution with upscaled standard deviation for each component of the Gaussian mixture: $1.5\sigma, 3\sigma,$ and $4.5\sigma$.

The training, test, and generated distributions are shown in Figure 25. We evaluate metrics on these datasets and report the values in Table 12.

FLS-POG seemingly performs correctly for detecting memorization in these regimes - with the memorized model receiving the highest score of 80% and the other models receiving a score hovering around 60%. However, a value of 60% on both the true and mode shrinkage examples indicates that it poorly flags mode shrinkage, one of its claimed usages [53], and the relative change of 5% from underfit to the true distributions is on the order of the change seen in the VDVAE controlled experiment, indicating that such relative changes can have multiple meanings. AuthPct indicates memorization ($< 50\%$) for the mode shrinkage model, the perfect model, and the slightly underfit model. From this, we gather that AuthPct incorrectly flags well-trained models that produce samples on the true manifold.

We also find that the $C_T$ score performs incorrectly here: the score of $-16.14$ it assigns to the mode shrinkage model – well below $0$ – has incorrectly flagged memorization. We can see why by looking closer at the metric: the $C_T$ score compares how much more often $(i)$ the distance from a generated data point to a training data point is lower than $(ii)$ the distance from a test data point to a training data point, and reports a highly negative score if $(i)$ is often lower than $(ii)$. In this case, we would expect our generated, mode-shrunken samples to be much closer to the training data than the test data is to the training data, which agrees with what we observe. The issue here boils down to a lack of symmetry between test and training set: the test set is only used for one of the comparisons, whereas the training set is used in both distance checks. The generated dataset is also only used once. We address this asymmetry by proposing a modification to the $C_T$ score – referred to as the modified $C_T$ score – wherein we essentially swap the roles of the training and the generated data within the base $C_T$ score. The modified $C_T$ score thus compares how much more often $(i)$ the distance from a generated data point to a training data point is lower than $(ii)$ the distance from a generated data

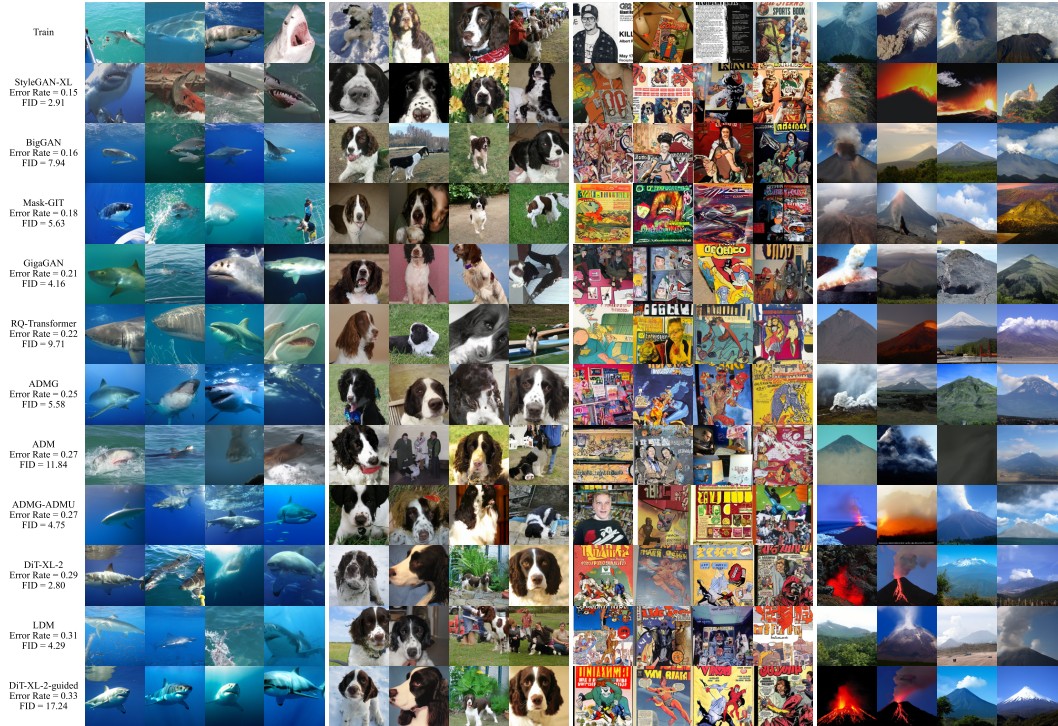

Figure 26: ImageNet samples for generative models (rows) sorted by error rate from our human subject experiments. Classes from left to right: great white shark, English Springer, comic book, and volcano.

point to a test data point, reporting a highly negative score if $(i)$ is often lower than $(ii)$. To fully summarize and perhaps visualize our change, consider the $C_T$ score as a function from training data $T_1$, generated data $G$, and test data $T_2$ to the real numbers, represented as $C_T(T_1, G, T_2)$. Then, the modified $C_T$ score – call it $C_{T,M}$ – can be represented as $C_{T,M}(T_1, G, T_2) \equiv C_T(G, T_1, T_2)$. We see that our change has produced the desired behaviour, as now the mode shrinkage dataset is not detected as memorized by $C_{T,M}$.

From this discussion we gather that the $C_T$ score incorrectly flags mode shrinkage, and AuthPct incorrectly flags well-trained models. Recall that none of the models except for the memorized one are actually derived from the test set. These experiments explain why AuthPct and $C_T$ seem to score models in a way that is almost invariant to the training set in Figure 7.

### D.7 Model samples

Figures 26, 27, 29, and 28 show image samples from the 41 generative models across the four datasets.

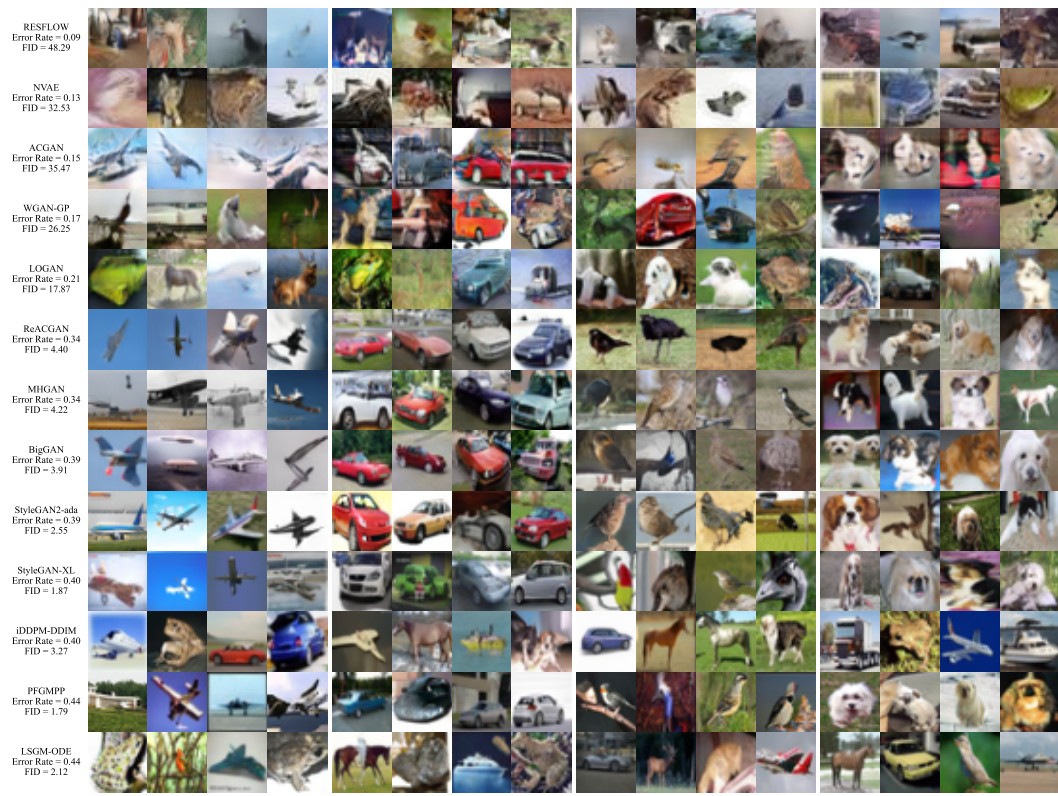

Figure 27: CIFAR10 samples for generative models (rows) sorted by error rate from our human subject experiments. Non class-conditional models show a random selection of generated samples while class-conditional models show classes airplane, automobile, bird, and dog.

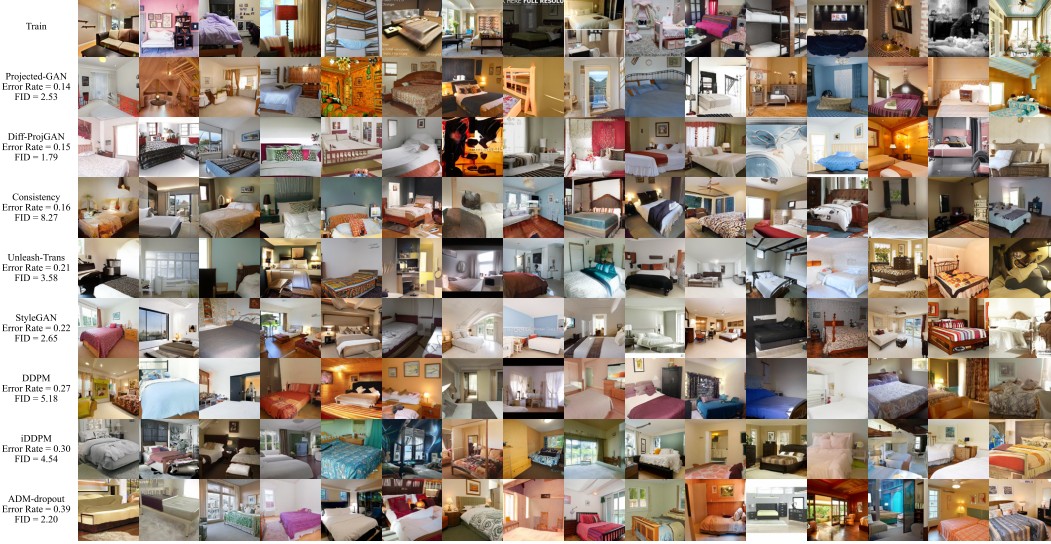

Figure 28: LSUN Bedroom samples for generative models (rows) sorted by error rate from our human subject experiments.

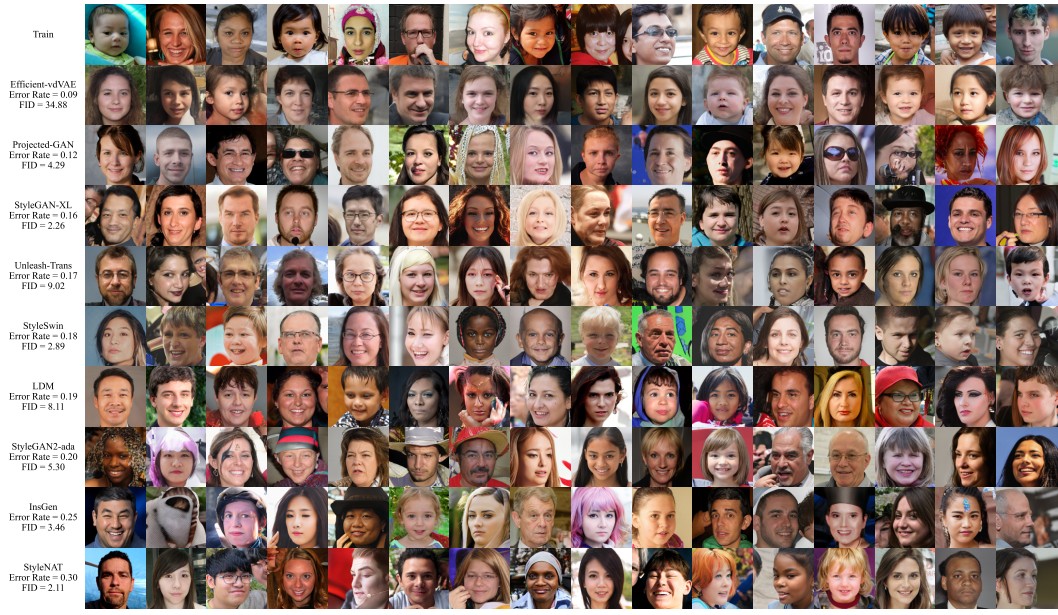

Figure 29: FFHQ samples for generative models (rows) sorted by error rate from our human subject experiments.

# E   FD$_{\text{DINOv2}}$ scores leaderboard

Table 13 shows our considered metrics across models and datasets.

Table 13: DINOv2-ViT-L/14 leaderboard.

| Dataset | Model | Human error rate | FD | $FD_\infty$ | KD | FLS | Precision | Recall | Density | Coverage | AuthPct | $C_T$ | $C_T$ mod. | FLS-POG |
|---|---|---|---|---|---|---|---|---|---|---|---|---|---|---|
| CIFAR10 | RESFLOW | 0.09±0.01 | 1301.38 | 1296.52 | 2.64 | 49.15 | 0.87 | 0.01 | 1.91 | 0.05 | 76.72 | 49.17 | 0.02 | 10.13 |
| | ACGAN | 0.15±0.01 | 1208.26 | 1205.42 | 2.41 | 52.31 | 0.84 | 0.01 | 2.06 | 0.07 | 82.38 | 59.90 | 0.56 | 13.93 |
| | WGAN-GP | 0.17±0.01 | 1088.56 | 1083.16 | 2.04 | 54.62 | 0.82 | 0.04 | 1.60 | 0.08 | 87.45 | 61.46 | -0.12 | 11.46 |
| | NVAE | 0.13±0.02 | 921.34 | 916.05 | 1.57 | 58.85 | 0.76 | 0.27 | 1.27 | 0.11 | 86.91 | 65.01 | 0.12 | 0.87 |
| | LOGAN | 0.21±0.02 | 881.73 | 876.35 | 1.51 | 59.38 | 0.71 | 0.25 | 0.98 | 0.12 | 91.51 | 69.40 | 0.10 | 3.17 |
| | MHGAN | 0.34±0.01 | 358.07 | 353.90 | 0.49 | 76.84 | 0.70 | 0.47 | 0.69 | 0.41 | 93.56 | 63.28 | -0.64 | 0.31 |
| | ReACGAN | 0.34±0.02 | 353.25 | 349.76 | 0.45 | 76.73 | 0.68 | 0.46 | 0.66 | 0.42 | 93.49 | 62.87 | -0.46 | -0.02 |
| | BigGAN | 0.39±0.01 | 326.66 | 320.71 | 0.39 | 77.31 | 0.69 | 0.44 | 0.66 | 0.44 | 94.04 | 62.64 | -0.21 | -0.76 |
| | StyleGAN2-ada | 0.39±0.01 | 305.92 | 300.96 | 0.45 | 81.55 | 0.71 | 0.51 | 0.70 | 0.52 | 93.22 | 57.98 | -0.63 | 1.23 |
| | iDDPM-DDIM | 0.40±0.01 | 212.30 | 207.69 | 0.26 | 85.37 | 0.74 | 0.66 | 0.78 | 0.64 | 92.13 | 48.64 | -0.89 | 0.57 |
| | StyleGAN-XL | 0.40±0.01 | 204.60 | 198.78 | 0.24 | 84.10 | 0.72 | 0.55 | 0.69 | 0.60 | 93.03 | 51.93 | -1.12 | 0.90 |
| | LSGM-ODE | 0.44±0.01 | 148.27 | 143.14 | 0.18 | 87.95 | 0.74 | 0.74 | 0.76 | 0.71 | 91.56 | 45.60 | -1.56 | -0.14 |
| | PFGM++ | 0.44±0.01 | 141.65 | 137.50 | 0.17 | 88.13 | 0.75 | 0.71 | 0.80 | 0.72 | 91.96 | 43.23 | -0.73 | 0.66 |
| | Test Set | - | 31.07 | 1.03 | -0.00 | 324.78 | 0.88 | 0.88 | 1.01 | 0.97 | 72.23 | - | - | - |
| ImageNet 256×256 | BigGAN | 0.16±0.01 | 401.22 | 393.13 | 0.52 | 66.48 | 0.81 | 0.39 | 0.95 | 0.54 | 92.11 | 121.94 | -1.56 | 2.86 |
| | RQ-Transformer | 0.22±0.01 | 304.05 | 294.75 | 0.34 | 70.43 | 0.76 | 0.76 | 0.69 | 0.58 | 94.30 | 124.25 | -1.61 | 1.58 |
| | GigaGAN | 0.21±0.01 | 228.37 | 219.27 | 0.24 | 74.61 | 0.81 | 0.77 | 0.73 | 0.68 | 93.77 | 106.24 | -2.73 | 2.99 |
| | StyleGAN-XL | 0.15±0.01 | 214.88 | 207.09 | 0.22 | 75.16 | 0.82 | 0.72 | 0.76 | 0.68 | 95.02 | 103.29 | -1.52 | 1.52 |
| | Mask-GIT | 0.18±0.02 | 214.45 | 206.36 | 0.24 | 78.89 | 0.84 | 0.76 | 0.77 | 0.75 | 92.60 | 97.40 | -1.58 | 2.71 |
| | ADM | 0.27±0.02 | 203.45 | 195.01 | 0.19 | 77.41 | 0.80 | 0.85 | 0.67 | 0.72 | 93.34 | 104.43 | -2.03 | 0.30 |
| | ADMG | 0.25±0.01 | 123.27 | 114.69 | 0.09 | 83.56 | 0.87 | 0.84 | 0.80 | 0.85 | 92.35 | 70.19 | -1.73 | 2.23 |
| | LDM | 0.31±0.02 | 112.40 | 103.73 | 0.09 | 87.92 | 0.92 | 0.76 | 1.00 | 0.89 | 89.56 | 51.33 | -1.74 | 3.43 |
| | ADMG-ADMU | 0.27±0.02 | 111.24 | 102.63 | 0.08 | 85.07 | 0.88 | 0.84 | 0.86 | 0.86 | 90.91 | 59.56 | -2.87 | 1.56 |
| | DiT-XL-2-guided | 0.33±0.01 | 99.16 | 90.04 | 0.03 | 97.23 | 0.99 | 0.60 | 1.31 | 0.95 | 61.46 | -43.29 | -5.16 | 5.66 |
| | DiT-XL-2 | 0.29±0.02 | 79.36 | 70.32 | 0.06 | 92.20 | 0.93 | 0.84 | 1.01 | 0.94 | 85.46 | 34.32 | -3.25 | 3.20 |
| | Validation Set | - | 21.04 | 12.38 | 0.00 | 124.59 | 0.95 | 0.94 | 1.00 | 0.96 | 69.83 | - | - | - |
| LSUN Bedroom | Projected-GAN | 0.14±0.02 | 636.35 | 634.58 | 2.43 | - | 0.80 | 0.23 | 0.78 | 0.24 | 93.60 | - | - | - |
| | Diff-ProjGAN | 0.15±0.02 | 547.61 | 544.73 | 2.05 | - | 0.79 | 0.28 | 0.74 | 0.29 | 93.85 | - | - | - |
| | Unleash-Trans | 0.21±0.02 | 440.04 | 437.68 | 1.48 | - | 0.78 | 0.41 | 0.67 | 0.44 | 94.25 | - | - | - |
| | Consistency | 0.16±0.02 | 428.99 | 425.76 | 1.66 | - | 0.83 | 0.23 | 0.75 | 0.45 | 91.72 | - | - | - |
| | StyleGAN | 0.22±0.02 | 239.79 | 236.64 | 0.72 | - | 0.85 | 0.41 | 0.87 | 0.71 | 90.21 | - | - | - |
| | DDPM | 0.27±0.02 | 229.76 | 227.69 | 0.65 | - | 0.79 | 0.61 | 0.68 | 0.68 | 92.65 | - | - | - |
| | iDDPM | 0.30±0.02 | 166.19 | 163.85 | 0.45 | - | 0.83 | 0.64 | 0.77 | 0.76 | 91.01 | - | - | - |
| | ADM-dropout | 0.39±0.02 | 59.64 | 57.73 | 0.13 | - | 0.85 | 0.75 | 0.93 | 0.90 | 87.60 | - | - | - |
| FFHQ 256×256 | Projected-GAN | 0.12±0.01 | 592.26 | 589.20 | 2.03 | - | 0.57 | 0.07 | 0.31 | 0.30 | 96.91 | - | - | - |
| | StyleGAN2-ada | 0.20±0.02 | 514.78 | 511.68 | 1.64 | - | 0.59 | 0.06 | 0.36 | 0.39 | 96.70 | - | - | - |
| | Efficient-vdVAE | 0.09±0.01 | 514.16 | 511.19 | 1.57 | - | 0.86 | 0.14 | 1.04 | 0.54 | 90.74 | - | - | - |
| | InsGen | 0.25±0.02 | 436.26 | 432.68 | 1.33 | - | 0.64 | 0.13 | 0.46 | 0.51 | 96.04 | - | - | - |
| | Unleash-Trans | 0.17±0.01 | 393.45 | 390.60 | 1.06 | - | 0.76 | 0.24 | 0.61 | 0.53 | 94.43 | - | - | - |
| | StyleSwin | 0.18±0.01 | 303.21 | 300.18 | 0.79 | - | 0.79 | 0.28 | 0.71 | 0.64 | 93.75 | - | - | - |
| | StyleGAN-XL | 0.16±0.01 | 240.07 | 236.98 | 0.56 | - | 0.77 | 0.43 | 0.68 | 0.63 | 93.10 | - | - | - |
| | StyleNAT | 0.30±0.02 | 229.42 | 226.04 | 0.56 | - | 0.79 | 0.41 | 0.77 | 0.71 | 91.91 | - | - | - |
| | LDM | 0.19±0.02 | 226.72 | 223.64 | 0.55 | - | 0.81 | 0.44 | 0.83 | 0.74 | 91.30 | - | - | - |

## F  Author contributions

Table 14 lists individual contributions of the authors (in alphabetical order).

Table 14: Author contributions

|  | AC | BR | ET | GL | GS | JC | RH | VV | YS | ZL |
|---|---|---|---|---|---|---|---|---|---|---|
| Conceived of ideas for project initially |  | X |  | X | X | X |  |  |  |  |
| Conceived human experiments |  |  | XX | X | X | XX |  |  |  |  |
| Created, performed, & analyzed human exp. |  |  | X |  |  | XX |  |  |  |  |
| Trialed human experiments | X | X | X | X | X | X | X | X | X | X |
| Generated CIFAR10 datasets |  | X |  |  | X |  |  |  | X | X |
| Generated ImageNet datasets | X |  |  |  | X |  | X |  | X | X |
| Generated FFHQ datasets |  | X |  |  | X |  |  | X | X |  |
| Generated LSUN-Bedroom datasets |  | XX |  |  |  |  |  |  |  | X |
| Implemented encoders |  | X |  |  | X |  | X | X |  | X |
| Grad-CAM encoder understanding |  |  |  | X | X |  |  | XX |  | X |
| Designed metrics |  | X |  | X | X |  | X |  | X |  |
| Implemented metrics |  | X |  | X | X |  | X |  | X | XX |
| Memorization design | X | X |  | X | X |  | X |  |  | X |
| Memorization implementation |  | X |  |  | X |  | X |  |  | X |
| Diversity |  |  |  | X | X |  |  |  | X |  |
| Other experiments |  |  |  |  | X | X |  |  |  |  |
| Structured Codebase |  | XX |  |  | X | X |  | X | XX |  |
| Prepared figures |  | X |  | X | X | X | X | X |  | X |
| Analyzed data |  | X |  | X | X | X | X | X | X | X |
| Wrote paper | X | X | X | XX | XX | XX | X | X | X | X |
| Managed the project |  |  |  | X | XX |  |  |  |  |  |

