# OpenReview forum: "Exposing flaws of generative model evaluation metrics and their unfair treatment of diffusion models"
_NeurIPS.cc/2023/Conference — NeurIPS 2023 poster_

### Official Review · Reviewer_SaoU · 2023-06-20

**Soundness:** 3 good
**Presentation:** 3 good
**Contribution:** 3 good
**Rating:** 6
**Confidence:** 4

**Summary:**

The authors performed extensive experimental study on various image-based generative models. Based on the study, it showed that no existing metric strongly correlated with human evaluations. The authors also included alternative self-supervised features extractors for evaluation. Additionally, data memorization was investigated. The experiments revealed limitations of existing evaluation metrics for generative models.

**Strengths:**

The authors implemented extensive experiments to evaluate various image-based generative models from different perspectives, eg., encoders, human evaluations, diversity, and memorization. The evaluation was performed using several state-of-the-art evaluation metrics including humans. The comparison results are expected to have a high impact on evaluating existing generative models.

**Weaknesses:**

The experiments showed that no existing metric strongly correlated with human evaluations. There is some concern about how the human error rate is calculated. The reviewer wonders whether the conclusion would change by improving the calculation of the human error rate. See detailed comments under Questions.


**Questions:**

. Human error rate: this metric is calculated as the fraction of images which were incorrectly classified. The incorrectly classified cases include both real-->fake and fake-->real. From the reviewer's perspective, these two cases should be separately treated, and the fake-->real case should be more important for evaluation. For example, if there are a lot of real-->fake cases from one human participant, this probably indicates that this human participant has some judgment issues and his/her results cannot be trusted. One thing we probably can try is to use the real-->fake case to evaluate whether this human participant can be trusted or not, and then use the fake-->real case as the human error rate. Of course, there are definitely other ways to improve that.

. Memorization: how to evaluate pixel-wise memorization and reconstructive memorization mathematically? Based on these two metrics, how to set the threshold values to determine memorized samples?

**Limitations:**

The authors have discussed the limitations of their work, and probably will address them in the future work.

---

> ### Author Rebuttal · Authors · 2023-08-10
>
> Thank you for your review, we appreciate the time and effort that went into evaluating our work, and we are glad you found that the “results are expected to have a high impact on evaluating existing generative models”.
>
> - On concerns with the “human error rate” metric: The human error rate metric was established and explored in [91] which we build on in our work. We agree that the individual real-->fake and fake-->real metrics are useful for various diagnostic purposes, but there are a number of conceptual reasons why we prefer the full human error rate as in [91] for ranking generative models by human preference. Still, we included the three metrics for all generative models in Appendix C.3 as the “Error Rate”, “R Error Rate“, “F Error Rate”, respectively, and they are all very highly correlated.
>
>     - On a conceptual basis, the average of R Error Rate and F Error Rate, what we simply call Error Rate, is our preferred choice to rank models as it has a few properties that neither R Error Rate nor F Error Rate have in isolation. Focusing on the suggestion that “the fake-->real case should be more important for evaluation”, we note that F Error Rate does not detect a scenario of “hyper-realistic generation”, where fake images look more realistic than the dataset on which a model is trained. For example, on our ImageNet and LSUN-bedroom datasets, some images from the training set look “less realistic” due to effects of data curation (lower resolution, aspect ratios, etc.). Nevertheless, these are real images that the model was trained on, and a generative model which has learned the true data distribution should be capable of generating images with these features. The F Error Rate metric does not capture this, but R Error Rate does, and hence Error Rate does as well.
>     - On a quantitative basis, we note that R Error Rate, F Error Rate, and Error Rate are very highly correlated with each other - *Pearson’s correlation coefficient between F Error Rate and Error Rate over all models is 0.99, and replacing the Error Rate with the F Error Rate results in no change to the trends seen in Figure 4 or the rest of the results*. One implication of the high correlation is that R Error Rate is not constant across experiments using the same training dataset, but different generated datasets, as one might suppose *a priori*. Instead, when the generated images are more realistic, humans make more mistakes on the real images as well. This phenomenon was previously discussed in [91], so we only discussed it in Appendix C.3.
>     - On a diagnostic basis, when analyzing individual participant trials, we found no evidence of a significant divergence of the measures (i.e. we saw no evidence of a high R Error Rate and low F Error Rate, which would be evidence of a participant simply selecting “fake” for every image). With the observation above that R Error Rate is higher when generated images are more realistic, we cannot filter out participants simply for having a high R Error Rate, because this is a normal outcome on the more challenging tasks. Thus, using this metric as a diagnostic to rule out poor performing participants would introduce a bias into the final results, as we would have to set a model-dependent threshold on the R Error Rate guided by prior knowledge.
>
> - On memorization: The questions you raise on defining the concepts of reconstructive memorization mathematically are very interesting, but to our knowledge there is no consistent formal definition for this in the literature and it is an open-ended question that deserves further study. While pixel-wise memorization is more direct to evaluate, it still requires a choice of threshold values to determine a sample as memorized. We followed the examples set by [11] and [58] for our analyses in Section 4.3, and included full details for reproducibility in Appendix B.2. We performed the memorization analysis as we felt it was necessary to determine if models with good FD_DINOv2 scores (i.e. mostly diffusion models) are just memorizing the training data, and thus “cheated” their human alignment; indeed we found that this is not the case. To our knowledge a memorization analysis has not been performed on the datasets in our paper.

---

> > ### Comment · Reviewer_SaoU · 2023-08-11
> > **comments on responses**
> >
> > Thanks. The reviewer has read all responses, which has generally addressed the concerns. By reading other reviewers' comments, the reviewer decided to keep the current score.

---

### Official Review · Reviewer_uTCz · 2023-06-29

**Soundness:** 3 good
**Presentation:** 3 good
**Contribution:** 3 good
**Rating:** 7
**Confidence:** 2

**Summary:**

This paper try to get rid of the limitations of current evaluation metrics for generative models and focuses on the perceptual fidelity of diffusion models. The authors conduct an extensive study using a wide range of image-based generative models across diverse datasets. They employ psychophysics to measure human perception of image realism and compare it with existing evaluation metrics. The paper reveals that commonly used metrics, such as FID, do not align with human evaluations of perceptual realism in diffusion models. The authors attribute this discrepancy to over-reliance on the Inception-V3 network. They propose using alternative self-supervised feature extractors to improve the evaluation of generative models. Additionally, the paper explores the issue of data memorization in generative models and highlights the limitations of current metrics in detecting memorization accurately. The authors release the generated image datasets, human evaluation data, and a library to compute 15 common metrics for 8 different encoders, facilitating further research in this area.

**Strengths:**

- The author comprehensively points out the issues with the existing evaluation metrics for generative models.
- The article presents a considerable number of insights and involves a substantial amount of work.

**Weaknesses:**

- Figure 1 lacks intuitiveness, and it is recommended to further improve it so that readers can grasp its meaning within a short period of time.

- The conclusion of the entire article is extensive, making it challenging to grasp the main points. It is suggested to enhance the writing by highlighting key conclusions, particularly in the introduction section.

- If possible, I suggest conducting a bias analysis on the 1000 paid participants, such as examining whether these participants are all students or individuals involved in the field of artificial intelligence, or if the age distribution is concentrated within a specific range, like 20-22 years old, and so on. This preliminary examination of bias can enhance the reliability of the experimental results. Of course, privacy concerns should be carefully addressed during the analysis.

Overall, I greatly appreciate this work and recommend acceptance of the article. Furthermore, I strongly suggest the authors share the code for analyzing the figures and charts in the article to facilitate further analysis by other researchers.

**Questions:**

see weakness.

**Limitations:**

Despite the considerable experimental analysis and insights presented by the authors in this paper, I suggest that they provide a detailed discussion of the limitations of the content being discussed. It would be preferable to dedicate a separate section at the end of the article specifically for this purpose.

---

> ### Author Rebuttal · Authors · 2023-08-10
>
> Thank you for your review, we appreciate the time and effort that went into evaluating our work, and we are glad you found it “extensive” and “comprehensive”. Please see our general rebuttal for an answer to your question about potential bias of the participants, where we provide additional examination of demographics. In short, the paid participants were very diverse in terms of age, gender, and ethnicity, but there is no significant connection between the demographic attributes and performance on our tests. We plan to release the demographic data along with the performance data, where we have the participants consent to do so, to instill confidence in the data we collected.
>
> As to your other points:
>
> - On Figure 1 lacking intuitiveness: Could you please elaborate what in the figure was unclear? We strive to provide our results in an easily-to-understand format and will happily update this figure accordingly to improve the clarity of the final paper.
>
> - On the main takeaways not being highlighted enough: Thank you for pointing this out, we will reformat the introduction in the final version of our paper so as to better summarize the key results of our paper.

---

> > ### Comment · Reviewer_uTCz · 2023-08-17
> > **re: Rebuttal by Authors**
> >
> > Thank you for your reply and I have also read the comments of other reviewers. I tend to keep my score.

---

### Official Review · Reviewer_6HZb · 2023-07-04

**Soundness:** 4 excellent
**Presentation:** 3 good
**Contribution:** 3 good
**Rating:** 7
**Confidence:** 5

**Summary:**

In this paper, the authors conduct a thorough investigation into the limitations of the Frechet Inception Distance (FID) metric for evaluating generative models. They address this issue by performing human evaluation and proposing a superior alternative for automatic generative model evaluation. Through dedicatedly designed experiments, the authors empirically demonstrate that FID, which relies on a pre-trained InceptionV3 model, exhibits a weak correlation with human evaluation. However, by replacing the InceptionV3 with a self-supervised model like DINO, the automatic evaluation becomes more closely aligned with human evaluation. The insights presented by the authors offer valuable guidance on the appropriate approach for evaluating image generative models.

**Strengths:**

[1] This paper addresses a highly significant issue within the generative modeling community. The authors' findings, demonstrating the lack of correlation between FID and human fidelity judgment, are particularly intriguing to me.

[2] The proposed alternative, FD_{DINO}, appears to be a sensible solution and holds promise for future evaluations of generative models.

[3] I think the provided benchmark tables are helpful for researchers trying to evaluate their generative models.

[4] In my opinion, the paper is well-written and makes a noteworthy contribution to the community.



**Weaknesses:**

[1] One potential concern is that all interpretations in the paper are based on the assumption that human evaluation is entirely correct, which may introduce some inherent risks.

[2] Providing further explanations of the evaluation metrics would enhance the paper's accessibility for individuals who are not experts in this particular field.

[3] Adding more detailed explanations about the differences between the authors' paper and closely related work [R1] would be beneficial for better understanding the novelty and contributions of the presented work.

[R1] M. Yang, C. Yang, Y. Zhang, Q. Bai, Y. Shen, and B. Dai. Revisiting the evaluation of image synthesis with GANs. arXiv preprint arXiv:2304.01999, 2023.

**Questions:**

please refer to the weaknesses.

**Limitations:**

I believe this paper delivers a crucial message to the community by emphasizing the inadequacy of FID as a metric for evaluating image generative models. Additionally, the authors propose a substantial improvement by changing from InceptionV3 to DINO as the backbone, which significantly mitigates this issue. While it may not completely resolve the problem, it remains a commendable contribution. Considering these contributions, I would give a score of 7 to this paper.

---

> ### Author Rebuttal · Authors · 2023-08-10
>
> Thank you for the detailed and helpful review. We are thrilled to hear you frame our insights as “valuable guidance on the appropriate approach for evaluating image generative models.” You rightly mentioned that including more details on the metrics in our paper will make it more accessible - please see the general response where we address that. We address your other concerns below:
>
> - “All interpretations in the paper are based on the assumption that human evaluation is entirely correct”:
>
>    This is essentially true. We carried out this study with the premise that human alignment is superior because humans are the users of the generated output. If a particular neural net architecture were the intended viewer, for instance, we would need to evaluate images differently. This is the premise of GANs, and as we demonstrate in our work, generative models that can fool a neural net classifier need not look the most realistic to humans.
>
>    Taking a practical perspective, we do believe that human perception is the best metric available for image fidelity. We highlight that we did our best to capture it with the cleanest crowd-sourced data possible through our experimental design informed by best practices from the literature [20, 19], namely by including a long training period and financial incentives for correct answers. This was also the reason why we structured our experiment to provide direct comparison between generated and training images, and only compared generative models that were trained on identical sets of images, as we wanted to separate out notions of fidelity to those of “style”. We will include this discussion in the final version of our paper.
>
> - "Further explaining the differences between our work and Yang et al. (2023) would be helpful":
>
>
>    We believe lines 172 to 181 of our manuscript do contain the fundamental differences between our work and Yang et al. (2023), which was concurrent with our study (the first public version was released April 4, less than two months before the NeurIPS submission deadline). Namely, this work’s trials targeted much more ambiguous tasks than our two alternative forced choice task. Rather than a direct measurement on the ability of humans to distinguish real and fake images, their main user study instead scored whether each generated image was “photorealistic”, yet participants had no knowledge or familiarity with data samples from the training set, and received no training or compensation. This lack of alignment between their human evaluation study and the goal of FID and similar generative metrics of quantifying distributional distances between real and generated images makes their analysis difficult to interpret. This difficult interpretation coupled with a much smaller and less rigorous human evaluation setup (they used fewer models and collected responses from 10 times fewer participants) are fundamental differences. Nevertheless, we are excited to see that others see value in evaluating generative metrics.

---

> > ### Comment · Reviewer_6HZb · 2023-08-18
> > **re: Rebuttal by Authors**
> >
> > The authors addressed all my concerns in the rebuttal. I am inclined to maintain my original score of 7.

---

### Official Review · Reviewer_uifn · 2023-07-06

**Soundness:** 3 good
**Presentation:** 3 good
**Contribution:** 3 good
**Rating:** 7
**Confidence:** 4

**Summary:**

This paper initially demonstrates which embedding space is similar to human evaluation criteria by utilizing various datasets and image generation models. It reveals that the embedding space of DINOv2 aligns most closely with the tendencies identified through a large-scale human survey. Moreover, it highlights the fact that most of the commonly used metrics do not align with human preferences. Additionally, the paper proposes approaches to address the issue of memorization, assessing whether the models are simply reproducing existing images.

**Strengths:**

The limitations of the traditional FID metric were widely recognized, and several papers were already trending towards using CLIP FD as the primary metric. While FID captures typical trends well, its clear limitations prompted this paper to explicitly demonstrate these constraints and advocate for the use of specific models' embedding spaces. The paper made significant contributions by conducting large-scale surveys to create metrics for human evaluation and by publicly sharing the generated images used in subsequent analyses, which is particularly valuable considering the time-consuming nature of Diffusion models. Furthermore, the paper addressed the issue of memorization and provided a quantitative representation of it.

**Weaknesses:**

The setting of using DINOv2's gradCAM is inevitably beneficial. It would be advantageous to include comparisons with metrics such as LPIPS as well.

**Questions:**

Is there an intention to make all the features publicly available?

Additionally, what distinguishes this paper from the next paper?
https://dreamsim-nights.github.io/


**Limitations:**

The novelty may not be significant; however, I believe the experiments conducted in this paper hold sufficient value.

---

> ### Author Rebuttal · Authors · 2023-08-10
>
> Thank you for the positive review and helpful feedback! To address your questions:
> 1. *Is there an intention to make all the features publicly available?*
>
>    Yes, we will be making all the features of the code publicly available, and all of the datasets. Hopefully this will help facilitate further research into evaluating deep generative models. (If you meant feature representations of the images, we did not intend to release these, but they are easy to compute using the code and images we will provide).
>
> 2. *What distinguishes this paper from https://dreamsim-nights.github.io/?*
>
>    In short, our paper is about evaluating deep generative models, while theirs focuses on image-to-image similarity similar to LPIPS. To be specific, they create a dataset of human-perceived image-to-image similarities and design an encoder that reflects these human judgements. On the other hand, our human trials focus on image quality rather than similarity, and our subsequent analysis uses these human judgements to evaluate current metrics for generative models. In other words, DreamSim focuses on obtaining an encoder which maps images that humans assess as similar to nearby points in latent space, whereas we focus on finding an encoder where distances between probability distributions on its latent space, such as FD, correlate with human judgment.
>
>     It is natural to wonder whether their improved DreamSim encoder might provide a better representation space for evaluating generative models than current self-supervised models such as DINOv2; however, when the paper came out, we ran a preliminary comparison and found it did not. For example, We found that the OpenCLIP-DreamSim fine-tuned encoder had nearly identical human/FD alignment as the original OpenCLIP encoder, and that their ensemble of encoders did not align with our human experiments as well as DINOv2 does. Unfortunately the DreamSim paper was first publicly released June 15, well after NeurIPS submissions had closed, so we were unable to describe these results in the paper (but will in the final version).

---

> > ### Comment · Reviewer_uifn · 2023-08-19
> >
> > Thank you. I decide to keep the current score.

---

### Official Review · Reviewer_xpCe · 2023-07-07

**Soundness:** 2 fair
**Presentation:** 3 good
**Contribution:** 3 good
**Rating:** 4
**Confidence:** 4

**Summary:**

This paper constructs an image dataset sampled from various generative models and scored by human participants in terms of their fidelity, and argues that existing metrics do not correlate well with this notion of fidelity. Then, it investigates how different choices of embedding space (i.e. different encoders) affect the metrics in terms of their ability to measure diversity and fidelity, and concludes that DINOv2 can be a superior choice of encoder. Finally, the paper studies whether existing memorization metrics can effectively measure memorization in high resolution datasets, and argues that none are reliable.

**Strengths:**

The paper asks interesting and important questions regarding the reliability of existing metrics of generative performance, and provides many valuable experiments with unique insights on the limitation of existing metrics for measuring fidelity and diversity of generative models. The proposed dataset of human evaluation can be a useful addition to the existing datasets for measuring and improving generative performance.

**Weaknesses:**

My main criticism of this paper is that it is not one coherent paper, rather parts of three different papers: one paper on constructing a dataset of human evaluation of generative models, a second paper on how the choice of encoders affects existing metrics, and a third paper on measuring memorization in high resolution datasets. As a result of this, many important results are shifted to Appendix (Appendices B and D in particular), and several observations are poorly explored and studied. I think each of these three directions deserves its own focused paper, carefully considering the caveats, and avoiding broad unjustified claims. I’ll elaborate on my specific concerns below.

***Regarding the dataset:***

1. I think the human experiment is flawed in the sense that “choose the fake sample” could result in choosing the unlikely samples as fake (confusing fakeness with unlikeliness). For example, a contorted low quality image of the common crow can be rated as “real” whereas a high quality image of an exotic rare parrot can be rated as “fake”. Therefore, a diverse model can receive a lower human error rate compared to a model of lower quality but less diversity. The experiments must propose a way to control for the effect of rarity, otherwise the dataset can be misleading.

2. Another issue with the human experiment is that it is unclear how the participants were selected. For example, if participants are mostly of one ethnicity, it is possible that they penalize diversity in place of fakeness. I understand that this issue is not easy to solve, but at the very least, I expect to see some effort in making sure the human participants are of sufficiently diverse backgrounds and origins.

3. For the results in Table 1, the difference being statistically significant alone is not enough, the amount of difference is itself of value. For example, a 1 percent difference is not as important, albeit statistically significant, as a 10 percent difference in human error rates. Reporting the actual numbers in a table similar to Table 1 will clarify this matter.

4. I am not sure how the main claim of “unfair treatment of diffusion models” in the title is justified. The brief explanation in L198-199 is too broad: “Coupling the results in Table 1 with the FID rankings in Figure 2, we conclude that current diffusion models produce the most realistic images, but are unfairly downranked by FID”. Could you be more specific about the connection from evidence to this claim?

***Regarding the metrics and encoders***

5. L225 claims that “We find that Inception does not perceive a holistic view of images even on its ImageNet training set”, however I don’t think the qualitative results in Fig 3 can support such a strong statement. A quantitative experiment is required to be able to claim this finding (I don’t see how the quantitative results in Appendix D.0.2 can support this claim, if so, please elaborate.)

6. L299 claims that: “We see this as strong evidence of a fundamental limitation of the use of the Inception network when computing FID”. I don’t see how this shows a fundamental limitation. To me, it only shows that for some generative models, FID seems to not correlate well with per class Vendi score. If the intention is to show that FID does not correlate with Vendi in general, why not explicitly report correlation? But even then, can you elaborate what “fundamental” means in this context? I do understand that different embeddings focus on different features, that is given by design, but that one embedding is fundamentally weaker than another needs a more formal argument.

7. The conclusion that DINOv2 is better than Inception is not well justified to me. The main evidence for this strong claim seems to be Figure 6, which is at best a motivating observation (reporting the correlation coefficients could help make Figure 6 more substantial). A systematic ablation study for the encoder on the effect of a) the number of training classes, b) various loss functions, and c) various architectures (only changing one factor at a time and fixing the other factors), on the performance of metrics can be more conclusive regarding which encoder is more reliable.

8. For a paper about metrics, the definition of considered metrics should be restated to facilitate the clarity and readability of the paper. This can be done for the main metrics in the body of the paper (FD, P/R, Vendi) and for others in the appendix. The current definitions in Appendix B lack the exact mathematical definitions except for FD.

***Regarding memorization***

9. The paper does not discuss why each of the considered memorization metrics fail, that is, what assumptions in the definition of each metric deviates from practice and potentially causes the observations. The lack of such discussions and followup experiments to pinpoint the cause, makes the empirical results inconclusive in general.

**Questions:**

See the weaknesses section.

**Limitations:**

See the weaknesses section, particularly on the collection of the dataset.

---

> ### Author Rebuttal · Authors · 2023-08-09
>
> Thank you for your detailed review. We are happy our paper “provides many valuable experiments with unique insights on the limitation of existing metrics”. We believe the replies below address your concerns, and kindly ask that if you agree, to consider raising your score.
>
> On our work being “three different papers”: Given the long history of generative evaluation, any proposed metric must be demonstrably superior on a wide range of quantitative and qualitative evaluations. The three components of our paper are a cohesive whole to accomplish this goal, and removing any would make the paper insufficient to build community trust and support.
>
> The first part – constructing the dataset – would be an incomplete contribution in terms of novelty, as we follow (yet improve) on the methodology of HYPE [91]. The resulting novel insight that FID does not correlate well with human judgment demands a subsequent investigation of alternate encoders and metrics. This second part is the most natural way to search for a new evaluation metric that correlates better with human evaluators.
>
> The third part resolves two critical caveats: Section 4.2.1 dismisses the concern that FID fails due to humans not measuring diversity, and Section 4.3 establishes whether the best generative models “cheated” their human alignment through memorization (a valid concern since FD does not detect it). We thank you for pointing out an area where we can improve our writing and will refine this narrative in the final version of our paper.
>
> 1\. We respectfully disagree that this is a flaw in our experimental design - our methodology accounts for and negates such an effect: each participant sees a total of 250 images from ImageNet (which has 1000 distinct classes), making it essentially impossible to learn any diversity within any class, and thus forcing a focus on fidelity. Using your crow/exotic parrot example, *in ImageNet these would be different classes, and generative models synthesized class-conditional images that closely resemble the semantic information of a class* (see Figure 20 for proof). Therefore, the crows/parrots from different generative models will have varying degrees of fidelity, which will be measured by the human error rate. Additionally, if participants were confusing fakeness with unlikeliness, *this effect would be consistent across models and therefore would not alter rankings*. Another control for any such effects is the training phase (Figure 18), in which participants gain a sense for the diversity of images before starting the test.  Finally, we highlight that our experimental design is highly similar to the widely adopted HYPE design, and was aided by experts in psychophysics precisely to avoid these types of flaws and measure human perception of realism accurately.
>
> 2\. & 8. See general rebuttal.
>
> 3\. & 4. We believe that the titular claim of “unfair treatment of diffusion models” is justified. Figure 2 clearly shows that diffusion models score the highest human error rates, and that GANs often score lower human error rates yet achieve a better FID ranking. Table 1 summarizes the statistical significance, and we report the exact values and error bars in Table 11. We agree that the word “unfair” in lines 198-199 is not yet justified at that part of the paper though, and will remove it. Nonetheless, the remainder of the paper investigates potential causes for this misalignment of FID and human evaluation, and ultimately builds up sufficient evidence to support the “unfair” conclusion.
>
> 5\. Thank you - we agree that this is a qualitative claim, which we will soften in the final version. We point out however that it is consistent with our quantitative analysis and that in Appendix C of [47], which we will also make clear in the paper.
>
> 6\. We believe this viewpoint stems from a misunderstanding of the goal of our diversity experiments, which we will clarify in the final version of our paper. The lack of correlation between FID and human evaluators could be due to either (a) FID being fundamentally flawed, or (b) FID accounting for diversity where human evaluators do not - perhaps diffusion models trade diversity for fidelity. The goal of our diversity experiments is to rule out option (b) (i.e. checking that differences in Vendi score do not explain flips in ordering between FID and human evaluators – this is not equivalent to correlation).  Our experiments show that images from diffusion models have good diversity, which, combined with results from our human experiments, allows us to rule out option (b) and conclude option (a).
>
> 7\. On Fig 6 please see the point above. On an ablation study: what you describe would require retraining self-supervised foundational models on internet-scale data, which would be prohibitively expensive (DINOv2 alone required 200k GPU-days [58]) and out of scope for this work. We also note that the self-supervised foundational encoders that we use were trained with a variety of objectives and architectures.
>
> 9\. Given the increasing attention placed on memorization of generative models [11, 58] we included our results which show that memorization metrics can not be relied on at this stage, which presents a natural call for further study. Recent work [C] (made publicly available after the NeurIPS submission deadline) has shown that various pathologies associated with precision and recall are due to the curse of dimensionality as a byproduct of nearest neighbors. These findings are consistent with our own observations of precision and recall, and we hypothesize that this could cause the failure of memorization metrics. We will include this discussion in the final version of our manuscript.
>
> [C] Emergent Asymmetry of Precision and Recall for Measuring Fidelity and Diversity of Generative Models in High Dimensions. ICML 2023.

---

> > ### Comment · Reviewer_xpCe · 2023-08-11
> > **Thank you for your response**
> >
> > Regarding “dataset”, “metric” and “memorization” sections being part of a coherent paper, I respectfully disagree with the authors: the “metric investigation” section, uses the dataset to study several metrics, but rather than focusing on why existing metrics do not correlate with Human Error and if that reveals a limitation in the Human Error metric itself or the existing metrics (since these metrics are backed by their own user studies and theories), it quickly takes Human Error as ground truth and moves on to improving metrics (hence a separate paper in my opinion); b) the “memorization” section does not use the proposed dataset or the Human Error metric, it does not even mention them.
> >
> > 1.1. You claim “each participant sees a total of 250 images from ImageNet (which has 1000 distinct classes), making it essentially impossible to learn any diversity within any class, and thus forcing a focus on fidelity”, the issue is not that your participants will learn to focus on diversity or not, it is that they might already associate rarity with fakeness, so you need to provide evidence for the claim that “... forcing a focus on fidelity”: one way to do so is to report Human Error rate on rare real samples versus common real samples (e.g. using Rarity Score, Han 2022), and see if the error rate is the same on both sets. Lack of such experiments, and any discussion of this potential issue in the paper, is why I do not think you can claim that “our methodology accounts for and negates such an effect” at present.
> >
> > 1.2. The parrot/crow is of course just an extreme example to clarify my point, within each ImageNet class you can find unlikely as well as likely samples.
> >
> > 1.3. You claim “if participants were confusing fakeness with unlikeliness, this effect would be consistent across models and therefore would not alter rankings”, I don’t see why the effect would not alter rankings: if participants confuse fakeness with rarity, they will make less mistakes on models that generate more rare samples (because they will just flag them as fake based on the samples being unlikely), so your score would unfairly rank models that are more diverse worse.
> >
> > 1.4. I don’t think being similar to HYPE or aided by psychophysicist answers any of my very specific concerns. What is lacking here is quite straightforward, as I explained in 1.1, you should provide a control experiment, otherwise the dataset – which I want to emphasize that I think is valuable and interesting – will incentivise a series of misleading and incorrect followup works.
> >
> > 2.. I don’t understand what this means: “The Normalized Error Rate accounts for the varying difficulty of tasks over different combinations of (dataset, model).”, please elaborate.
> >
> > 3.1. You claim “Figure 2 clearly shows that diffusion models score the highest human error rates, and that GANs often score lower human error rates yet achieve a better FID ranking”, but in Figure 2 CIFAR10 the diffusion model is ranking best in terms of FID too, so FID is not unfair. Same in ImageNet. My point is by just looking at Figure 2, there is no concrete evidence to back “unfairness”. You need to be more specific about what is the mathematical definition of “unfair” in your work, and how you measure it. For example, you could define unfair as low correlation between FID and Human Error Rate, and then report the correlation coefficients and claim that the correlation is higher for GANs, but lower in Diffusions.
> >
> > 3.2. You claim “Table 1 summarizes the statistical significance”, yet you report no significance test results, it is unclear what the ordering in this table are based on. Table 11 also does not clearly show “unfairness” towards diffusion models.
> >
> > 6.. I understand your motivation, but what I still do not understand is what “fundamental” means in this context. In any way, I do not consider this a main concern, I acknowledge that whether something is a “fundamental” flaw, or simply a lack of correlation between some metrics, is subjective. I appreciate the additional explanations by the authors.
> >
> > 7.. If computational restrictions do not allow you to sufficiently support the claim that DINOv2 is better than Inception, please avoid making that claim in your abstract. If Figure 6 is able to sufficiently support this claim, please elaborate.
> >
> > 8.. I agree that there might be interesting connections, but lack of a study on those connections makes the memorization results inconclusive.
> >
> > Han, Jiyeon, et al. "Rarity score: A new metric to evaluate the uncommonness of synthesized images." ICLR 2022.

---

> > > ### Author Response · Authors · 2023-08-14
> > > **Second Reply (1/2)**
> > >
> > > Thank you for replying quickly and engaging in discussion, we appreciate the added clarification and hope to continue discussion on any lingering concerns.
> > >
> > > 0. You are correct that we take human assessment of fidelity as ground truth, we will make this more explicit in the final version of our paper. We believe this is an extremely reasonable assumption though, as humans are the end users of these models. The goal of our diversity experiments was to verify if this assumption holds: to see if diversity (which we do not believe humans are particularly good at detecting, nor were our experiments designed to detect) could potentially explain discrepancies between FID and human error rate (HER; see point 6 of our rebuttal). The memorization part of the paper indeed does not use our FD_DINOv2 score or HER because neither FD metrics nor HER are expected to detect memorization (note we include the value of all memorization metrics for DINOv2 in Table 11). Our main goal with these experiments is not to further validate the FD_DINOv2 score itself, but rather to ensure models at the top of the FD_DINOv2 leaderboard did not “cheat” their way in by memorizing training data. We see this as a relevant step to establish community trust in the leaderboard. We emphasize once again that we will update our manuscript to better convey this narrative.
> > >
> > > 1. Thank you for clarification, and for suggesting the Rarity Score (RS) of Han et al. 2022. We agree this analysis, which we have now performed, provides more evidence for whether participants confuse fakeness with unlikeliness. Our rebuttal pdf cannot be updated and no further attachments can be added, so here we describe the results in detail and outline what we will add to the final paper.
> > >
> > >     *Experiment*: We focus on your suggestion of “Error rate on rare real samples versus common real samples”. For each of the 2000 real images we used from each dataset (evaluated by an average of 13 humans), we determined the fraction of humans that labeled it as fake, as well as its RS. We performed the calculation using both Inception and DINOv2 to quantify the dependency of RS on the embedding space.
> > >
> > >     *Results*: We find no correlation between HER and RS on ImageNet and FFHQ. On CIFAR-10 and LSUN-Bedroom we find a small (e.g. see Table 1 in https://www.ncbi.nlm.nih.gov/pmc/articles/PMC3576830/) but statistically significant correlation, which we identify as driven by dataset issues: the non-zero correlation is caused by a very small percentage of “real” images which are clearly taken from 3D-generated scenes (instead of bedroom photographs in LSUN-Bedroom), or from 2D-generated scenes or low quality (extremely blurry) images (in CIFAR-10). These results show that 1.) humans are more likely to label generated scenes as fake/generated (LSUN-Bedroom, CIFAR-10), and 2.) humans are more likely to label low-quality images as fake/generated (CIFAR-10). Such images have a higher than average RS, and hence *the small correlation between human evaluation and RS on CIFAR10 and LSUN-Bedroom is due to humans properly identifying these dataset issues*. We find that removing just 6% of the “fakest” (as measured by humans) real images on LSUN-Bedroom removes the correlation of RS and HER - quantitative proof that the small correlation is driven by dataset issues, and not due to humans associating diversity with fakeness. Rare defects in the training set are not enough to affect our results: being so rare (~6%) means they barely affect the average error rate; and *this training-set effect is the same for every generative model evaluation and thus does not change their rankings*. We have prepared a few additional scatter-plots and image visualizations for the final version of our paper (the lack of correlation is visually evident), and summarize the correlations and their significance on the table below. We will also include the rarity score in our public codebase.
> > >
> > > Table 12: Pearson correlation of the fraction of humans that labeled a real image as fake, and the Rarity Score (RS; Han 2022) of that image. The RS can only be determined for images that fall “on manifold”
> > >
> > > |      |   | % on manifold | r      | p-value | r (94%) | p-value (94%) |
> > > |--------------|-----------|-----------------|--------|---------|----------|----------------|
> > > | CIFAR10      | Inception | 76            | 0.28  | 0.00   | 0.216    | 0.000          |
> > > |      | DINOv2    | 88            | 0.062  | 0.01   | 0.00   | 0.86          |
> > > | ImageNet     | Inception | 82            | -0.03 | 0.16   | -0.00   | 0.92          |
> > > |   | DINOv2    | 93            | 0.01  | 0.73   | -0.01   | 0.62          |
> > > | LSUN-Bedroom | Inception | 70           | 0.11  | 0.00   | 0.05    | 0.11          |
> > > | | DINOv2    | 90            | 0.10  | 0.00   | 0.02    | 0.39          |
> > > | FFHQ         | Inception | 79            | -0.03 | 0.29   | -0.02   | 0.36          |
> > > |         | DINOv2    | 90            | -0.02 | 0.43   | -0.04   | 0.10          |

---

> > > ### Author Response · Authors · 2023-08-14
> > > **Second Reply (2/2)**
> > >
> > > 2\. Tasks (defined as dataset/model pairs) have varying difficulties - if a generative model is very poor, humans have low error rates as they easily distinguish fake samples from real ones. Thus we cannot average error rates across tasks, as these numbers are not directly comparable. To avoid this issue, if $x_i$ is the error rate of participant $i$ at task $t(i)$ (the task performed by participant $i$), the normalized score is given by $(x_i - \mu_{t(i)}) / \sigma_{t(i)}$, where $\mu_{t(i)}$ and $\sigma_{t(i)}$ are the mean and standard deviation, respectively, of the error rate at task $t(i)$ across participants. Normalized scores are comparable across participants and tasks.
> > >
> > > 3\.1 Note that we discuss CIFAR-10 being an exception in Fig 2 on L249-252. Nonetheless, our argument is *not* that Fig 2 shows unfair treatment of diffusion models by FID, but the figure does show that FID and human error rate (HER) are uncorrelated (except on CIFAR-10). This is what we meant in point (3 & 4) of our rebuttal: we agree that at this point of the paper, there is not yet enough evidence to call the lack of correlation between FID and HER unfair. However, we believe that in the rest of the paper, we present enough evidence to call this unfair: (a) the Inception network focuses on the wrong aspects of images (again, we do assume here that humans provide ground truth, and understand “wrong” as “unlike humans”), which is shown in Sec 4.1; (b) metrics which *do* focus on the correct parts of an image, like DINOv2, have a much stronger correlation with HER, which is shown in Sec 4.2; and (c) diversity is *not* the reason FID is uncorrelated with HER, which is shown in Sec 4.2.1. Together, all this evidence does allow us to say the treatment given by FID to diffusion models is indeed unfair, even if we have no formal mathematical definition of what “unfair” means. Again, we will more clearly convey this narrative in the final version of the paper.
> > >
> > > 3\.2 We apologize for poor phrasing in our rebuttal. Table 11 (sorted in decreasing order of FD_DINOv2) summarizes the mean and standard error HER (which is what you asked us for in your original review), not statistical significance of the tests from Table 1 (which as mentioned in L191-193, are all highly significant). Again, we do not see Table 11 as showing diffusion models are treated unfairly by FID: we believe our paper as a whole establishes this conclusion, not just the lack of correlation between FID and HER or the numbers in this table.
> > >
> > > 6\. We will happily change the phrasing from “fundamental” to something less ambiguous.
> > >
> > > 7\-1 We believe the claims in our paper allow us to say that DINOv2 is better than Inception, *but not just because of Fig 6*: (a) Figs 2 and 4 show that FID does not correlate with HER, whereas FD_DINOv2 does; (b) Fig 3 shows that DINOv2 focuses on more human-relevant aspects of images than Inception; (c) Fig 5 shows that diversity does *not* explain differences between HER and FID: for example, let’s focus on LDM (a diffusion model) and StyleGAN-XL (a GAN). LDM has both better HER than StyleGAN-XL (Fig 2), and has diversity that more closely matches the data’s (Fig 5), yet has a worse FID (Fig 2). FD_DINOv2 does not exhibit this “unfair” ranking of LDM and StyleGAN-XL. (d) Fig 6 shows that diversity *does* explain some differences between HER and FD_DINOv2 score (e.g. DiT-guided has better HER than DiT, yet has worse FD_DINOv2 score as its diversity is a much worse match to the data’s than DiT’s). Together, all this evidence does let us say DINOv2 is better than Inception. Again, we will make sure this line of thinking is clearer in the final version of our paper.
> > >
> > > 7\-2 Our claims about computational restrictions in the rebuttal are simply about the impossibility of a full ablation study of our own foundational models to understand exactly *why* they outperform Inception, and have no bearing on whether the evidence in our paper is enough to “sufficiently support the claim that DINOv2 is better than Inception”. We specifically designed the selection of encoders in our study to shed light on what components of training/architectures benefit generative evaluation. As discussed in L214-240, an improved supervised model (ConvNeXt) indicates whether more modern supervised models share the same issues as Inception, we chose both self-supervised CNNs and ViTs trained with different objectives, models trained on ImageNet and those trained on internet-scale datasets, and performed a study of the effect of architecture in App B.4.1 (6 varieties of CLIP and 4 of DINOv2).
> > >
> > > 8\. We agree our experiments on memorization are inconclusive *in terms of establishing the reason or a solution for the failures of memorization metrics*, but not in terms of establishing that models at the top of the DINOv2 leaderboard did not “cheat their way to the top” through memorization, which was the main objective of these experiments. We will further clarify this in the paper.

---

> > > > ### Comment · Reviewer_xpCe · 2023-08-20
> > > >
> > > > Thank you for the additional results, I will slightly increase my score in recognition of the additional results. However, my concerns regarding stating claims without clear definitions and measures remain (points 3 and 4 in my response). I also disagree with the authors' justification of treating human rating as ground truth, namely that: "We believe this is an extremely reasonable assumption though, as humans are the end users of these models." Humans are not always the end users, for example: the use of generative models as augmentations to improve performance of classifiers, or as debiasing methods to improve the fairness of classifiers, in both cases the end user is the downstream classifier which does not necessarily benefit from generative models that humans rate as being better.

---

> > > > > ### Author Response · Authors · 2023-08-21
> > > > > **Third Reply**
> > > > >
> > > > > Thank you very much for increasing your score and for continuing to discuss!
> > > > >
> > > > > Our view is that any attempt to compare two probability distributions (in this case the model’s and the ground truth) using a single scalar score must make inherent trade-offs as to how different attributes of these distributions are “weighted” when computing the score. We also believe that a “weighting” being sensible is task-dependent, and thus we do not think a single metric can be designed to be useful at every task. In this sense, we agree with you that human error rate and FD_DINOv2 need not be useful metrics for every task (and the same is true of any metric). We also agree that data augmentation for classification accuracy is an example task where FD-based metrics do not make sense: the distribution of augmented images need not be similar to the target distribution (e.g. AugMix [A]), yet can still help accuracy. Similarly, if attempting to debias a model which correctly learned a distribution from a biased dataset, FD-based metrics need not be appropriate. The same is true of dataset distillation [B], where the distribution of distilled images can be very different from the true distribution of images, yet still be useful.
> > > > >
> > > > > Yet, one of the main uses of image-based generative models is to generate realistic images, which is what we meant by saying humans are the end users. FID (which is not used as a metric for data augmentation, bias, or dataset distillation) is often thought about as a way to compare distributions in such a way that realism is “highly weighted”. Our claims on FD_DINOv2 are not that it should be used as the one and only metric to evaluate generative models, but that it is much better than FID as a metric of generative quality that “highly weights” realism. We will happily make this point clearer in the final version of the paper.
> > > > >
> > > > >
> > > > > [A] *AugMix: A Simple Data Processing Method to Improve Robustness and Uncertainty*, Hendrycks et al., ICLR 2020
> > > > >
> > > > > [B] *Dataset Distillation by Matching Training Trajectories*, Cazenavette et al., CVPR 2022

---

### Author Rebuttal · Authors · 2023-08-10

We thank all the reviewers for their thoughtful feedback and the time they spent assessing our work. We are very encouraged by the largely positive feedback, including that our paper provides “significant contributions” (uifn), “presents a considerable number of insights” (uTCz), and “delivers a crucial message to the community” (6HZb). Below we address themes raised by more than one reviewer, and otherwise reply to each reviewer individually.

1. On selection, diversity, and potential bias of participants (xpCe, uTCz): Thank you for bringing this up. We agree that it is relevant to discuss, and that doing so will enhance our paper. Our attached rebuttal pdf includes demographic information about the participants (only those who explicitly agreed to provide and to allow us to use this information are included). It is clear that the set of participants is diverse, and that the results of the human evaluation experiment exhibited no biases from any of the diverse groups of participants (as the difference in means of normalized error rates between demographic groups is much smaller than the corresponding standard deviations). While our participants were recruited from a global crowd-sourcing platform, we did not explicitly select participants based on diversity due to various reasons:
    - In the first experiments we ran (on CIFAR-10), we found no systematic difference in participant’s performance based on demographics, and we thus decided to not filter based on demographics on subsequent experiments. This is indeed confirmed once again in the post-hoc analysis (on all datasets) included on the rebuttal pdf.
    - Not only do we not observe an empirical difference in responses based on demographics, but there is also no theoretical reason to believe there might be one. The dominant perspective in visual cognition is that vision is impenetrable, which is another way of saying bottom-up, or not influenced by cognition [A,B]. Another way of thinking about this would be to imagine the architecture of the visual system as a CNN. The processing is predominantly in the direction of the forward pass, or, from the retinae to the primary visual cortex to areas of higher cognition. Backward connections exist but serve the stability of perception rather than the content. In other words, a person’s culture or ethnicity is unlikely to affect the forward pass.
        - [A] Pylyshyn, Z. (1999). Is vision continuous with cognition?: The case for cognitive impenetrability of visual perception. Behavioral and brain sciences, 22(3), 341-365.
        - [B] Firestone, C., & Scholl, B. J. (2016). Cognition does not affect perception: Evaluating the evidence for “top-down” effects. Behavioral and brain sciences, 39, e229.
    - Conventionally, studies in psychophysics do not report demographic information on the participant pool beyond age and gender. This largely reflects widespread belief in the views expressed in the point above. As of August 4th 2023, a cursory examination of the 20 most recent open access articles in *Visual Cognition*, a popular journal dedicated to studies in visual psychophysics, confirms this. 17 of the 20 were empirical studies. Of these, all report age and gender. Only two report any demographic information on the participants’ ethnicity. Both were studies on face perception, which is a subfield where exposure to faces of different ethnicities has known effects on performance. In other words, unless there is an *a priori* reason to expect an effect of ethnicity or other demographic variable (other than age and gender), it would be unusual to report them. Nonetheless, we are happy to report additional demographic information anyway in the final version of our paper.

    In summary, our participants are diverse in spite of diversity not being a source of concern for bias in our study, and we find no systematic difference in participant’s performance based on demographics.

2. On including more detailed definitions of the metrics used in our paper (xpCe, 6HZb): Thank you for pointing out this area of improvement. In the final paper we will expand upon the current descriptions to add the full mathematical descriptions and explanations for each of the metrics used throughout the paper.

---

### Decision · Program_Chairs · 2023-09-21

**Decision:**

Accept (poster)

**Comment:**

This submission was reviewed by five knowledgeable referees. The reviewers raised concerns w.r.t. (1) the number of topics covered by the paper, which made the presentation slightly confusing (xpCe, uTCz); (2) the effect of rarity in the user study (xpCe); (3) the user's study participant selection (xpCe, uTCz); (4) missing comparisons with LPIPS (uifn); and (5) the positioning of the contribution w.r.t. prior work (uifn, 6HZb). The reviewers found the claims of the paper too strong given the evidence (xpCe), missed an in-depth limitations discussion (uTCz), and raised questions about the computation of the human error (SaoU). The rebuttal addressed most of the reviewers concerns by extensively discussing the participant selection process, the participant diversity, and the human error rate metric; by adding results to strengthen the claims; and by positioning their work w.r.t. the papers shared by the reviewers. The reviewers acknowledge the effort put by the authors in the rebuttal and discussion, the AC also acknowledges this effort. After discussion, most concerns appear addressed and 4/5 reviewers lean towards acceptance. Reviewer xpCe still has concerns w.r.t. the overstated subjective claims, and the human rating which they find not necessarily a good ground truth. Although the AC understands the arguments put forward by reviewer xpCe (humans are not the only end users of generative models), the AC also agrees with the argument of assessing the realism of images. This paper also received ethics reviews. Ethics reviewers raised a concern that the paper had not gone through IRB. The authors stated that their institution does not have an IRB process and appealed to the NeurIPS 2023 guidelines making authors at institutions with no IRB process exempt. The AC and SAC discussed this issue and side with the authors.

The AC recommends to accept and expects the authors to reword the overstated claims. In particular, the AC expect the authors to tone down the claims w.r.t. unfair treatment of diffusion models, Dino-v2, and Inception-v3, as suggested by the reviewers. The AC also encourages the authors to make the conclusions more concise in the final version of their manuscript.